# Provable Model-based Nonlinear Bandit and Reinforcement Learning: Shelve Optimism, Embrace Virtual Curvature

**Kefan Dong**
Stanford University
kefandong@stanford.edu

**Jiaqi Yang**
Tsinghua University
yangjq17@gmail.com

**Tengyu Ma**
Stanford University
tengyuma@stanford.edu

## Abstract

This paper studies model-based bandit and reinforcement learning (RL) with non-linear function approximations. We propose to study convergence to approximate local maxima because we show that global convergence is statistically intractable even for one-layer neural net bandit with a deterministic reward. For both nonlinear bandit and RL, the paper presents a model-based algorithm, Virtual Ascent with Online Model Learner (ViOlin), which provably converges to a local maximum with sample complexity that only depends on the sequential Rademacher complexity of the model class. Our results imply novel sample complexity bounds on several concrete settings such as linear bandit with finite or sparse model class, and two-layer neural net bandit. A key algorithmic insight is that optimism may lead to over-exploration even for two-layer neural net model class. On the other hand, for convergence to local maxima, it suffices to maximize the virtual return if the model can also reasonably predict the gradient and Hessian of the real return.

## 1 Introduction

Recent progresses demonstrate many successful applications of deep reinforcement learning (RL) in robotics [49], games [7, 66], computational biology [56], etc. However, theoretical understanding of deep RL algorithms is limited. Last few years witnessed a plethora of results on linear function approximations in RL [80, 65, 40, 73, 71, 19, 3, 32], but the analysis techniques appear to strongly rely on (approximate) linearity and hard to generalize to neural networks.

The goal of this paper is to theoretically analyze model-based nonlinear bandit and RL with neural net approximation, which achieves amazing sample-efficiency in practice (see e.g., [37, 10, 30, 31, 12]). We focus on the setting where the state and action spaces are continuous.

Past theoretical work on model-based RL studies families of dynamics with restricted complexity measures such as Eluder dimension [60], witness rank [68], the linear dimensionality [77], and others [58, 42, 17]. Implications of these complexity measures have been studied, e.g., finite mixture of dynamics [6] and linear models [64] have bounded Eluder dimensions. However, it turns out that none of the complexity measures apply to the family of MDPs with even barely nonlinear dynamics, e.g., MDPs with dynamics parameterized by all one-layer neural network with a single activation unit (and with bounded weight norms). For example, in Theorem 5.2, we will prove that one-layer neural nets do not have polynomially-bounded Eluder dimension.[1] (See more evidence below.)

The limited progress on neural net approximation is to some extent not surprising. Given a deterministic dynamics with *known* neural net parameters, finding the best parameterized policy still involves optimizing a complex non-concave function, which is in general computationally intractable. More

---

[1]This result is also proved by the concurrent Li et al. [50, Theorem 8] independently.

35th Conference on Neural Information Processing Systems (NeurIPS 2021).

fundamentally, we find that it is also *statistically intractable* for solving the one-hidden-layer neural net bandit problem (which is a strict sub-case of deep RL). In other words, it requires exponential (in the input dimension) samples to find the global maximum (see Theorems 5.1). This also shows that conditions in past work that guarantee global convergence cannot apply to neural nets.

Given these strong impossibility results, we propose to reformulate the problem to finding an approximate local maximum policy with guarantees. This is in the same vein as the recent fruitful paradigm in non-convex optimization where researchers disentangle the problem into showing that all local minima are good and fast convergence to local minima (e.g., see [28, 26, 27, 29, 48]). In RL, local maxima can often be global as well for many cases [4].[2]

Zero-order optimization or policy gradient algorithms can converge to local maxima and become natural potential competitors. They are widely believed to be less sample-efficient than the model-based approach because the latter can leverage the extrapolation power of the parameterized models. Theoretically, our formulation aims to characterize this phenomenon with results showing that the model-based approach's sample complexity mostly depends (polynomially) on the complexity of the model class, whereas policy gradient algorithms' sample complexity polynomially depend on the dimensionality of policy parameters (in RL) or actions (in bandit). Our technical goal is to answer the following question:

> Can we design algorithms that converge to approximate local maxima with sample complexities that depend *only and polynomially* on the complexity measure of the *dynamics/reward* class?

We note that this question is open even if the dynamics hypothesis class is finite, and the complexity measure is the logarithm of its size. The question is also open even for nonlinear bandit problems (where dynamics class is replaced by reward function class), with which we start our research. We consider first nonlinear bandit with *deterministic* reward where the reward function is given by $\eta(\theta, a)$ for action $a \in \mathcal{A}$ under instance $\theta \in \Theta$. We use sequential Rademacher complexity [62, 63] to capture the complexity of the reward function $\eta$. Our main result for nonlinear bandit is stated as follows.

**Theorem 1.1** (Informal version of Theorem 3.1)**.** *Suppose the sequential Rademacher complexity of a loss function class (defined later) induced by the reward function class $\{\eta(\theta, \cdot) : \theta \in \Theta\}$ is bounded by $\sqrt{R(\Theta)T}\mathrm{polylog}(T)$. Then, there exists an algorithm (ViOlin, Alg. 1) that finds an $\epsilon$-approximate local maximum with $\widetilde{\mathcal{O}}(R(\Theta)\epsilon^{-8})$ samples.*

In contrast to zero-order optimization, which does not use the parameterization of $\eta$ and has a sample complexity depending on the action dimension, our bound only depends on the complexity of the reward function class. This suggests that our algorithm exploits the extrapolation power of the reward function class. To the best of our knowledge, this is the first action-dimension-free result for both linear and nonlinear bandit problems. More concretely, we instantiate our theorem to the following settings and get new results that leverage the model complexity (more in Section 3.1).

1. Linear bandit with finite parameter space $\Theta$. Because $\eta$ is concave in action $a$, our result leads to a sample complexity $\mathcal{O}(\mathrm{poly}(\log|\Theta|, 1/\epsilon))$ for finding an $\epsilon$-approximate optimal action. In this case both zero-order optimization and the SquareCB algorithm in Foster, Rakhlin [24] have sample complexity/regret that depend on the dimension of action space $d_{\mathcal{A}}$.
2. Linear bandit with $s$-sparse or structured instance parameters. Our algorithm ViOlin achieves an $\mathcal{O}(\mathrm{poly}(s, 1/\epsilon))$ sample complexity when the instance/model parameter is $s$-sparse and the reward is deterministic. The sample complexity of zero-order optimization depends polynomially on $d_{\mathcal{A}}$. Carpentier, Munos [9] achieve a stronger $\widetilde{\mathcal{O}}(s\sqrt{T})$ regret bound for $s$-sparse linear bandits with actions set $\mathcal{A} = S^{d-1}$. In contrast, our ViOlin algorithm applies more generally to any structured instance parameter set. Other related results either leverage the rather strong anti-concentration assumption on the action set [72], or have implicit dimension dependency [33, Remark 4.3].
3. Two-layer neural nets bandit. Our algorithm finds an $\epsilon$-approximate local maximum $\widetilde{\mathcal{O}}(\mathrm{poly}(1/\epsilon))$. Zero-order optimization can also find a local maximum but with $\Omega(d_{\mathcal{A}})$ samples.

---

[2]The all-local-maxima-are-global condition only needs to hold to the ground-truth total expected reward function. This potentially can allow disentangled assumptions on the ground-truth instance and the hypothesis class.

Optimistic algorithms in this case have an exponential sample complexity (see Theorem 5.3). Moreover, when the second layer of the ground-truth network contains all negative weights and the activation is convex and monotone, all local maxima are global because the reward is concave in the input (action) [5].

The results for bandit can be extended to model-based RL with deterministic nonlinear dynamics and deterministic reward.

**Theorem 1.2** (Informal version of Theorem 4.4). *Consider RL problems with deterministic dynamics class and stochastic policy class that satisfy some Lipschitz properties. Suppose the sequential Rademacher complexity of the $\ell_2$ losses for learning the dynamics is bounded by $\sqrt{R(\Theta)T}\operatorname{polylog}(T)$. Then, ViOlin for RL (Alg. 2) finds an $\epsilon$-approximate local maximum policy with $\widetilde{\mathcal{O}}(R(\Theta)\epsilon^{-8})$ samples.*

To the best of our knowledge, this is the first model-based RL algorithms with provable finite sample complexity guarantees (for local convergence) for general nonlinear dynamics. The work of [55] is the closest prior work which also shows local convergence, but its conditions likely cannot be satisfied by any parameterized models (including linear models). We also present a concrete example of RL problems with nonlinear models satisfying our Lipschitz assumptions in Example 4.3 of Section 4, which may also serve as a testbed for future model-based deep RL analysis. Other prior works on model-based RL do not apply to one-hidden-layer neural nets because they conclude global convergence which is not possible for one-hidden-layer neural nets in the worst case.

**Technical novelty: exploring by model-based curvature estimate.** The main challenge is that the optimism-in-face-of-uncertainty exploration principle seems to be too aggressive—for a nonlinear model class, even if the ground-truth model is linear, optimistic exploration leads to exponentially large model cumulative prediction errors and exponential sample complexity (see Theorem 5.3). We use an online learning oracle to learn the dynamics, which can dramatically reduce the model prediction errors by hedging the risks. However, now, choosing actions or policies by maximizing virtual return may lack exploration. The main novelty of our algorithm is that we augment the online learning loss with gradient and Hessian estimation errors. The key insight is that, in order to ensure sufficient exploration for converging to local maxima, it suffices for the model to predict the virtual return, its gradient and Hessian reasonably accurately. We refer to the approach as "model-based curvature estimate". Our algorithm alternates between maximizing virtual return (over action or policy) and learning the model parameters by an online learner.

Because the algorithm leverages model extrapolation, the sample complexity of model-based curvature prediction depends on the model complexity instead of action dimension in the zero-optimization approach. Foster, Rakhlin [24] also propose algorithms that do not rely on UCB—their exploration strategy either relies on the finite discrete action space, or leverages the linear structure in the action space and has action-dimension dependency. In contrast, our algorithms' exploration relies more on the learning of the model. Consequently, our sample complexity can be action-dimension-free.

## 2 Problem Setup and Preliminaries

In this section, we first introduce our problem setup for nonlinear bandit and reinforcement learning, and then the preliminary for online learning and sequential Rademacher complexity.

### 2.1 Nonlinear Bandit Problem with Deterministic Reward

We consider *deterministic* nonlinear bandits with continuous actions. Let $\theta \in \Theta$ be the parameter that specifies the bandit instance, $a \in \mathbb{R}^{d_{\mathcal{A}}}$ the action, and $\eta(\theta, a) \in [0, 1]$ the reward function. Let $\theta^\star$ denote the unknown ground-truth parameter. Throughout the paper, we work under the *realizability assumption* that $\theta^\star \in \Theta$. A bandit algorithm aims to maximize $\eta(\theta^\star, a)$. Let $a^\star = \operatorname{argmax}_a \eta(\theta^\star, a)$ be the optimal action (breaking tie arbitrarily). Let $\|H\|_{\mathrm{sp}}$ be the spectral norm of a matrix $H$. We assume that the reward function, its gradient and Hessian matrix are Lipschitz, which are standard in the optimization literature (e.g., Johnson, Zhang [41], Ge et al. [26]).

**Assumption 2.1.** *We assume that for all* $\theta \in \Theta$, $\sup_a \|\nabla_a \eta(\theta, a)\|_2 \leq \zeta_g$ *and* $\sup_a \|\nabla_a^2 \eta(\theta, a)\|_{\mathrm{sp}} \leq \zeta_h$. *For every* $\theta \in \Theta$ *and* $a_1, a_2 \in \mathbb{R}^{d_\mathcal{A}}$, $\|\nabla_a^2 \eta(\theta, a_1) - \nabla_a^2 \eta(\theta, a_2)\|_{\mathrm{sp}} \leq \zeta_{\mathrm{3rd}} \|a_1 - a_2\|_2$.

As a motivation to consider deterministic rewards, we prove in Theorem B.1 for a special case that no algorithm can find a local maximum in less than $\sqrt{d_\mathcal{A}}$ steps. The result implies that an action-dimension-free sample complexity bound is impossible with standard sub-Gaussian noise.[3]

**Approximate local maxima.** In this paper, we aim to find a local maximum of the real reward function $\eta(\theta^\star, \cdot)$. A point $x$ is an $(\epsilon_g, \epsilon_h)$-approximate local maximum of a twice-differentiable function $f(x)$ if $\|\nabla f(x)\|_2 \leq \epsilon_g$, and $\lambda_{max}(\nabla^2 f(x)) \leq \epsilon_h$. As argued in Sec. 1 and proved in Sec. 5, because reaching a global maximum is computational and statistically intractable for nonlinear problems, we only aim to reach a local maximum.

**Sample complexity and local regret for converging to local maxima.** Let $a_t$ be the action that the algorithm takes at time step $t$. The sample complexity for converging to approximate local maxima is defined to be the minimal number of steps $T$ such that there exists $t \in [T]$ where $a_t$ is an $(\epsilon_g, \epsilon_h)$ approximate local maximum with probability at least $1 - \delta$. We also define the "local regret" by comparing with an approximate local maximum. We defer the discussion to Appendix C.5.

## 2.2 Reinforcement Learning

A finite horizon Markov decision process (MDP) with deterministic dynamics is defined by a tuple $\langle T, r, H, \mu_1 \rangle$. Let $\mathcal{S}$ and $\mathcal{A}$ be the state and action spaces. The dynamics $T : \mathcal{S} \times \mathcal{A} \to \mathcal{S}$ gives next state, $r : \mathcal{S} \times \mathcal{A} \to [0, 1]$ is the reward function. $H$ and $\mu_1$ denote the horizon and distribution of initial state respectively. Without loss of generality, we assume that the state space is disjoint for different time steps. That is, there exists disjoint sets $\mathcal{S}_1, \cdots, \mathcal{S}_H$ such that $\mathcal{S} = \cup_{h=1}^H \mathcal{S}_h$, and for any $s_h \in \mathcal{S}, a_h \in a, T(s_h, a_h) \in \mathcal{S}_{h+1}$.

We consider parameterized policy and dynamics. Let $\Pi = \{\pi_\psi : \psi \in \Psi\}$ be the policy class and $\{T_\theta : \theta \in \Theta\}$ the dynamics class. The value function is $V_T^\pi(s_h) \triangleq \mathbb{E}[\sum_{h'=h}^H r(s_{h'}, a_{h'})]$, where $a_h \sim \pi(\cdot \mid s_h), s_{h+1} = T(s_h, a_h)$. Sharing the notation with the bandit setting, let $\eta(\theta, \psi) = \mathbb{E}_{s_1 \sim \mu_1} V_{T_\theta}^{\pi_\psi}(s_1)$ be the expected return of policy $\pi_\psi$ under dynamics $T_\theta$. Let $\rho_T^\pi$ be the distribution of state action pairs when running policy $\pi$ in dynamics $T$. For simplicity, we do not distinguish $\psi, \theta$ from $\pi_\psi, T_\theta$ when the context is clear. For example, we write $V_\theta^\psi = V_{T_\theta}^{\pi_\psi}$.

## 2.3 Preliminary on Online Learning with Stochastic Input Components

Consider a prediction problem where we aim to learn a function that maps from $\mathcal{X}$ to $\mathcal{Y}$ parameterized by parameters in $\Theta$. Let $\ell((x, y); \theta)$ be a loss function that maps $(\mathcal{X} \times \mathcal{Y}) \times \Theta \to \mathbb{R}_+$. An online learner $\mathcal{R}$ aims to solve the prediction tasks under the presence of an adversarial nature iteratively. At time step $t$, the following happens.

1. The learner computes a distribution $p_t = \mathcal{R}(\{(x_i, y_i)\}_{i=1}^{t-1})$ over the parameter space $\Theta$.
2. The adversary selects a point $\bar{x}_t \in \bar{\mathcal{X}}$ (which may depend on $p_t$) and generates a sample $\xi_t$ from some fixed distribution $q$. Let $x_t \triangleq (\bar{x}_t, \xi_t)$, and the adversary picks a label $y_t \in \mathcal{Y}$.
3. The data point $(x_t, y_t)$ is revealed to the online learner.

The online learner aims to minimize the expected regret in $T$ rounds of interactions, defined as

$$\mathrm{REG}_T^{\mathrm{OL}} \triangleq \mathop{\mathbb{E}}_{\substack{\xi_t \sim q, \theta_t \sim p_t \\ \forall 1 \leq t \leq T}} \left[ \sum_{t=1}^T \ell((x_t, y_t); \theta_t) - \inf_{\theta \in \Theta} \sum_{t=1}^T \ell((x_t, y_t); \theta) \right]. \tag{1}$$

The difference of the formulation from the most standard online learning setup is that the $\xi_t$ part of the input is randomized instead of adversarially chosen (and the learner knows the distribution of $\xi_t$ before making the prediction $p_t$). It was introduced by Rakhlin et al. [61], who considered a more generalized setting where the distribution $q$ in round $t$ can depend on $\{x_1, \cdots, x_{t-1}\}$.

---

[3]We rely on deterministic reward to estimate the gradient by finite difference. This method can be extended to stochastic rewards with multiple-point feedback [54].

We adopt the notation from Rakhlin et al. [61, 62] to define the (distribution-dependent) sequential Rademacher complexity of the loss function class $\mathcal{L} = \{(x, y) \mapsto \ell((x, y); \theta) : \theta \in \Theta\}$. For any set $\mathcal{Z}$, a $\mathcal{Z}$-valued tree with length $T$ is a set of functions $\{\boldsymbol{z}_i : \{\pm1\}^{i-1} \to \mathcal{Z}\}_{i=1}^T$. For a sequence of Rademacher random variables $\epsilon = (\epsilon_1, \cdots, \epsilon_T)$ and for every $1 \leq t \leq T$, we denote $\boldsymbol{z}_t(\epsilon) \triangleq \boldsymbol{z}_t(\epsilon_1, \cdots, \epsilon_{t-1})$. For any $\bar{\mathcal{X}}$-valued tree $\boldsymbol{x}$ and any $\mathcal{Y}$-valued tree $\boldsymbol{y}$, we define the sequential Rademacher complexity as

$$\mathfrak{R}_T(\mathcal{L}; \boldsymbol{x}, \boldsymbol{y}) \triangleq \mathbb{E}_{\xi_1, \cdots, \xi_t} \mathbb{E}_\epsilon \left[ \sup_{\ell \in \mathcal{L}} \sum_{t=1}^T \epsilon_t \ell((\boldsymbol{x}(\epsilon), \xi_t), \boldsymbol{y}(\epsilon)) \right]. \tag{2}$$

We also define $\mathfrak{R}_T(\mathcal{L}) = \sup_{\boldsymbol{x},\boldsymbol{y}} \mathfrak{R}_T(\mathcal{L}; \boldsymbol{x}, \boldsymbol{y})$, where the supremum is taken over all $\bar{\mathcal{X}}$-valued and $\mathcal{Y}$-valued trees. Rakhlin et al. [61] proved the existence of an algorithm whose online learning regret satisfies $\text{REG}_T^{\text{OL}} \leq 2\mathfrak{R}_T(\mathcal{L})$.

# 3 Model-based Algorithms for Nonlinear Bandit

We first study model-based algorithms for nonlinear continuous bandits problem, which is a simplification of model-based reinforcement learning. We use the notations and setup in Section 2.1.

**Abstraction of analysis for model-based algorithms.** Typically, a model-based algorithm explicitly maintains an estimated model $\hat{\theta}_t$, and sometimes maintains a distribution, posterior, or confidence region of $\hat{\theta}_t$. We will call $\eta(\theta^\star, a)$ the *real reward* of action $a$, and $\eta(\hat{\theta}_t, a)$ the *virtual reward*. Most analysis for model-based algorithms (including UCB and ours) can be abstracted as showing the following two properties:

(i) the virtual reward $\eta(\hat{\theta}_t, a_t)$ is sufficiently high.
(ii) the virtual reward $\eta(\hat{\theta}_t, a_t)$ is close to the real reward $\eta(\theta^\star, a_t)$ in the long run.

One can expect that a proper combination of property (i) and (ii) leads to showing the real reward $\eta(\theta^\star, a_t)$ is high in the long run. Before describing our algorithms, we start by inspecting and summarizing the pros and cons of UCB from this viewpoint.

**Pros and cons of UCB.** The UCB algorithm chooses an action $a_t$ and an estimated model $\hat{\theta}_t$ that maximize the virtual reward $\eta(\hat{\theta}_t, a_t)$ among those models agreeing with the observed data. The pro is that it satisfies property (i) by definition—$\eta(\hat{\theta}_t, a_t)$ is higher than the optimal real reward $\eta(\theta^\star, a^\star)$. The downside is that ensuring (ii) is challenging and often requires strong complexity measure bound such as Eluder dimension (which is not polynomial for even barely nonlinear models, as shown in Theorem 5.2). The difficulty largely stems from our very limited control of $\hat{\theta}_t$ except its consistency with the observed data. To bound the difference between the real and virtual rewards, we essentially require that *any* model that agrees with the past history should extrapolate to any future data accurately (as quantitatively formulated in Eluder dimension). Moreover, the difficulty of satisfying property (ii) is fundamentally caused by the over-exploration of UCB—As shown in the Theorem 5.3, UCB suffers from bad sample complexity with barely nonlinear family of models.

**Our key idea: natural exploration via model-based curvature estimate.** We deviate from UCB by readjusting the priority of the two desiderata. We prioritize ensuring property (ii) on large model class. We leverage a strong online learning algorithm to predict $\hat{\theta}_t$ with the objective that $\eta(\hat{\theta}_t, a_t)$ matches $\eta(\theta^\star, a_t)$. As a result, the difference between the virtual and real reward depends on the online learnability or the sequential Rademacher complexity of the model class. Sequential Rademacher complexity turns out to be a fundamentally more relaxed complexity measure than Eluder dimension—e.g., two-layer neural networks' sequential Rademacher complexity is polynomial in parameter norm and dimension, but the Eluder dimension is at least exponential in dimension (even with a constant parameter norm). However, an immediate consequence of using online-learned $\hat{\theta}_t$ is that we lose optimism/exploration that ensured property (i).[4]

---

[4]More concretely, the algorithm can get stuck when (1) $a_t$ is optimal for $\hat{\theta}_t$, (2) $\hat{\theta}_t$ fits actions $a_t$ (and history) accurately, but (3) $\hat{\theta}_t$ does not fit $a^\star$ (because online learner never sees $a^\star$). The passivity of online learning formulation causes this issue—the online learner is only required to predict well for the point that it saw and will see, but not for those points that it never observes.

**Algorithm 1** ViOlin: **Vi**rtual Ascent with **Onlin**e Model Learner (for Bandit)

---

1: Set parameter $\kappa_1 = 2\zeta_g$ and $\kappa_2 = 640\sqrt{2}\zeta_h$. Let $\mathcal{H}_0 = \emptyset$; choose $a_0 \in \mathcal{A}$ arbitrarily.
2: **for** $t = 1, 2, \cdots$ **do**
3:      Run online learner $\mathcal{R}$ on $\mathcal{H}_{t-1}$ with loss function $\ell$ (defined in equation (5)) and obtain $p_t = \mathcal{R}(\mathcal{H}_{t-1})$.
4:      Let $a_t \leftarrow \operatorname{argmax}_a \mathbb{E}_{\theta_t \sim p_t}[\eta(\theta_t, a)]$.
5:      Sample $u_t, v_t \sim \mathcal{N}(0, I_{d_{\mathcal{A}} \times d_{\mathcal{A}}})$ independently.
6:      Let $\xi_t = (u_t, v_t)$, $\bar{x}_t = (a_t, a_{t-1})$, and $x_t = (\bar{x}_t, \xi_t)$
7:      Compute $y_t = [\eta(\theta^\star, a_t), \eta(\theta^\star, a_{t-1}), \langle \nabla_a \eta(\theta^\star, a_{t-1}), u_t \rangle, \langle \nabla_a^2 \eta(\theta^\star, a_{t-1}) u_t, v_t \rangle] \in \mathbb{R}^4$
     by applying a finite number of actions in the real environments using equation (3) and (4).
8:      Update $\mathcal{H}_t = \mathcal{H}_{t-1} \cup \{(x_t, y_t)\}$

---

Our approach realizes property (i) in a sense that the virtual reward will improve iteratively if the real reward is not yet near a local maximum. This is much weaker than what UCB offers (i.e., that the virtual reward is higher than the optimal real reward), but suffices to show the convergence to a local maximum of the real reward function. We achieve this by demanding the estimated model $\hat{\theta}_t$ not only to predict the real reward accurately, but also to predict the gradient $\nabla_a \eta(\theta^\star, a)$ and Hessian $\nabla_a^2 \eta(\theta^\star, a)$ accurately. In other words, we augment the loss function for the online learner so that the estimated model satisfies $\eta(\hat{\theta}_t, a_t) \approx \eta(\theta^\star, a_t)$, $\nabla_a \eta(\hat{\theta}_t, a_t) \approx \nabla_a \eta(\theta^\star, a_t)$, and $\nabla_a^2 \eta(\hat{\theta}_t, a_t) \approx \nabla_a^2 \eta(\theta^\star, a_t)$ in the long run. This implies that when $a_t$ is not at a local maximum of the real reward function $\eta(\theta^\star, \cdot)$, then it's not at a maximum of the virtual reward $\eta(\hat{\theta}_t, \cdot)$, and hence the virtual reward will improve in the next round if we take the greedy action that maximizes it.

**Estimating projections of gradients and Hessians.** To guide the online learner to predict $\nabla_a \eta(\theta^\star, a_t)$ correctly, we need a supervision for it. However, we only observe the reward $\eta(\theta^\star, a_t)$. Leveraging the deterministic reward property, we use rewards at $a$ and $a + \alpha_1 u$ to estimate the projection of the gradient at a random direction $u$:

$$\langle \nabla_a \eta(\theta^\star, a), u \rangle = \lim_{\alpha_1 \to 0} (\eta(\theta^\star, a + \alpha_1 u) - \eta(\theta^\star, a)) / \alpha_1 \tag{3}$$

It turns out that the number of random projections $\langle \nabla_a \eta(\theta^\star, a), u \rangle$ needed for ensuring a large virtual gradient *does not* depend on the dimension, because we only use these projections to estimate the norm of the gradient but not necessarily the exact direction of the gradient (which may require $d$ samples.) Similarly, we can also estimate the projection of Hessian to two random directions $u, v \in d_{\mathcal{A}}$ by:

$$\langle \nabla_a^2 \eta(\theta^\star, a) v, u \rangle = \lim_{\alpha_2 \to 0} (\langle \nabla_a \eta(\theta^\star, a + \alpha_2 v), u \rangle - \langle \nabla_a \eta(\theta^\star, a), u \rangle) / \alpha_2 \tag{4}$$
$$= \lim_{\alpha_2 \to 0} \lim_{\alpha_1 \to 0} ((\eta(\theta^\star, a + \alpha_1 u + \alpha_2 v) - \eta(\theta^\star, a + \alpha_2 v)) - (\eta(\theta^\star, a + \alpha_1 u) - \eta(\theta^\star, a))) / (\alpha_1 \alpha_2)$$

Algorithmically, we can choose infinitesimal $\alpha_1$ and $\alpha_2$. Note that $\alpha_1$ should be at least an order smaller than $\alpha_2$ because the limitations are taken sequentially.

We create the following prediction task for an online learner: let $\theta$ be the parameter, $x = (a, a', u, v)$ be the input, $\hat{y} = [\eta(\theta, a), \eta(\theta, a'), \langle \nabla_a \eta(\theta, a'), u \rangle, \langle \nabla_a^2 \eta(\theta, a') u, v \rangle] \in \mathbb{R}^4$ be the output, and $y = [\eta(\theta^\star, a), \eta(\theta^\star, a'), \langle \nabla_a \eta(\theta^\star, a'), u \rangle, \langle \nabla_a^2 \eta(\theta^\star, a') u, v \rangle] \in \mathbb{R}^4$ be the supervision, and the loss function be

$$\ell(((a, a', u, v), y); \theta) \triangleq ([\hat{y}]_1 - [y]_1)^2 + ([\hat{y}]_2 - [y]_2)^2 + \min\left(\kappa_1^2, ([\hat{y}]_3 - [y]_3)^2\right)$$
$$+ \min\left(\kappa_2^2, ([\hat{y}]_4 - [y]_4)^2\right) \tag{5}$$

Here we used $[y]_i$ to denote the $i$-th coordinate of $y \in \mathbb{R}^4$ to avoid confusing with $y_t$ (the supervision at time $t$.) Our algorithm is formally stated in Alg. 1 with its sample complexity bound below.

**Theorem 3.1.** *Suppose the sequential Rademacher complexity of the loss function (defined in Eq. (5)) is bounded by $\sqrt{R(\Theta)T}\operatorname{polylog}(T)$. The sample complexity of Alg. 1 (for finding an $(\epsilon, 6\sqrt{\zeta_{3rd}\epsilon})$-approximate local maximum) is bounded by $\widetilde{\mathcal{O}}(C_1^4 R(\Theta) \max\left(\zeta_h^4 \epsilon^{-8}, \zeta_{3rd}^2 \epsilon^{-6}\right))$.*

We present a proof sketch in Appendix C.1 . Proof of Theorem 3.1 is deferred to Appendix C.4. We can also boost the success probability by running Alg. 1 multiple times. In addition, we can prove that our algorithm enjoys a sublinear local regret. The theorem statement and proof is shown in Appendix C.5 and Appendix C.6 respectively.

### 3.1 Instantiations of Theorem 3.1

In the sequel we sketch some instantiations of our main theorem, whose proofs are deferred to Appendix C.7.

**Linear bandit with finite model class.** Consider a linear bandit problem with action set $\mathcal{A} = \{a \in \mathbb{R}^d : \|a\|_2 \leq 1\}$ and finite model class $\Theta \subset \{\theta \in \mathbb{R}^d : \|\theta\|_2 = 1\}$. Let $\eta(\theta, a) = \langle \theta, a \rangle$ be the reward. We deal with the constrained action set by using a surrogate loss $\tilde{\eta}(\theta, a) \triangleq \langle \theta, a \rangle - \frac{1}{2}\|a\|_2^2$ and apply Theorem C.3 with reward $\tilde{\eta}$. We claim that the sample complexity (of finding an $\epsilon$-suboptimal action for $\eta(\theta^\star, a)$) is bounded by $\widetilde{\mathcal{O}}(\log|\Theta|\epsilon^{-8})$.[5] Note that the bound is *independent* of the dimension $d$. By contrast, the SquareCB algorithm [24] depends polynomially on $d$ (see Theorem 7 of [24]). Zero-order optimization approach [20] in this case also gives a $\mathrm{poly}(d)$ regret bound.

**Linear bandit with sparse or structured model vectors.** We consider the deterministic linear bandit setting where the model class $\Theta = \{\theta \in \mathbb{R}^d : \|\theta\|_0 \leq s, \|\theta\|_2 = 1\}$ consists of all $s$-sparse vectors on the unit sphere. Similarly to finite hypothesis case, we claim that the sample complexity of Alg. 1 is $\widetilde{\mathcal{O}}(\log|\Theta|\epsilon^{-8}\mathrm{polylog}(d))$. The sample complexity of our algorithm only depends on the sparsity level $s$ (up to logarithmic factors), whereas the Eluder dimension of sparse linear hypothesis is still $\Omega(d)$ (see Lemma C.4). Lattimore, Szepesvári [46] show a $\Omega(d)$ sample complexity lower bound for the sparse linear bandit problem with stochastic reward. But here we only consider a deterministic reward and continuous action.

Moreover, we can further extend the result to other linear bandit settings where $\theta$ has an additional structure. Suppose $\Theta = \{\theta = \phi(z) : z \in \mathbb{R}^s\}$ for some Lipschitz function $\phi$. Then, a simlar approach gives sample complexity bound that only depends on $s$ but not $d$ (up to logarithmic factors). Our results can also be extended to non-linear bandits such as deterministic logistic bandits.

**Two-layer neural nets.** We consider the reward function given by two-layer neural networks with width $m$. For matrices $W_1 \in \mathbb{R}^{m \times d}$ and $W_2 \in \mathbb{R}^{1 \times m}$, let $\eta((W_1, W_2), a) = W_2\sigma(W_1 a) - \frac{1}{2}\|a\|_2^2$ for some nonlinear link function $\sigma : \mathbb{R} \to [0,1]$ with bounded derivatives up to the third order. Recall that the $(1, \infty)$-norm of $W_1^\top$ is defined by $\max_{i \in [m]} \sum_{j=1}^d |[W_1]_{i,j}|$. That is, the max 1-norm of the rows of $W_1$. Let the model hypothesis space be $\Theta = \{(W_1, W_2) : \|W_1^\top\|_{1,\infty} \leq 1, \|W_2\|_1 \leq 1\}$ and $\theta \triangleq (W_1, W_2)$. We claim that Alg .1 finds an $(\epsilon, 6\sqrt{\zeta_{3\mathrm{rd}}\epsilon})$-approximate local maximum in $\widetilde{\mathcal{O}}(\epsilon^{-8}\mathrm{polylog}(d))$ steps. To the best of our knowledge, this is the first result analyzing nonlinear bandit with neural network parameterization. The result follows from analyzing the sequential Rademacher complexity for $\eta$, $\langle \nabla_a \eta, u \rangle$, and $\langle u, \nabla_a^2 \eta \cdot v \rangle$, and finally the resulting loss function $\ell$. See Theorem C.6 in Section C.7 for details. We remark here that zero-order optimization in this case has a $\mathrm{poly}(d)$ sample complexity. In addition, if the second layer of the neural network $W_2$ contains all negative entries, and the activation function $\sigma$ is monotone and convex, then $\eta((W_1, W_2), a)$ is *concave* in the action. (This is a special case of input convex neural networks [5].) In this case, Alg. 1 finds an $\epsilon$-suboptimal action (see Theorem C.6).

## 4 Model-based Reinforcement Learning

In this section, we extend the results in Section 3 to model-based reinforcement learning with deterministic dynamics and reward function.

We can always view a model-based reinforcement learning problem with parameterized dynamics and policy as a nonlinear bandit problem in the following way. The policy parameter $\psi$ corresponds to the action $a$ in bandit, and the dynamics parameter $\theta$ corresponds to the model parameter $\theta$ in

---

[5]Recall that in bandit literature, action $a$ is an $\epsilon$-approximate optimal action if $\eta(\theta^\star, a) \geq \eta(\theta^\star, a^\star) - \epsilon$.

bandit. The expected total return $\eta(\theta, \psi) = \mathbb{E}_{s_1 \sim \mu_1} V_{T_\theta}^{\pi_\psi}(s_1)$ is the analogue of reward function in bandit. We intend to make the same regularity assumptions on $\eta$ as in the bandit case (that is, Assumption 2.1) with $a$ being replaced by $\psi$. However, when the policy is deterministic, the reward function $\eta$ has Lipschitz constant with respect to $\psi$ that is exponential in $H$ (even if dynamics and policy are both deterministic with good Lipschitzness). This prohibits efficient optimization over policy parameters. Therefore we focus on *stochastic policies* in this section, for which we expect $\eta$ and its derivatives to be Lipschitz with respect to $\psi$.

Blindly treating RL as a bandit only utilizes the reward but not the state observations. In fact, one major reason why model-based methods are more sample efficient is that it supervises the learning of dynamics by state observations. To reason about the learning about local steps and the dynamics, we make the following additional Lipschitzness of value functions w.r.t to the states and Lipschitzness of policies w.r.t to its parameters, beyond those assumptions for the total reward $\eta(\theta, \psi)$ in Assumption 2.1.

**Assumption 4.1.** *We assume the following (analogous to Assumption 2.1) on the value function:* $\forall \psi \in \Psi, \theta \in \Theta, s, s' \in \mathcal{S}$ *we have* $|V_\theta^\psi(s) - V_\theta^\psi(s')| \leq L_0 \|s - s'\|_2$; $\|\nabla_\psi V_\theta^\psi(s) - \nabla_\psi V_\theta^\psi(s')\|_2 \leq L_1 \|s - s'\|_2$; $\|\nabla_\psi^2 V_\theta^\psi(s) - \nabla_\psi^2 V_\theta^\psi(s')\|_{\mathrm{sp}} \leq L_2 \|s - s'\|_2$.

**Assumption 4.2.** *We assume the following Lipschitzness assumptions on the stochastic policies parameterization* $\pi_\psi$.[6] $\|\mathbb{E}_{a \sim \pi_\psi(\cdot|s)}[(\nabla_\psi \log \pi_\psi(a \mid s))(\nabla_\psi \log \pi_\psi(a \mid s))^\top]\|_{\mathrm{sp}} \leq \chi_g$; $\|\mathbb{E}_{a \sim \pi_\psi(\cdot|s)}[(\nabla_\psi \log \pi_\psi(a \mid s))^{\otimes 4}]\|_{\mathrm{sp}} \leq \chi_f$; $\|\mathbb{E}_{a \sim \pi_\psi(\cdot|s)}[(\nabla_\psi^2 \log \pi_\psi(a \mid s))(\nabla_\psi^2 \log \pi_\psi(a \mid s))^\top]\|_{\mathrm{sp}} \leq \chi_h$.

Our results depends polynomially on the parameters $L_0, L_1, L_2, \chi_g, \chi_f$ and $\chi_h$. To demonstrate that the Assumption 4.1 and 4.2 contain interesting RL problems with nonlinear models and stochastic policies, we give the following example where these parameters are all on the order of $O(1)$.

**Example 4.3.** *Let state space $\mathcal{S}$ be the unit ball in $\mathbb{R}^d$ and action space $\mathcal{A}$ be $\mathbb{R}^d$. The (deterministic) dynamics $T$ is given by $T(s, a) = \mathrm{N}_\theta(s + a)$, where $\mathrm{N}$ is a nonlinear model parameterized by $\theta$, e.g., a neural network. We assume that $\theta$ belongs to a finite hypothesis class $\Theta$ that satisfies $\|\mathrm{N}_\theta(s + a)\|_2 \leq 1$ for all $\theta \in \Theta, s \in \mathcal{S}, a \in \mathcal{A}$. Assume that the reward function $r(s, a)$ is $L_r$-Lipschitz w.r.t $\ell_2$-norm, that is, satisfying $|r(s_1, a_1) - r(s_2, a_2)| \leq L_r(\|s_1 - s_2\|_2 + \|a_1 - a_2\|_2)$. We consider a family of stochastic Gaussian policies with the mean being linear in the state: $\pi_\psi(s) = \mathcal{N}(\psi s, \sigma^2 I)$, parameterized by $\psi \in \mathbb{R}^{d \times d}$ with $\|\psi\|_{\mathrm{op}} \leq 1$. We consider $\sigma \in (0, 1)$ as a small constant on the order of 1.*

*In this setting, Assumption 2.1, 4.1, and 4.2 hold with all parameters $\zeta_g, \zeta_h, \zeta_{3\mathrm{rd}}, L_0, L_1, L_2, \chi_g, \chi_f$ and $\chi_h$ bounded by $\mathrm{poly}(\sigma, 1/\sigma, H, L_r)$.*

Bounding the Lipschitz parameters is highly nontrivial and deferred to Appendix E.

We show that the difference of gradient and Hessian of the total reward can be upper-bounded by the difference of dynamics. Let $\tau_t = (s_1, a_1, \cdots, s_H, a_H)$ be a trajectory sampled from policy $\pi_{\psi_t}$ under the real dynamics $T_{\theta^\star}$. Similarly to [79], when the value function is Lipschitz, we upper bound $\Delta_{t,1} = |\eta(\theta_t, \psi_t) - \eta(\theta^\star, \psi_t)|$ by the one-step model prediction errors. Similarly, we can upper bound the gradient and Hessian errors by model prediction errors. As a result, the loss function simply can be set to

$$\ell((\tau_t, \tau_t'); \theta) = \sum_{(s_h, a_h) \in \tau_t} \|T_\theta(s_h, a_h) - T_{\theta^\star}(s_h, a_h)\|_2^2 + \sum_{(s_h', a_h') \in \tau_t'} \|T_\theta(s_h', a_h') - T_{\theta^\star}(s_h', a_h')\|_2^2 \quad (6)$$

for two trajectories $\tau, \tau'$ sampled from policy $\pi_{\psi_t}$ and $\pi_{\psi_{t-1}}$ respectively. Compared to Alg. 1, the loss function is here simpler without relying on finite difference techniques to query gradients projections. Our algorithm for RL is analogous to Alg. 1 by using the loss function in Eq. (6). Our algorithm is presented in Alg. 2 in Appendix D. Main theorem for Alg. 2 is shown below.

**Theorem 4.4.** *Let $c_1 = HL_0^2(4H^2\chi_h + 4H^4\chi_f + 2H^2\chi_g + 1) + HL_1^2(8H^2\chi_g + 2) + 4HL_2^2$ and $C_1 = 2 + \frac{\zeta_g}{\zeta_h}$. Suppose the sequential Rademacher complexity of the loss function (defined*

---

[6]Recall that the injective norm of a $k$-th order tensor $A \in \mathbb{R}^{d^{\otimes k}}$ is defined as $\|A^{\otimes k}\|_{\mathrm{sp}} = \sup_{u \in S^{d-1}} \langle A, u^{\otimes k} \rangle$.

*in Eq.* ([6](#)*)) is bounded by* $\sqrt{R(\Theta)T}\,\mathrm{polylog}(T)$. *The sample complexity of Alg.* [2](#) *(for finding an* $(\epsilon, 6\sqrt{\zeta_{3\mathrm{rd}}\epsilon})$-*approximate local maximum) is bounded by* $\widetilde{\mathcal{O}}(c_1^2 C_1^4 R(\Theta) \max(\zeta_h^4 \epsilon^{-8}, \zeta_{3\mathrm{rd}}^2 \epsilon^{-6}))$.

**Instantiation of Theorem [4.4](#) on Example [4.3](#).**  Applying Theorem [4.4](#) to the Example [4.3](#) with $\sigma = \Theta(1)$ we get the sample complexity upper bound $\widetilde{\mathcal{O}}(\mathrm{poly}(\sigma, 1/\sigma, H, L_r) \log|\Theta|\epsilon^{-8})$.

**Comparison with policy gradient.**  To the best of our knowledge, the best analysis for policy gradient [74] shows convergence to a local maximum with a sample complexity that depends polynomially on $\|\nabla_\psi \log \pi_\psi(a \mid s)\|_2$ [4]. For the instance in in Example [4.3](#), this translates to a sample complexity guarantee on the order of $\sqrt{d}/\sigma$. In contrast, our sample complexity is independent of the dimension $d$. Instead, our bound depends on the complexity of the model family $\Theta$ which could be much smaller than the ambient dimension—this demonstrates that we leverage the model extrapolation.

# 5   Lower Bounds

We prove several lower bounds to show (a) the hardness of finding global maxima, and (b) the inefficiency of using optimism in nonlinear bandit.

**Hardness of Global Optimality.**  In the following theorem, we show it statistically intractable to find the global optimal policy when the function class is chosen to be the neural networks with ReLU activation. That is, the reward function can be written in the form of $\eta((w, b), a) = \mathrm{ReLU}(\langle w, a \rangle + b)$. Note that the reward function can also be made smooth by replacing the activation by a smoothed version. For example, $\eta((w, b), a) = \mathrm{ReLU}(\langle w, a \rangle + b)^2$.

**Theorem 5.1.** *When the function class is chosen to be one-layer neural networks with ReLU activation, the minimax sample complexity is* $\Omega(\varepsilon^{-(d-2)})$.

We can also prove that the eluder dimension of the constructed reward function class is exponential.

**Theorem 5.2.** *The* $\varepsilon$-*eluder dimension of one-layer neural networks is at least* $\Omega(\varepsilon^{-(d-1)})$.

This result is concurrently established by Li et al. [50, Theorem 8]. The proofs of both theorems are deferred to Appendices [B.1](#) and [B.2](#), respectively. We also note that Theorem [5.1](#) does require ReLU activation, because if the ReLU function is replaced by a *strictly* monotone link function with bounded derivatives (up to third order), then this is the setting of deterministic generalized linear bandit problem, which does allow a global regret that depends polynomially on dimension [23, 14, 52]. In this case, our Theorem [C.3](#) can also give polynomial global regret result: because all local maxima of the reward function is global maximum [34, 43] and it also satisfies the strict-saddle property [26], the local regret result translates to a global regret result. This shows that our framework does separate the intractable cases from the tractable by the notions of local and global regrets.

With two-layer neural networks, we can relax the use of ReLU activation—Theorem [5.2](#) holds with two-layer neural networks and leaky-ReLU activations [75] because $O(1)$ leaky-ReLU can implement a ReLU activation. We conjecture that with more layers, the impossibility result also holds for a broader sets of activations.

**Inefficiency caused by optimism in nonlinear models.**  The next theorem states that the UCB algorithm that uses optimism-in-face-of-uncertainty principle can overly explore in the action space, even if the ground-truth is simple. In fact the UCB algorithm will keep exploring for an exponential number of steps. So, UCB doesn't even converge to local max, indicating more sophisticated algorithms like ours are necessary. Proof of the theorem is deferred to Appendix [B.3](#).

**Theorem 5.3.** *Consider the case where the ground-truth reward function is linear:* $\langle \theta^\star, a \rangle$ *and the action set is* $a \in S^{d-1}$. *If the hypothesis is chosen to be two-layer neural network with width $d$, UCB algorithm with tightest upper confidence bound suffers exponential sample complexity .*

# 6 Additional Related Work

There are several provable efficient algorithms without optimism for contextual bandits. Most of the results achieve a action-dimension dependent regret bound for contextual bandits [21, 1, 76, 45]. Foster, Rakhlin [24] and Simchi-Levi, Xu [67] exploit the exploration probability inversely depends on the empirical gap. The SquareCB algorithm [24] also extends to infinite actions with linear structure, but the regret depends on the action dimension. The exploration strategy in Foster, Rakhlin [24] and Simchi-Levi, Xu [67] relies on the structure in the action space whereas ours exploits the extrapolation in the model space. Recently, Foster et al. [25] prove an instance-dependent regret bound for contextual bandit. Zhou et al. [81] also consider non-linear contextual bandits and propose the NeuralUCB algorithm by leveraging the NTK approach. It converges to a globally optimal solution with number of samples polynomial in the number of contexts and the number of actions. For nonlinear bandit problems with continuous actions, the sample complexity depends on the "effective dimension" of some kernel matrix. Because no algorithm can find the global maximum for the hard instances in Theorem 5.1 with polynomial samples, the effective dimension should be exponential in dimension for the hard instances.

Deterministic nonlinear bandits can also be formulated as zero-order optimization without noise (see Duchi et al. [20], Liu et al. [54] and references therein), where the reward is assumed to be any 1-Lipschitz function. In contrast, our algorithm exploits the knowledge of the reward function parameterization and achieves an action-dimension-free sample complexity. For stochastic nonlinear bandit, Filippi et al. [23] consider generalized linear model. Valko et al. [70], Zhou et al. [81] focus on rewards in a RKHS or NTK. Yang et al. [78] extends this approach to reinforcement learning.

Another line of research focuses on solving reinforcement learning by optimization on the policy space. Agarwal et al. [4] prove that natural policy gradient can solve tabular MDPs efficiently. Cai et al. [8] incorporate exploration bonus in proximal policy optimization algorithm and achieves polynomial regret in linear MDP setting.

Beyond linear function approximations, there are also extensive studies on various settings that allow efficient algorithms. For example, rich observation MDPs [44, 11, 18, 57], state aggregation [16, 51], Bellman rank [38, 13] and others [17, 53, 59, 80, 42].

# 7 Conclusion

In this paper, we design new algorithms whose local regrets are bounded by the sequential Rademacher complexity of particular loss functions. By rearranging the priorities of exploration versus exploitation, our algorithms avoid over-aggressive explorations caused by the optimism in the face of uncertainty principle, and hence apply to nonlinear models and dynamics.

We raise the following questions as future works:

1. Since we mainly focus on proving a regret bound that depends only on the complexity of dynamics/reward class, our convergence rate in $T$ is likely not minimax optimal. Can our algorithms (or analysis) be modified to achieve minimax optimal regret for some of the instantiations such as sparse linear bandit and linear bandit with finite model class?

2. In the bandit setting, we focus on deterministic reward because our ViOlin algorithm relies on finite difference to estimate the gradient and Hessian of reward function. In fact, Theorem B.1 shows that action-dimension-free regret bound for linear models is impossible under standard Gaussian noise. Can we extend our algorithm to stochastic environments with additional assumptions on noises?

3. In the reinforcement learning setting, we use policy gradient lemma to upper bound the gradient/Hessian loss by the dynamics loss, which inevitable require the policies being stochastic. Despite the success of stochastic policies in deep reinforcement learning, the optimal policy may not be stochastic. Can we extend the ViOlin algorithm to reinforcement learning problems with deterministic policy hypothesis?

**Acknowledgment**

The authors would like to thank Yuanhao Wang, Daogao Liu, Zhizhou Ren, Jason D. Lee, Colin Wei, Akshay Krishnamurthy, Alekh Agarwal and Csaba Szepesvári for helpful discussions. TM is also partially supported by the Google Faculty Award, Lam Research, Toyota research, JD.com, NSF IIS 2045685, the Sloan Fellowship.

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
