# List of Appendices

# A  Limitations of Our Algorithms

In this section we discuss the limitations of our algorithms.

**Deterministic reward.**    In the nonlinear bandit setting, we rely on deterministic reward to estimate the gradient by finite difference. However, similar to zero-order optimization literature, this method can be extended to stochastic rewards with multiple-point feedback [54]. We also prove that under standard sub-Gaussian noise, no algorithm can achieve action-dimension-free sample complexity (see Theorem B.1).

**Sub-optimal dependency on $\epsilon$.**    As suggested in Section 3.1, the sample complexity of our algorithm is not minimax-optimal regarding the dependence of $\epsilon$. This is because we solve concrete instances by loose reductions. In addition, we didn't try to optimize the dependence on $\epsilon$. Achieving optimal sample complexity is left as future work.

**How large is sequential Rademacher complexity?**    Bounding the sequential Rademacher complexity for the loss class can be non-trivial. However, we generally believe that both the direction gradient term and the projected Hessian term in the loss have similar complexities as the original neural nets, and therefore our loss function is not much more complex than the reward function itself. There is some evidence that corroborates this conjecture.

- In the neural net example in Section 3.1, the $(1, \infty)$-norm bound for the sequential Rademacher complexity of the loss class is the same as the best known bound the neural net family Rakhlin et al. [62, Proposition 15].
- If a neural net has $p$ parameters, its directional gradient and hessian can both be expressed by neural nets with $\mathcal{O}(p)$ parameters.
- For finite model class, the sequential Rademacher complexity for the loss function classes are also bounded by log size of the hypothesis.

# B  Missing Proofs in Section 5

In this section, we prove several negative results.

## B.1  Proof and Remarks of Theorem 5.1

*Proof.* We consider the class $\mathcal{I} = \{I_{\theta,\varepsilon} : \|\theta\|_2 \leq 1, \varepsilon > 0\}$ of infinite-armed bandit instances, where in the instance $I_{\theta,\varepsilon}$, the reward of pulling action $x \in \mathbb{B}_2^d(1)$ is deterministic and is equal to

$$\eta(I_{\theta,\varepsilon}, x) = Ax\{\langle x, \theta \rangle - 1 + \varepsilon, 0\}. \tag{7}$$

We prove the theorem by proving the minimax regret. The sample complexity then follows from the canonical sample complexity-regret reduction [39, Section 3.1]. Let $\mathcal{A}$ denote any algorithm. Let $R_{\mathcal{A},I}^T$ be the $T$-step regret of algorithm $\mathcal{A}$ under instance $I$. Then we have

$$\inf_{\mathcal{A}} \sup_{I \in \mathcal{I}} \mathbb{E}[R_{\mathcal{A},I}^T] \geq \Omega(T^{\frac{d-2}{d-1}}).$$

Fix $\varepsilon = c \cdot T^{-1/(d-1)}$. Let $\Theta$ be an $\varepsilon$-packing of the sphere $\{x \in \mathbb{R}^d : \|x\|_2 = 1\}$. Then we have $|\Theta| \geq \Omega(\varepsilon^{-(d-1)})$. So we choose $c > 0$ to be a numeric constant such that $T \leq |\Theta|/2$. Let $\mu$ be the distribution over $\Theta$ such that $\mu(\theta) = \Pr[\exists t \leq T \text{ s.t. } \eta(I_{\theta,\varepsilon}, \boldsymbol{a}_t) \neq 0 \text{ when } r_\tau \equiv 0 \text{ for } \tau = 1, \ldots, T]$. Note that for any action $\boldsymbol{a}_t \in \mathbb{B}_2^d(1)$, there is at most one $\theta \in \Theta$ such that $\eta(I_{\theta,\varepsilon}, \boldsymbol{a}_t) \neq 0$, because $\Theta$ is a packing. Since $T \leq |\Theta|/2$, there exists $\theta^* \in \Theta$ such that $\mu(\theta^*) \leq 1/2$. Therefore, with probability $1/2$, the algorithm $\mathcal{A}$ would obtain reward $r_t = \eta(I_{\theta^*,\varepsilon}, \boldsymbol{a}_t) = 0$ for every time step $t = 1, \ldots, T$. Note that under instance $I_{\theta^*,\varepsilon}$, the optimal action is to choose $\boldsymbol{a}_t \equiv \theta^*$, which would give reward $r_t^* \equiv \varepsilon$. Therefore, with probability $1/2$, we have $\mathbb{E}[R_{\mathcal{A},I_{\theta,\varepsilon}}^T] \geq \varepsilon T/2 \geq \Omega(T^{\frac{d-2}{d-1}})$.  □

We also note that Theorem 5.1 does require ReLU activation, because if the ReLU function is replaced by a *strictly* monotone link function with bounded derivatives (up to third order), it is the

setting of deterministic generalized linear bandit problem, which does allow a global regret that depends polynomially on dimension [23, 14, 52]. In this case, our Theorem C.3 can also give polynomial global regret result: because all local maxima of the reward function is global maximum [34, 43] and it also satisfies the strict-saddle property [26], all approximate local maxima are global. This shows that our framework does separate the intractable cases from the tractable by the notions of local and global regrets.

With two-layer neural networks, we can relax the use of ReLU activation—Theorem 5.2 holds with two-layer neural networks and leaky-ReLU activations [75] because $O(1)$ leaky-ReLU can implement a ReLU activation. We conjecture that with more layers, the impossibility result also holds for a broader sets of activations.

## B.2 Proof of Theorem 5.2

*Proof.* We adopt the notations from Appendix B.1. We use $\dim_E(\mathcal{F}, \varepsilon)$ to denote the $\varepsilon$-eluder dimension of the function class $\mathcal{F}$. Let $\Theta$ be an $\varepsilon$-packing of the sphere $\{x \in \mathbb{R}^d : \|x\|_2 = 1\}$. We write $\Theta = \{\theta_1, \ldots, \theta_n\}$. Then we have $n \geq \Omega(\varepsilon^{-(d-1)})$. Next we establish that $\dim_E(\mathcal{F}, \varepsilon) \geq \Omega(\varepsilon^{-(d-1)})$. For each $i \in [n]$, we define the function $f_i(\boldsymbol{a}) = \eta(I_{\theta_i, \varepsilon}, \boldsymbol{a}) \in \mathcal{F}$. Then for $i \leq n-1$, we have $f_i(\theta_j) = f_{i+1}(\theta_j)$ for $j \leq i-1$, while $\varepsilon = f_i(\theta_i) \neq f_{i+1}(\theta_i) = 0$. Therefore, $\theta_i$ is $\frac{\varepsilon}{2}$-independent of its predecessors. As a result, we have $\dim_E(\mathcal{F}, \varepsilon) \geq n-1$. $\qquad\square$

## B.3 Proof of Theorem 5.3

First of all, we review the UCB algorithm in deterministic environments.

We formalize UCB algorithm under deterministic environments as follows. At every time step $t$, the algorithm maintains a upper confidence bound $C_t : \mathcal{A} \to \mathbb{R}$. The function $C_t$ satisfies $\eta(\theta^\star, a) \leq C_t(a)$. And then the action for time step $t$ is $a_t \leftarrow \arg\max C_t(a)$. Let $\Theta_t$ be the set of parameters that is consistent with $\eta(\theta^\star, a_1), \cdots, \eta(\theta^\star, a_{t-1})$. That is, $\Theta_t = \{\theta \in \Theta : \eta(\theta, a_\tau) = \eta(\theta^\star, a_\tau), \forall \tau < t\}$. In a deterministic environment, the tightest upper confidence bound is $C_t(a) = \sup_{\theta \in \Theta_t} \eta(\theta, a)$.

We first provide a proof sketch to the theorem. We consider the following reward function.

$$\eta((\theta_1^\star, \theta_2^\star, \alpha), a) = \frac{1}{64} \langle a, \theta_1^\star \rangle + \alpha \max\left(\langle \theta_2^\star, a \rangle - \frac{31}{32}, 0\right).$$

Note that the reward function $\eta$ can be clearly realized by a two-layer neural network with width $2d$. When $\alpha = 0$ we have $\eta((\theta_1^\star, \theta_2^\star, \alpha), a) = \frac{1}{64} \langle \theta_1^\star, a \rangle$, which represents a linear reward. Informally, optimism based algorithm will try to make the second term large (because optimistically the algorithm hopes $\alpha = 1$), which leads to an action $a_t$ that is suboptimal for ground-truth reward (in which case $\alpha = 0$). In round $t$, the optimism algorithm observes $\langle \theta_2^\star, a_t \rangle = 0$, and can only eliminate an exponentially small fraction of $\theta_2^\star$ from the hypothesis. Therefore the optimism algorithm needs exponential number of steps to determine $\alpha = 0$ and stops exploration. Formally, the prove is given below.

*Proof.* Consider a bandit problem where $\mathcal{A} = S^{d-1}$ and

$$\eta((\theta_1^\star, \theta_2^\star, \alpha), a) = \frac{1}{64} \langle a, \theta_1^\star \rangle + \alpha \max\left(\langle \theta_2^\star, a \rangle - \frac{31}{32}, 0\right).$$

The hypothesis space is $\Theta = \{\theta_1, \theta_2, \alpha : \|\theta_1\|_2 \leq 1, \|\theta_2\|_2 \leq 1, \alpha \in [0, 1]\}$. Then the reward function $\eta$ can be clearly realized by a two-layer neural network with width $d$. Note that when $\alpha = 0$ we have $\eta((\theta_1^\star, \theta_2^\star, \alpha), a) = \frac{1}{64} \langle \theta_1^\star, a \rangle$, which represents a linear reward. In the following we use $\boldsymbol{\theta}^\star = (\theta_1^\star, \theta_2^\star, 0)$ as a shorthand.

The UCB algorithm is described as follows. At every time step $t$, the algorithm maintains a upper confidence bound $C_t : \mathcal{A} \to \mathbb{R}$. The function $C_t$ satisfies $\eta(\boldsymbol{\theta}^\star, a) \leq C_t(a)$. And then the action for time step $t$ is $a_t \leftarrow \text{argmax}\, C_t(a)$.

Let $\mathcal{P} = \{p_1, p_2, \cdots, p_n\}$ be an $\frac{1}{2}$-packing of the sphere $S^{d-1}$, where $n = \Omega(2^d)$. Let $\mathcal{B}(p_i, \frac{1}{4})$ be the ball with radius $1/4$ centered at $p_i$, and $B_i = \mathcal{B}(p_i, \frac{1}{4}) \cup S^{d-1}$. We prove the theorem by

showing that the UCB algorithm will explore every packing in $\mathcal{P}$. That is, for any $i \in [n]$, there exists $t$ such that $a_t \in B_i$. Since we have $\sup_{a_j \in B_j} \langle p_i, a_j \rangle \leq 31/32$ for all $j \neq i$, this over-exploration strategy leads to a sample complexity (for finding a $(31/2048)$-suboptimal action) at least $\Omega(2^d)$ when $\boldsymbol{\theta}^\star = (p_i, p_i, 0)$.

Let $\Theta_t$ be the set of parameters that is consistent with $\eta(\boldsymbol{\theta}^\star, a_1), \cdots, \eta(\boldsymbol{\theta}^\star, a_{t-1})$. That is, $\Theta_t = \{\boldsymbol{\theta} \in \Theta : \eta(\boldsymbol{\theta}, a_\tau) = \eta(\boldsymbol{\theta}^\star, a_\tau), \forall \tau < t\}$. Since our environment is deterministic, a tightest upper confidence bound is $C_t(a) = \sup_{\boldsymbol{\theta} \in \Theta_t} \eta(\boldsymbol{\theta}, a)$. Let $A_t = \{a_1, \cdots, a_t\}$. It can be verified that for any $\theta_2 \in S^{d-1}$, $\eta((\theta_1^\star, \theta_2, 1), \cdot)$ is consistent with $\eta(\boldsymbol{\theta}^\star, \cdot)$ on $A_{t-1}$ if $\mathcal{B}(\theta_2, \frac{1}{4}) \cup A_{t-1} = \emptyset$. As a result, for any $\theta_2$ such that $\mathcal{B}(\theta_2, \frac{1}{4}) \cup A_{t-1} = \emptyset$ we have

$$C_t(\theta_2) \geq \frac{1}{32} > \frac{1}{128} + \sup_a \eta(\boldsymbol{\theta}^\star, a). \tag{8}$$

Next we prove that for any $i \in [n]$, there exists $t$ such that $a_t \in \mathcal{B}(p_i, \frac{1}{4})$. Note that $\eta(\boldsymbol{\theta}, \cdot)$ is $\frac{65}{64}$ Lipschitz for every $\boldsymbol{\theta} \in \Theta$. As a result, $C_t(a_\tau + \xi) \leq C_t(a_\tau) + \frac{65}{64} \|\xi\|_2 = \eta(\boldsymbol{\theta}^\star, a_\tau) + \frac{65}{64} \|\xi\|_2$ for all $\tau < t$. Consequently,

$$C_t(a_\tau + \xi) \leq \sup_a \eta(\boldsymbol{\theta}^\star, a) + \frac{1}{128} = \frac{3}{128} \tag{9}$$

for any $\tau < t$ and $\xi$ such that $\|\xi\|_2 \leq \frac{1}{130}$. In other words, Eq. (9) upper bounds the upper confidence bound for actions that is taken by the algorithm, and Eq. (8) lower bounds the upper confidence bound for actions that is not taken.

Now, for the sake of contradiction, assume that actions in $\mathcal{B}(\theta_2, \frac{1}{4})$ is never taken by the algorithm. By Eq. (9) we have $C_t(\theta_2) \geq \frac{1}{32}$ for all $t$. Let $\mathcal{H}_t = \cup_{\tau=1}^{t-1} \mathcal{B}(a_\tau, \frac{1}{130})$. By Eq. (9) we have $C_t(a) \leq \frac{3}{128}$ for all $a \in \mathcal{H}_t$. Because $a_t \leftarrow \operatorname{argmax}_a C_t(a)$ and $\max_{a \in \mathcal{H}_t} C_t(a) < C_t(\theta_2)$, we conclude that $a_t \notin \mathcal{H}_t$. Therefore, $\{a_t\}$ is a $(1/130)$-packing. However, the $(1/130)$-packing of $S^{d-1}$ has a size bounded by $130^d$, which leads to contradiction.

For any $\theta_2$ there exists $t \leq 130^d$ such that $a_t \in \mathcal{B}(\theta_2, \frac{1}{4})$. $\qquad \square$

### B.4 Hardness of stochastic environments

As a motivation to consider deterministic rewards, the next theorem proves that a $\operatorname{poly}(\log|\Theta|)$ sample complexity is impossible for finding local optimal action even under mild stochastic environment.

**Theorem B.1.** *There exists an bandit problem with stochastic reward and hypothesis class with size* $\log|\Theta| = \widetilde{\mathcal{O}}(1)$, *such that any algorithm requires* $\Omega(d)$ *sample to find a* $(0.1, 1)$-*approximate second order stationary point with probability at least* $3/4$.

A similar theorem is proved in Lattimore, Szepesvári [46, Section 23.3] (in a somewhat different context) with minor differences in the constructed hard instances.

*Proof.* We consider a linear bandit problem with hypothesis class $\Theta = \{e_1, \cdots, e_d\}$. The action space is $S^{d-1}$. The stochastic reward function is given by $\eta(\theta, a) = \langle \theta, a \rangle + \xi$ where $\xi = \mathcal{N}(0, 1)$ is the noise. Define the set $A_i = \{a \in S^{d-1} : |\langle a, e_i \rangle| \geq 0.9\}$. By basic algebra we get, $A_i \cap A_j = \emptyset$ for all $i \neq j$.

The manifold gradient of $\eta(\theta, \cdot)$ on $S^{d-1}$ is

$$\operatorname{grad} \eta(\theta, a) = \left(I - aa^\top\right)\theta.$$

By triangular inequality we get $\|\operatorname{grad} \eta(\theta, a)\|_2 \geq \|\theta\|_2 - |\langle a, \theta \rangle|$. Consequently, $\|\operatorname{grad} \eta(\theta_i, a)\|_2 \geq 0.1$ for $a \notin A_i$. In other words, $\left(S^{d-1} \setminus A_i\right)$ does not contain any $(0.1, 1)$-approximate second order stationary point for $\eta(\theta_i, \cdot)$.

For a fixed algorithm, let $a_1, \cdots, a_T$ be the sequence of actions chosen by the algorithm, and $x_t = \langle \theta^\star, a_t \rangle + \xi_t$. Next we prove that with $T \lesssim d$ steps, there exists $i \in [d]$ such that $\Pr_i[a_T \in A_i] \leq 1/2$, where $\Pr_i$ denotes the probability space generated by $\theta^\star = \theta_i$. Let $\Pr_0$ be the probability space

generated by $\theta^\star = 0$. Let $E_{i,T}$ be the event that the algorithm outputs an action $a \in A_i$ at time step $T$. By Pinsker inequality we get,

$$\mathbb{E}_i[E_{i,T}] \leq \mathbb{E}_0[E_{i,T}] + \sqrt{\frac{1}{2}D_{\mathrm{KL}}(\mathrm{Pr}_i, \mathrm{Pr}_0)}. \tag{10}$$

Using the chain rule of KL-divergence and the fact that $D_{\mathrm{KL}}(\mathcal{N}(0,1), \mathcal{N}(a,1)) = \frac{a^2}{2}$, we get

$$\mathbb{E}_i[E_{i,T}] \leq \mathbb{E}_0[E_{i,T}] + \sqrt{\frac{1}{4}\mathbb{E}_0\left[\sum_{t=1}^T \langle a_t, \theta_i \rangle^2\right]}. \tag{11}$$

Consequently,

$$\sum_{i=1}^d \mathbb{E}_i[E_{i,T}] \leq \sum_{i=1}^d \mathbb{E}_0[E_{i,T}] + \sum_{i=1}^d \sqrt{\frac{1}{4}\mathbb{E}_0\left[\sum_{t=1}^T \langle a_t, \theta_i \rangle^2\right]} \tag{12}$$

$$\leq 1 + \sqrt{\frac{d}{4}\mathbb{E}_0\left[\sum_{i=1}^d \sum_{t=1}^T \langle a_t, \theta_i \rangle^2\right]} \leq 1 + \sqrt{\frac{dT}{4}}, \tag{13}$$

which means that

$$\min_{i \in [d]} \mathbb{E}_i[E_{i,T}] \leq \frac{1}{d} + \sqrt{\frac{T}{4d}}. \tag{14}$$

Therefore when $T \leq d$, there exists $i \in [d]$ such that $\mathbb{E}_i[E_{i,T}] \leq \frac{3}{4}$. $\qquad\square$

## C   Missing Proofs in Section 3

In this section, we show missing proofs in Section 3. We also define the notion of local regret, and prove a sublinear (local) regret result.

### C.1   Proof Sketch for Theorem 3.1

Proof of Theorem 3.1 consists of the following parts:

i. Because of the design of the loss function (Eq. 5), the online learner guarantees that $\theta_t$ can estimate the reward, its gradient and hessian accurately, that is, for $\theta_t \sim p_t$, $\eta(\theta^\star, a_t) \approx \eta(\theta_t, a_t)$, $\nabla_a \eta(\theta^\star, a_{t-1}) \approx \nabla_a \eta(\theta_t, a_{t-1})$, and $\nabla_a^2 \eta(\theta^\star, a_{t-1}) \approx \nabla_a^2 \eta(\theta_t, a_{t-1})$.
ii. Because of (i), maximizing the virtual reward $\mathbb{E}_{\theta_t} \eta(\theta_t, a)$ w.r.t $a$ leads to improving the real reward function $\eta(\theta^\star, a)$ iteratively (in terms of finding second-order local improvement direction.)

Concretely, define the errors in rewards and its derivatives: $\Delta_{t,1} = |\eta(\theta_t, a_t) - \eta(\theta^\star, a_t)|$, $\Delta_{t,2} = |\eta(\theta_t, a_{t-1}) - \eta(\theta^\star, a_{t-1})|$, $\Delta_{t,3} = \|\nabla_a \eta(\theta_t, a_{t-1}) - \nabla_a \eta(\theta^\star, a_{t-1})\|_2$, and $\Delta_{t,4} = \|\nabla_a^2 \eta(\theta_t, a_{t-1}) - \nabla_a^2 \eta(\theta^\star, a_{t-1})\|_{\mathrm{sp}}$. Let $\Delta_t^2 = \sum_{i=1}^4 \Delta_{t,i}^2$ be the total error which measures how closeness between $\theta_t$ and $\theta^\star$.

Assuming that $\Delta_{t,j}$'s are small, to show (ii), we essentially view $a_t = \operatorname{argmax}_{a \in \mathcal{A}} \mathbb{E}_{\theta_t} \eta(\theta_t, a)$ as an approximate update on the real reward $\eta(\theta^\star, \cdot)$ and show it has local improvements if $a_{t-1}$ is not a critical point of the real reward:

$$\eta(\theta^\star, a_t) \gtrsim_{\Delta_t} \mathbb{E}_{\theta_t}[\eta(\theta_t, a_t)] \tag{15}$$

$$\geq \sup_a \mathbb{E}_{\theta_t}\left[\eta(\theta_t, a_{t-1}) + \langle a - a_{t-1}, \nabla_a \eta(\theta_t, a_{t-1})\rangle - \frac{\zeta_h}{2}\|a - a_{t-1}\|_2^2\right] \tag{16}$$

$$\gtrsim_{\Delta_t} \sup_a \mathbb{E}_{\theta_t}\left[\eta(\theta^\star, a_{t-1}) + \langle a - a_{t-1}, \nabla_a \eta(\theta^\star, a_{t-1})\rangle - \frac{\zeta_h}{2}\|a - a_{t-1}\|_2^2\right] \tag{17}$$

$$\geq \eta(\theta^\star, a_{t-1}) + \frac{1}{2\zeta_h}\|\nabla_a \eta(\theta^\star, a_{t-1})\|_2^2. \tag{18}$$

Here in equations (15) and (17), we use the symbol $\gtrsim_{\Delta_t}$ to present *informal* inequalities that are true up to some additive errors that depend on $\Delta_t$. This is because equation (15) holds up to

errors related to $\Delta_{t,1} = |\eta(\theta_t, a_t) - \eta(\theta^\star, a_t)|$, and equation (17) holds up to errors related to $\Delta_{t,2} = |\eta(\theta_t, a_{t-1}) - \eta(\theta^\star, a_{t-1})|$ and $\Delta_{t,3} = \|\nabla_a \eta(\theta_t, a_{t-1}) - \nabla_a \eta(\theta^\star, a_{t-1})\|_2$. Eq. (16) is a second-order Taylor expansion around the previous iteration $a_{t-1}$ and utilizes the definition $a_t = \arg\max_{a \in \mathcal{A}} \mathbb{E}_{\theta_t} \eta(\theta_t, a)$. Eq. (18) is a standard step to show the first-order improvement of gradient descent (the so-called "descent lemma"). We also remark that $a_t$ is the maximizer of the expected reward $\mathbb{E}_{\theta_t} \eta(\theta_t, a)$ instead of $\eta(\theta_t, a)$ because the adversary in online learning cannot see $\theta_t$ when choosing adversarial point $a_t$.

The following lemma formalizes the proof sketch above, and also extends it to considering second-order improvement. The proof can be found in Appendix C.2.

**Lemma C.1.** *In the setting of Theorem C.3, when $a_{t-1}$ is not an $(\epsilon, 6\sqrt{\zeta_{3\mathrm{rd}}\epsilon})$-approximate second order stationary point, we have $\eta(\theta^\star, a_t) \geq \eta(\theta^\star, a_{t-1}) + \min\left(\zeta_h^{-1}\epsilon^2/4, \zeta_{3\mathrm{rd}}^{-1/2}\epsilon^{3/2}\right) - C_1 \mathbb{E}_{\theta_t \sim p_t}[\Delta_t]$.*

Next, we show part (i) by linking the error $\Delta_t$ to the loss function $\ell$ (Eq. (5)) used by the online learner. The errors $\Delta_{t,1}, \Delta_{t,2}$ are already part of the loss function. Let $\tilde{\Delta}_{t,3} = \langle \nabla_a \eta(\theta_t, a_{t-1}) - \nabla_a \eta(\theta^\star, a_{t-1}), u_t \rangle$ and $\tilde{\Delta}_{t,4} = \langle \nabla_a^2 \eta(\theta_t, a_{t-1}) - \nabla_a^2 \eta(\theta^\star, a_{t-1}) u_t, v_t \rangle$ be the remaining two terms (without the clipping) in the loss (Eq. (5)). Note that $\tilde{\Delta}_{t,3}$ is supposed to bound $\Delta_{t,3}$ because $\mathbb{E}_{u_t}[\tilde{\Delta}_{t,3}^2] = \Delta_{t,3}^2$. Similarly, $\mathbb{E}_{u_t, v_t}[\tilde{\Delta}_{t,4}^2] = \|\nabla_a^2 \eta(\theta_t, a_{t-1}) - \nabla_a^2 \eta(\theta^\star, a_{t-1})\|_\mathrm{F}^2 \geq \Delta_{t,4}^2$. We clip $\tilde{\Delta}_{t,3}$ and $\tilde{\Delta}_{t,4}$ to make them uniformly bounded and improve the concentration with respect to the randomness of $u$ and $v$ (the clipping is conservative and is often not active). Let $\tilde{\Delta}_t^2 = \Delta_{t,1}^2 + \Delta_{t,2}^2 + \min\left(\kappa_1^2, \tilde{\Delta}_{t,3}^2\right) + \min\left(\kappa_2^2, \tilde{\Delta}_{t,4}^2\right)$ be the error received by the online learner at time $t$. The argument above can be rigorously formalized into a lemma that upper bound $\Delta_t$ by $\tilde{\Delta}_t$, which will be bounded by the sequential Rademacher complexity.

**Lemma C.2.** *In the setting of Theorem C.3, we have $\mathbb{E}_{u_{1:T}, v_{1:T}, \theta_{1:T}}[\sum_{t=1}^T \tilde{\Delta}_t^2] \geq \frac{1}{2}\mathbb{E}_{\theta_{1:T}}[\sum_{t=1}^T \Delta_t^2]$.*

We defer the proof to Appendix C.3. With Lemma C.1 and Lemma C.2, we can prove Theorem 3.1 by keeping track of the performance $\eta(\theta^\star, a_t)$. The full proof can be found in Appendix C.4.

## C.2 Proof of Lemma C.1

*Proof.* We prove the lemma by showing that algorithm 1 improves reward $\eta(\theta^\star, a_t)$ in the following two cases:

1. $\|\nabla_a \eta(\theta^\star, a_{t-1})\|_2 \geq \epsilon$, or

2. $\|\nabla_a \eta(\theta^\star, a_{t-1})\|_2 \leq \epsilon$ and $\lambda_{\max}\left(\nabla_a^2 \eta(\theta^\star, a_{t-1})\right) \geq 6\sqrt{\zeta_{3\mathrm{rd}}\epsilon}$.

**Case 1:** For simplicity, let $g_t = \nabla_a \eta(\theta^\star, a_{t-1})$. In this case we assume $\|g_t\|_2 \geq \epsilon$. Define function

$$\bar{\eta}_t(\theta, a) = \eta(\theta, a_{t-1}) + \langle a - a_{t-1}, \nabla_a \eta(\theta, a_{t-1}) \rangle - \zeta_h \|a - a_{t-1}\|_2^2 \tag{19}$$

to be the local first order approximation of function $\eta(\theta, a)$. By the Lipschitz assumption (namely, Assumption 2.1), we have $\eta(\theta, a) \geq \bar{\eta}_t(\theta, a)$ for all $\theta \in \Theta, a \in \mathcal{A}$. By the definition of $\Delta_{t,2}$ and $\Delta_{t,3}$, we get

$$\bar{\eta}_t(\theta_t, a) \geq \bar{\eta}_t(\theta^\star, a) - \Delta_{t,2} - \|a - a_{t-1}\|_2 \Delta_{t,3}. \tag{20}$$

In this case we have

$$
\begin{aligned}
\eta(\theta^\star, a_t) &\geq \mathbb{E}_{\theta_t \sim p_t}[\eta(\theta_t, a_t) - \Delta_{t,1}] \\
&\geq \sup_a \mathbb{E}_{\theta_t \sim p_t}[\eta(\theta_t, a) - \Delta_{t,1}] && \text{(By the optimality of } a_t\text{)} \\
&\geq \sup_a \mathbb{E}_{\theta_t \sim p_t}[\bar{\eta}_t(\theta_t, a) - \Delta_{t,1}] \\
&\geq \sup_a \mathbb{E}_{\theta_t \sim p_t}[\bar{\eta}_t(\theta^\star, a) - \Delta_{t,1} - \Delta_{t,2} - \|a - a_{t-1}\|_2 \Delta_{t,3}] && \text{(By Eq. (20))}
\end{aligned}
$$

$$\geq \mathbb{E}_{\theta_t \sim p_t} \left[ \eta(\theta^\star, a_{t-1}) + \frac{1}{4\zeta_h} \|g_t\|_2^2 - \Delta_{t,1} - \Delta_{t,2} - \frac{\|g_t\|_2}{2\zeta_h} \Delta_{t,3} \right] \qquad \text{(Take } a = a_{t-1} + \frac{g_t}{2\zeta_h})$$

$$\geq \eta(\theta^\star, a_{t-1}) + \frac{\epsilon^2}{4\zeta_h} - \mathbb{E}_{\theta_t \sim p_t} \left[ \left(2 + \frac{\zeta_g}{\zeta_h}\right) \Delta_t \right] \qquad \text{(By Cauchy-Schwarz)}$$

**Case 2:** Let $H_t = \nabla_a^2 \eta(\theta^\star, a_{t-1})$. Define $v_t \in \operatorname{argmax}_{v:\|v\|_2=1} v^\top H_t v$. In this case we have $\|g_t\|_2 \leq \epsilon$ and

$$v_t^\top H_t v_t \geq 6\sqrt{\zeta_{3\mathrm{rd}}\epsilon} \|v_t\|_2^2. \tag{21}$$

Define function

$$\hat{\eta}_t(\theta, a) = \eta(\theta, a_{t-1}) + \langle a - a_{t-1}, \nabla_a \eta(\theta, a_{t-1}) \rangle$$
$$+ \frac{1}{2} \langle \nabla_a^2 \eta(\theta_t, a_{t-1})(a - a_{t-1}), a - a_{t-1} \rangle - \frac{\zeta_{3\mathrm{rd}}}{2} \|a - a_{t-1}\|_2^3 \tag{22}$$

to be the local second order approximation of function $\eta(\theta, a)$. By the Lipschitz assumption (namely, Assumption 2.1), we have $\eta(\theta, a) \geq \hat{\eta}_t(\theta, a)$ for all $\theta \in \Theta, a \in \mathcal{A}$.

By Eq. (21), we can exploit the positive curvature by taking $a' = a_{t-1} + 4\sqrt{\frac{\epsilon}{\zeta_{3\mathrm{rd}}}} v_t$. Concretely, by basic algebra we get:

$$\hat{\eta}_t(\theta^\star, a') \geq \eta(\theta^\star, a_{t-1}) - \epsilon \|a' - a_{t-1}\|_2 + 3\sqrt{\zeta_{3\mathrm{rd}}\epsilon} \|a' - a_{t-1}\|_2^2 - \frac{\zeta_{3\mathrm{rd}}}{2} \|a' - a_{t-1}\|_2^3$$

$$\geq \eta(\theta^\star, a_{t-1}) + 12\sqrt{\frac{\epsilon^3}{\zeta_{3\mathrm{rd}}}}. \tag{23}$$

Combining with the definition of $\Delta_{t,2}, \Delta_{t,3}$ and $\Delta_{t,4}$, for any $a \in \mathcal{A}$ we get

$$\hat{\eta}_t(\theta_t, a) \geq \hat{\eta}_t(\theta^\star, a) - \Delta_{t,2} - \|a - a_{t-1}\|_2 \Delta_{t,3} - \frac{1}{2} \|a - a_{t-1}\|_2^2 \Delta_{t,4}. \tag{24}$$

As a result, we have

$$\eta(\theta^\star, a_t) \geq \mathbb{E}_{\theta_t \sim p_t} [\eta(\theta_t, a_t) - \Delta_{t,1}]$$

$$\geq \mathbb{E}_{\theta_t \sim p_t} [\eta(\theta_t, a') - \Delta_{t,1}] \qquad \text{(By the optimality of } a_t)$$

$$\geq \mathbb{E}_{\theta_t \sim p_t} \left[ \hat{\eta}_t(\theta^\star, a') - \Delta_{t,1} - \Delta_{t,2} - \|a - a_{t-1}\|_2 \Delta_{t,3} - \frac{1}{2} \|a - a_{t-1}\|_2^2 \Delta_{t,4} \right] \qquad \text{(By Eq. (24))}$$

$$\geq \eta(\theta^\star, a_{t-1}) + 12\sqrt{\frac{\epsilon^3}{\zeta_{3\mathrm{rd}}}} - \mathbb{E}_{\theta_t \sim p_t} \left[ \Delta_{t,1} + \Delta_{t,2} + 4\sqrt{\frac{\epsilon}{\zeta_{3\mathrm{rd}}}} \Delta_{t,3} + \frac{8\epsilon}{\zeta_{3\mathrm{rd}}} \Delta_{t,4} \right] \qquad \text{(By Eq. (23))}$$

$$\geq \eta(\theta^\star, a_{t-1}) + 12\sqrt{\frac{\epsilon^3}{\zeta_{3\mathrm{rd}}}} - \mathbb{E}_{\theta_t \sim p_t} [2\Delta_t]. \qquad \text{(When } 16\epsilon \leq \zeta_{3\mathrm{rd}})$$

Combining the two cases together, we get the desired result. $\qquad \square$

### C.3 Proof of Lemma C.2

*Proof.* Define $\mathcal{F}_t$ to be the $\sigma$-field generated by random variable $u_{1:t}, v_{1:t}, \theta_{1:t}$. In the following, we use $\mathbb{E}_t[\cdot]$ as a shorthand for $\mathbb{E}[\cdot \mid \mathcal{F}_t]$.

Let $g_t = \nabla_a \eta(\theta_t, a_{t-1}) - \nabla_a \eta(\theta^\star, a_{t-1})$. Note that condition on $\theta_t$ and $\mathcal{F}_{t-1}$, $\langle g_t, u_t \rangle$ follows the distribution $\mathcal{N}(0, \|g_t\|_2^2)$. By Assumption 2.1, $\|g_t\|_2 \leq 2\zeta_g = \kappa_1$. As a result,

$$\mathbb{E}_{t-1} \left[ \min\left( \kappa_1^2, \langle g_t, u_t \rangle^2 \right) \mid \theta_t \right] \geq \frac{1}{2} \mathbb{E}_{t-1} \left[ \langle g_t, u_t \rangle^2 \mid \theta_t \right] = \frac{1}{2} \|g_t\|_2^2. \tag{25}$$

By the tower property of expectation we get

$$\mathbb{E}_{t-1} \left[ \min\left( \kappa_1^2, \tilde{\Delta}_{t,3}^2 \right) \right] \geq \frac{1}{2} \mathbb{E}_{t-1} [\Delta_{t,3}^2]. \tag{26}$$

Now we turn to the term $\tilde{\Delta}_{t,4}^2$. Let $H_t = \nabla_a^2 \eta(\theta_t, a_{t-1}) - \nabla_a^2 \eta(\theta^\star, a_{t-1})$. Define a random variable $x = \left(u_t^\top H_t v_t\right)^2$. Note that $u_t, v_t$ are independent, we have

$$\mathbb{E}_{t-1}[x \mid \theta_t] = \mathbb{E}_{t-1}\left[\|H_t v_t\|_2^2 \mid \theta_t\right] = \|H_t\|_{\mathrm{F}}^2 \geq \|H_t\|_{\mathrm{sp}}^2. \tag{27}$$

Since $u_t, v_t$ are two Gaussian vectors, random variable $x$ has nice concentratebility properties. Therefore we can prove that the $\min$ operator in the definition of $\tilde{\Delta}_t$ does not change the expectation too much. Formally speaking, by Lemma F.6, condition on $\mathcal{F}_{t-1}$ and $\theta_t$, we have $\mathbb{E}\left[\min\left(\kappa_2^2, x\right)\right] \geq \frac{1}{2}\min\left(\zeta_h^2, \mathbb{E}[x]\right)$, which leads to

$$\mathbb{E}_{t-1}\left[\min(\kappa_2^2, \tilde{\Delta}_{t,4}^2)\right] \geq \frac{1}{2}\mathbb{E}_{t-1}\left[\min(\zeta_h^2, \|H_t\|_{\mathrm{F}}^2)\right] \geq \frac{1}{2}\mathbb{E}_{t-1}\left[\min(\zeta_h^2, \|H_t\|_{\mathrm{sp}}^2)\right] = \frac{1}{2}\mathbb{E}_{t-1}\left[\|H_t\|_{\mathrm{sp}}^2\right]. \tag{28}$$

Combining Eq. (26) and Eq. (28), we get the desired inequality. $\qquad\square$

## C.4 Proof of Theorem 3.1

In this section we show that Alg. 1 finds a $(\epsilon, 6\sqrt{\zeta_{\mathrm{3rd}}\epsilon})$-approximate local maximum in polynomial steps. In the following, we treat $\zeta_g, \zeta_h, \zeta_{\mathrm{3rd}}$ as constants.

*Proof of Theorem 3.1.* We prove this theorem by contradiction. Suppose $\Pr\left[a_{t+1} \in \mathfrak{A}_{\epsilon,6\sqrt{\zeta_{\mathrm{3rd}}\epsilon}}\right] \leq 0.5$ for all $t \in [T]$, we prove that $T \lesssim \widetilde{\mathcal{O}}(C_1^4 R(\Theta) \max\left(\zeta_h^4 \epsilon^{-8}, \zeta_{\mathrm{3rd}}^2 \epsilon^{-6}\right))$.

Define $\upsilon = \min\left(\frac{1}{4\zeta_h}\epsilon^2, \frac{1}{\zeta_{\mathrm{3rd}}^{1/2}}\epsilon^{3/2}\right)$. Recall that $C_1 = 2 + \frac{\zeta_g}{\zeta_h}$. By Lemma C.1, when $a_t$ is not a $(\epsilon, 6\sqrt{\zeta_{\mathrm{3rd}}\epsilon})$-approximate local maximum we have

$$\eta(\theta^\star, a_{t+1}) \geq \eta(\theta^\star, a_t) + \upsilon - C_1 \mathbb{E}_t[\Delta_{t+1}]. \tag{29}$$

Similar to the proof of Theorem C.3, when $a_t$ is a $(\epsilon, 6\sqrt{\zeta_{\mathrm{3rd}}\epsilon})$-approximate local maximum we have

$$\eta(\theta^\star, a_{t+1}) \geq \eta(\theta^\star, a_t) - C_1 \mathbb{E}_t[\Delta_{t+1}]. \tag{30}$$

As a result, when $\Pr\left[a_{t+1} \in \mathfrak{A}_{\epsilon,6\sqrt{\zeta_{\mathrm{3rd}}\epsilon}}\right] \leq 0.5$ we get

$$\mathbb{E}[\eta(\theta^\star, a_{t+1})] \geq \mathbb{E}[\eta(\theta^\star, a_t)] + \frac{\upsilon}{2} - C_1 \mathbb{E}[\Delta_{t+1}]. \tag{31}$$

Take summation of Eq. (31) over $t \in [T]$ leads to

$$\mathbb{E}[\eta(\theta^\star, a_T) - \eta(\theta^\star, a_0)] \geq \frac{\upsilon T}{2} - C_1 \mathbb{E}\left[\sum_{t=1}^T \Delta_t\right]. \tag{32}$$

Lemma C.2 leads to

$$\mathbb{E}\left[\sum_{t=1}^T \Delta_t\right] \leq \sqrt{2T\mathbb{E}\left[\sum_{t=1}^T \tilde{\Delta}_t^2\right]} \leq 2T^{3/4}(R(\Theta)\mathrm{polylog}(T))^{1/4}. \tag{33}$$

Combining with Eq. (32) we have

$$1 \geq \mathbb{E}[\eta(\theta^\star, a_T) - \eta(\theta^\star, a_0)] \geq \frac{\upsilon T}{2} - 2C_1 T^{3/4}(R(\Theta)\mathrm{polylog}(T))^{1/4}. \tag{34}$$

As a result, we can solve an upper bound of $T$. In particular, we get

$$T \lesssim C_1^4 R(\Theta) \max\left(\zeta_h^4 \epsilon^{-8}, \zeta_{\mathrm{3rd}}^2 \epsilon^{-6}\right) \mathrm{polylog}(R(\Theta), 1/\epsilon, C_1, \zeta_h, \zeta_{\mathrm{3rd}}). \tag{35}$$

Consequently, when $T \geq \widetilde{\mathcal{O}}(C_1^4 R(\Theta) \max\left(\zeta_h^4 \epsilon^{-8}, \zeta_{\mathrm{3rd}}^2 \epsilon^{-6}\right))$, there exists $t \in [T]$ such that $\Pr\left[a_{t+1} \in \mathfrak{A}_{\epsilon,6\sqrt{\zeta_{\mathrm{3rd}}\epsilon}}\right] > 0.5$.

Finally, by running running Alg. 1 $\log(1/\delta)$ times, we can get a high probability guarantee. $\qquad\square$

## C.5 Local Regret

We also define the "local regret" by comparing with an approximate local maximum. Formally speaking, let $\mathfrak{A}_{\epsilon_g,\epsilon_h}$ be the set of all $(\epsilon_g, \epsilon_h)$-approximate local maximum of $\eta(\theta^\star, \cdot)$. The $(\epsilon_g, \epsilon_h)$-local regret of a sequence of actions $a_1, \ldots, a_T$ is defined as

$$\text{REG}_{\epsilon_g,\epsilon_h}(T) = \sum_{t=1}^{T} (\inf_{a \in \mathfrak{A}_{\epsilon_g,\epsilon_h}} \eta(\theta^\star, a) - \eta(\theta^\star, a_t)). \tag{36}$$

Our goal is to achieve a $(\epsilon_g, \epsilon_h)$-local regret that is sublinear in $T$ and inverse polynomial in $\epsilon_g$ and $\epsilon_h$. With a sublinear regret (i.e., $\text{REG}_{\epsilon_g,\epsilon_h}(T) = o(T)$), the average performance $\frac{1}{T} \sum_{t=1}^{T} \eta(\theta^\star, a_t)$, converges to that of an approximate local maximum of $\eta(\theta^\star, \cdot)$. Hazan et al. [35] also work on local regret in the setting of online non-convex games. Their local regret notation is different from ours.

The main theorem for local regret guarantee is stated below.

**Theorem C.3.** *Let $\mathfrak{R}_T$ be the sequential Rademacher complexity of the family of the losses defined in Eq. (5). Let $C_1 = 2 + \zeta_g/\zeta_h$. Under Assumption 2.1, for any $\epsilon \leq \min\left(1, \zeta_{3\text{rd}}/16\right)$, we can bound the $(\epsilon, 6\sqrt{\zeta_{3\text{rd}}\epsilon})$-local regret of Alg. 1 from above by*

$$\mathbb{E}\big[\text{REG}_{\epsilon,6\sqrt{\zeta_{3\text{rd}}\epsilon}}(T)\big] \leq \left(1 + C_1\sqrt{4T\mathfrak{R}_T}\right) \max\left(4\zeta_h\epsilon^{-2}, \sqrt{\zeta_{3\text{rd}}}\epsilon^{-3/2}\right). \tag{37}$$

Note that when the sequential Rademacher complexity $\mathfrak{R}_T$ is bounded by $\widetilde{\mathcal{O}}(R\sqrt{T})$ (which is typical), we have $\mathcal{O}(\sqrt{T\mathfrak{R}_T}) = \widetilde{\mathcal{O}}(T^{3/4}) = o(T)$ regret. As a result, Alg. 1 achieves a $\mathcal{O}(\text{poly}(1/\epsilon))$ sample complexity by the sample complexity-regret reduction [39, Section 3.1]. The proof is deferred to Section C.6.

## C.6 Proof of Theorem C.3

*Proof.* Let $\delta_t = \inf_{a \in \mathfrak{A}_{(\epsilon),6\sqrt{\zeta_{3\text{rd}}\epsilon}}} \eta(\theta^\star, a) - \eta(\theta^\star, a_t)$. By the definition of regret we have $\text{REG}_{\epsilon,6\sqrt{\zeta_{3\text{rd}}\epsilon}}(T) = \sum_{t=1}^{T} \delta_t$. Define $\upsilon = \min\left(\frac{1}{4\zeta_h}\epsilon^2, \frac{1}{\zeta_{3\text{rd}}^{1/2}}\epsilon^{3/2}\right)$ for simplicity. Recall that $C_1 = 2 + \frac{\zeta_g}{\zeta_h}$. In the following we prove by induction that for any $t_0$,

$$\mathbb{E}_{t_0-1}\left[\sum_{t=t_0}^{T} \delta_t\right] \leq \mathbb{E}_{t_0-1}\left[\frac{1}{\upsilon}\left(\delta_{t_0} + C_1 \sum_{t=t_0+1}^{T} \Delta_t\right)\right]. \tag{38}$$

For the base case where $t_0 = T$ Eq. (38) trivially holds because $\upsilon \leq 1$.

Now suppose Eq. (38) holds for any $t > t_0$ and consider time step $t_0$. When $a_{t_0} \notin \mathfrak{A}_{(\epsilon),6\sqrt{\zeta_{3\text{rd}}\epsilon}}$, applying Lemma C.1 we get $\eta(\theta^\star, a_{t_0+1}) \geq \eta(\theta^\star, a_{t_0}) + \upsilon - C_1\mathbb{E}_{t_0}[\Delta_{t_0+1}]$. By basic algebra we get,

$$\delta_{t_0+1} \leq \delta_{t_0} - \upsilon + C_1\mathbb{E}_{t_0}[\Delta_{t_0+1}]. \tag{39}$$

As a result,

$$\begin{aligned}
\mathbb{E}_{t_0-1}\left[\sum_{t=t_0}^{T} \delta_t\right] &= \mathbb{E}_{t_0-1}\left[\delta_{t_0} + \sum_{t=t_0+1}^{T} \delta_t\right] \\
&\leq \mathbb{E}_{t_0-1}\left[\delta_{t_0} + \frac{1}{\upsilon}\left(\delta_{t_0+1} + C_1 \sum_{t=t_0+2}^{T} \Delta_t\right)\right] \quad \text{(By induction hypothesis)} \\
&\leq \mathbb{E}_{t_0-1}\left[\delta_{t_0} - 1 + \frac{1}{\upsilon}\left(\delta_{t_0} + C_1\Delta_{t_0+1} + C_1 \sum_{t=t_0+2}^{T} \Delta_t\right)\right] \quad \text{(By Eq. (39))} \\
&\leq \mathbb{E}_{t_0-1}\left[\frac{1}{\upsilon}\left(\delta_{t_0} + C_1 \sum_{t=t_0+1}^{T} \Delta_t\right)\right].
\end{aligned}$$

On the other hand, when $a_{t_0} \in \mathfrak{A}_{(\epsilon),6\sqrt{\zeta_{3rd}\epsilon}}$ we have

$$\eta(\theta^\star, a_{t_0+1}) \geq \mathbb{E}_{\theta_{t_0+1}}[\eta(\theta_{t_0+1}, a_{t_0+1}) - \Delta_{t_0+1,1}]$$

$$\geq \mathbb{E}_{\theta_{t_0+1}}[\eta(\theta_{t_0+1}, a_{t_0}) - \Delta_{t_0+1,1}] \qquad \text{(By the optimality of } a_{t_0+1})$$

$$\geq \mathbb{E}_{\theta_{t_0+1}}[\eta(\theta^\star, a_{t_0}) - \Delta_{t_0+1,1} - \Delta_{t_0+1,2}] \geq \eta(\theta^\star, a_{t_0}) - C_1 \mathbb{E}_{\theta_{t_0+1}}[\Delta_{t_0+1}].$$

Consequently, by basic algebra we get $\delta_{t_0+1} \leq \delta_{t_0} + C_1 \mathbb{E}_{t_0}[\Delta_{t_0+1}]$. Note that since $a_{t_0} \in \mathfrak{A}_{(\epsilon),6\sqrt{\zeta_{3rd}\epsilon}}$, we have $\delta_{t_0} \leq 0$. As a result,

$$\mathbb{E}_{t_0-1}\left[\sum_{t=t_0}^{T} \delta_t\right] \leq \mathbb{E}_{t_0-1}\left[\delta_{t_0} + \frac{1}{\upsilon}\left(\delta_{t_0+1} + C_1 \sum_{t=t_0+2}^{T} \Delta_t\right)\right] \qquad \text{(By induction hypothesis)}$$

$$\leq \mathbb{E}_{t_0-1}\left[\frac{1}{\upsilon}\left(\delta_{t_0+1} + C_1 \sum_{t=t_0+2}^{T} \Delta_t\right)\right] \qquad (\delta_{t_0} \leq 0)$$

$$\leq \mathbb{E}_{t_0-1}\left[\frac{1}{\upsilon}\left(\delta_{t_0} + C_1\Delta_{t_0+1} + C_1 \sum_{t=t_0+2}^{T} \Delta_t\right)\right]$$

$$\leq \mathbb{E}_{t_0-1}\left[\frac{1}{\upsilon}\left(\delta_{t_0} + C_1 \sum_{t=t_0+1}^{T} \Delta_t\right)\right].$$

Combining the two cases together we prove Eq. (38). It follows that

$$\mathbb{E}\left[\text{REG}_{\epsilon,6\sqrt{\zeta_{3rd}\epsilon}}(T)\right] = \mathbb{E}\left[\sum_{t=0}^{T} \delta_t\right] \leq \mathbb{E}\left[\frac{1}{\upsilon}\left(\delta_0 + C_1 \sum_{t=1}^{T} \Delta_t\right)\right] \qquad (40)$$

$$\leq \frac{1}{\upsilon}\left(1 + C_1\mathbb{E}\left[\sqrt{T\sum_{t=1}^{T} \Delta_t^2}\right]\right) \leq \frac{1}{\upsilon}\left(1 + C_1\sqrt{T\mathbb{E}\left[\sum_{t=1}^{T} \Delta_t^2\right]}\right). \qquad (41)$$

Note that when realizability holds, we have $\inf_\theta \sum_{t=1}^{T} \ell((x_t, y_t); \theta) = 0$. Therefore, by Lemma C.2 and the definition of online learning regret (see Eq. (1)) we have

$$\mathbb{E}\left[\text{REG}_{\epsilon,6\sqrt{\zeta_{3rd}\epsilon}}(T)\right] \leq \frac{1}{\upsilon}\left(1 + C_1\sqrt{2T\mathbb{E}\left[\sum_{t=1}^{T} \tilde{\Delta}_t^2\right]}\right) \leq \frac{1}{\upsilon}\left(1 + C_1\sqrt{4T\mathfrak{R}_T}\right). \qquad (42)$$

$\square$

## C.7 Instantiations of Theorem 3.1

In this section we rigorously prove the instantiations discussed in Section 3.

**Linear bandit with finite model class.** A full proof of this claim needs a few steps: (i) realizing that $\eta(\theta^\star, a)$ is concave in $a$ with no bad local maxima, and therefore our local regret and the standard regret coincide (up to some conversion of the errors); (ii) invoking Rakhlin et al. [63, Lemma 3] to show that the sequential Rademacher complexity $\mathfrak{R}_T$ is bounded by $\mathcal{O}(\sqrt{(2\log|\Theta|)/T})$, and (iii) verifying $\tilde{\eta}$ satisfies the conditions (Assumption 2.1) on the actions that the algorithm will visit.

Recall that the linear bandit reward is given by $\eta(\theta, a) = \langle\theta, a\rangle$, and the constrained reward is $\tilde{\eta}(\theta, a) = \eta(\theta, a) - \frac{1}{2}\|a\|_2^2$.

In order to deal with $\ell_2$ regularization which violates Assumption 2.1, we bound the set of actions Alg. 1 takes. Consider the regularized reward $\tilde{\eta}(\theta, a)$. Recall that Alg. 1 chooses action $a_t = \text{argmax}_{a\in\mathcal{A}} \mathbb{E}_{\theta_t\sim p_t}[\tilde{\eta}(\theta_t, a)]$. By optimality condition we have $a_t = \mathbb{E}_{\theta_t\sim p_t}[\theta_t]$. Consequently $\|a_t\|_2 \leq \mathbb{E}_{\theta_t\sim p_t}[\|\theta_t\|_2] \leq 1$.

Because we only apply Lemma C.1 and Lemma C.2 to actions that is taken by the algorithm, Theorem C.3 holds even if Assumption 2.1 is satisfied locally for $\|a\|_2 \lesssim 1$. Since the gradient and

Hessian of regularization term is $a$ and $I_d$ respectively, we have $\|\nabla_a \tilde\eta(\theta, a)\|_2 \lesssim \|\nabla_a \eta(\theta, a)\|_2 + 1$ and $\|\nabla_a^2 \tilde\eta(\theta, a)\|_{\mathrm{sp}} \lesssim \|\nabla_a^2 \eta(\theta, a)\|_{\mathrm{sp}} + 1$ when $\|a\|_2 \lesssim 1$, which verifies Assumption 2.1.

In the following we prove that any $(\epsilon, 1)$-approximate local maximum for $\tilde\eta(\theta^\star, a)$ is an $\epsilon$-suboptimal action for $\eta(\theta^\star, a)$. Note that $\nabla_a \tilde\eta(\theta, a) = \theta - a$. Therefore, for any $a \in \mathfrak{A}_{(\epsilon, 1)}$ we have $\|\theta^\star - a\|_2 \leq \epsilon$. Applying Lemma F.11 we have

$$(1 - \langle \theta^\star, a \rangle)^2 \leq \|\theta^\star - a\|_2^2 \leq \epsilon^2. \tag{43}$$

Combining with the fact that $\|a_t\|_2 \leq 1$ for any $t \in [T]$, we prove the claim.

**Linear bandit with sparse or structured model vectors.**   In this case, the reduction is exactly the same as that in linear bandit. In the following we prove that the sparse linear hypothesis has a small covering number. Note that the $\log|\Theta|$ sample complexity bound fits perfectly with the covering number technique. That is, we can discretize the hypothesis $\Theta$ by finding a $1/\mathrm{poly}(d, 1/\epsilon)$-covering of the loss function $\mathcal{L} = \{\ell(\cdot, \theta) : \theta \in \Theta\}$. And then the sample complexity of our algorithm depends polynomially on the log-covering number. Since the log-covering number of the set of $s$-sparse vectors is bounded by $\mathcal{O}(s \log(dT))$, we get the desired result.

For completeness, in the following we prove that the Eluder dimension for sparse linear model is $\Omega(d)$.

**Lemma C.4.** *Let $e_1, \cdots, e_d$ be the basis vectors and $f_i(a) = \langle e_i, a \rangle$. Specifically, define $f_0(a) = 0$. Define the function class $\mathcal{F} = \{f_i : 0 \leq i \leq d\}$. The Eluder dimension of $\mathcal{F}$ is at least $d$.*

*Proof.* In order to prove the lower bound for Eluder dimension, we only need to find a sequence $a_1, \cdots, a_d$ such that $a_i$ is independent with its predecessors. In the sequel we consider the action sequence $a_1 = e_1, a_2 = e_2, \cdots, a_d = e_d$.

Now we prove that for any $i \in [d]$, $a_i$ is independent with $a_j$ where $j < i$. Indeed, consider functions $f_i$ and $f_0$. By definition we have $f_i(a_j) = f_0(a_j), \forall j < i$. However, $f_i(a_i) = 1 \neq 0 = f_0(a_i)$.  $\square$

**Deterministic logistic bandits.**   For deterministic logistic bandits, the reward function is given by $\eta(\theta, a) = (1 + e^{-\langle \theta, a \rangle})^{-1}$. The model class is $\Theta \subseteq S^{d-1}$ and the action space is $\mathcal{A} = S^{d-1}$. Similarly, we run Alg. 1 on an unbounded action space with regularized loss $\tilde\eta(\theta, a) = \eta(\theta, a) - \frac{c}{2}\|a\|_2^2$ where $c = e(e+1)^{-2}$ is a constant. The optimal action in this case is $a^\star = \theta^\star$. Note that the loss function is not concave, but it satisfies that all local maxima are global. As a result, we claim that our algorithm finds an $\epsilon$-suboptimal in $\widetilde{\mathcal{O}}(\log|\Theta|\epsilon^{-8})$ steps. Compared with algorithms that specially designed for logistic bandits [22, 15, 23, 52], our regret bound obtained by reduction is not optimal.

In the following we prove that any $(\epsilon, 1)$-approximate local maximum for $\tilde\eta(\theta^\star, a)$ is an $\mathcal{O}(\epsilon)$-suboptimal action for $\eta(\theta^\star, a)$. First of all, we bound the set of actions Alg. 1 can choose. By basic algebra we get

$$\nabla_a \tilde\eta(\theta, a) = \frac{\exp(-\theta^\top a)}{(1 + \exp(-\theta^\top a))^2} \theta - ca. \tag{44}$$

As a result, we have $a^\star = \theta^\star$. Recall that $a_t = \mathrm{argmax}_{a \in \mathcal{A}} \mathbb{E}_{\theta_t \sim p_t}[\tilde\eta(\theta_t, a)]$. By optimality condition we get

$$ca = \mathbb{E}_{\theta_t}\left[\frac{\exp(-\theta_t^\top a)}{(1 + \exp(-\theta_t^\top a))^2} \theta_t\right]. \tag{45}$$

Multiply $a^\top$ to both hand side we get

$$c\|a\|_2^2 = \mathbb{E}_{\theta_t}\left[\frac{\exp(-\theta_t^\top a)}{(1 + \exp(-\theta_t^\top a))^2} \theta_t^\top a\right]. \tag{46}$$

Define $f(x) = \frac{x \exp(-x)}{c(1 + \exp(-x))^2}$. Eq. (46) implies

$$\|a\|_2^2 = \mathbb{E}_{\theta_t}[f(\theta_t^\top a)] \leq \sup_{x \in [-\|a\|_2, \|a\|_2]} f(x). \tag{47}$$

Solving Eq. (47) we get $\|a\|_2 \le 1$.

Now translate an $(\epsilon, 1)$-approximate local maximum for $\tilde{\eta}(\theta^\star, a)$ to an $\mathcal{O}(\epsilon)$-approximate optimal action for $\eta(\theta^\star, a)$. Note that $\tilde{\eta}(\theta^\star, \cdot)$ is $(1/20)$-strongly concave. As a result, for any $\epsilon \in \mathfrak{A}_{(\epsilon, 1)}$ we get

$$\tilde{\eta}(\theta^\star, a^\star) - \tilde{\eta}(\theta^\star, a) \lesssim \|\nabla_a \tilde{\eta}(\theta^\star, a)\|_2^2 \lesssim \epsilon^2. \tag{48}$$

Define $r(x) \triangleq (1 + \exp(-x))$ for shorthand. By Taylor expansion, for any $x \in \mathbb{R}$ there exists $\xi \in \mathbb{R}$ such that $r(x) = r(1) + (x-1)r'(x) + (x-1)^2 r''(\xi)$. As a result,

$$
\begin{aligned}
\tilde{\eta}(\theta^\star, a^\star) - \tilde{\eta}(\theta^\star, a) &= r(1) - \frac{c}{2} - r(\langle \theta^\star, a \rangle) + \frac{c}{2}\|a\|_2^2 \\
&= (1 - \langle \theta^\star, a \rangle)r'(1) - (1 - \langle \theta^\star, a \rangle)^2 r''(\xi) - \frac{c}{2} + \frac{c}{2}\|a\|_2^2 \\
&= \frac{c}{2} - c\langle \theta^\star, a \rangle + \frac{c}{2}\|a\|_2^2 - (1 - \langle \theta^\star, a \rangle)^2 r''(\xi) &&\text{(Recall that } r'(1) = c.) \\
&= \frac{c}{2}\|\theta^\star - a\|_2^2 - (1 - \langle \theta^\star, a \rangle)^2 r''(\xi) \\
&\ge \left(\frac{c}{2} - r''(\xi)\right)(1 - \langle \theta^\star, a \rangle)^2 &&\text{(By Lemma F.11)} \\
&\ge (1 - \langle \theta^\star, a \rangle)^2 / 50 \gtrsim (r(1) - r(\langle \theta^\star, a \rangle))^2 &&\text{(The reward function is Lipschitz.)} \\
&\gtrsim (\eta(\theta^\star, a^\star) - \eta(\theta^\star, a))^2.
\end{aligned}
$$

Consequently, $\eta(\theta^\star, a) \ge \eta(\theta^\star, a^\star) - \mathcal{O}(\epsilon)$.

**Two-layer neural network.** Recall that a two-layer neural network is defined by $\eta((W_1, W_2), a) = W_2 \sigma(W_1 a)$, where $\sigma$ is the activation function. For a matrix $W_1 \in \mathbb{R}^{m \times d}$, the $(1, \infty)$-norm is defined by $\max_{i \in [m]} \sum_{j=1}^d |[W_1]_{i,j}|$. We make the following assumptions regarding the activation function.

**Assumption C.5.** *For any $x, y \in \mathbb{R}$, the activation function $\sigma(\cdot)$ satisfies*

$$\sup_x |\sigma(x)| \le 1, \quad \sup_x |\sigma'(x)| \le 1, \quad \sup_x |\sigma''(x)| \le 1, \tag{49}$$

$$|\sigma''(x) - \sigma''(y)| \le |x - y|. \tag{50}$$

The following theorem summarized our result in this setting.

**Theorem C.6.** *Let $\Theta = \{(W_1, W_2) : \|W_2\|_1 \le 1, \|W_1\|_{1,\infty} \le 1\}$ be the parameter hypothesis. Under the setting of Theorem 3.1 with Assumption C.5, Alg. 1 finds an $(\epsilon, 6\sqrt{\zeta_{3rd}\epsilon})$-approximate local maximum in $\widetilde{\mathcal{O}}(\epsilon^{-8}\text{polylog}(d))$ steps. In addition, if the neural network is input concave, Alg. 1 finds an $\epsilon$-suboptimal action in $\widetilde{\mathcal{O}}(\epsilon^{-4}\text{polylog}(d))$ steps.*

*Proof.* We prove the theorem by first bounding the sequential Rademacher complexity of the loss function, and then applying Theorem C.3. Let $\theta = (W_1, W_2)$. Recall that $u \odot v$ denotes the element-wise product. By basic algebra we get,

$$\langle \nabla_a \eta(\theta, a), u \rangle = W_2(\sigma'(W_1 a) \odot W_1 u), \tag{51}$$

$$u^\top \nabla_a^2 \eta(\theta, a) v = W_2(\sigma''(W_1 a) \odot W_1 u \odot W_1 v). \tag{52}$$

First of all, we verify that the regularized reward $\tilde{\eta}(\theta, a) \triangleq \eta(\theta, a) - \frac{1}{2}\|a\|_2^2$ satisfies Assumption 2.1. Indeed we have

$$\|\nabla_a \eta(\theta, a)\|_2 = \sup_{u \in S^{d-1}} \langle \nabla_a \eta(\theta, a), u \rangle \le 1,$$

$$\|\nabla_a^2 \eta(\theta, a)\|_{\text{sp}} = \sup_{u, v \in S^{d-1}} u^\top \nabla_a^2 \eta(\theta, a) v \le 1,$$

$$\|\nabla_a^2 \eta(\theta, a_1) - \nabla_a^2 \eta(\theta, a_2)\|_{\text{sp}} = \sup_{u, v \in S^{d-1}} W_2((\sigma''(W_1 a_1) - \sigma''(W_1 a_2)) \odot W_1 u \odot W_1 v)$$

$$\le \|a_1 - a_2\|_2.$$

Observe that $|\eta(\theta, a)| \leq \|a\|_\infty$, we have $\tilde{\eta}(\theta, a) < 0$ when $\|a\|_2 > 2$. As a result, action $a_t$ taken by Alg. 1 satisfies $\|a_t\|_2 \leq 2$ for all $t$. Since the gradient and Hessian of regularization term is $a$ and $I_d$ respectively, we have $\|\nabla_a \tilde{\eta}(\theta, a)\|_2 \lesssim \|\nabla_a \eta(\theta, a)\|_2 + 1$ and $\|\nabla_a^2 \tilde{\eta}(\theta, a)\|_{sp} \lesssim \|\nabla_a^2 \eta(\theta, a)\|_{sp} + 1$. It follows that Assumption 2.1 holds with constant Lipschitzness for actions $a$ such that $\|a\| \lesssim 1$.

In the following we bound the sequential Rademacher complexity of the loss function. By Rakhlin et al. [62, Proposition 15], we can bound the sequential Rademacher complexity of $\Delta_{t,1}^2$ and $\Delta_{t,2}^2$ by $\widetilde{\mathcal{O}}\left(\sqrt{T \log d}\right)$. Next we turn to higher order terms.

First of all, because the $(1, \infty)$ norm of $W_1$ is bounded, we have $\|W_1 u\|_\infty \leq \|u\|_\infty$. It follows from the upper bound of $\sigma'(x)$ that $\|\sigma'(W_1 a) \odot W_1 u\|_\infty \leq \|u\|_\infty$. Therefore we get

$$\langle \nabla_a \eta(\theta, a), u \rangle \leq \|W_2\|_1 \|\sigma'(W_1 a) \odot W_1 u\|_\infty \leq \|u\|_\infty. \tag{53}$$

Similarly, we get

$$u^\top \nabla_a^2 \eta(\theta, a) v \leq \|u\|_\infty \|v\|_\infty. \tag{54}$$

Let $B = (1 + \|u\|_\infty)(1 + \|v\|_\infty)$ for shorthand. We consider the error term $\tilde{\Delta}_{t,3}^2 = (\langle \nabla_a \eta(\theta, a), u \rangle - [y_t]_3)^2$. Let $\mathcal{G}_1$ be the function class $\{(\langle \nabla_a \eta(\theta, a), u \rangle - [y_t]_3)^2 : \theta \in \Theta\}$, and $\mathcal{G}_2 = \{\langle \nabla_a \eta(\theta, a), u \rangle : \theta \in \Theta\}$. Applying Rakhlin et al. [62, Lemma 4] we get

$$\mathfrak{R}_T(\mathcal{G}_1) \lesssim B \log^{3/2}(T^2) \mathfrak{R}_T(\mathcal{G}_2).$$

Define $\mathcal{G}_3 = \{\sigma'(w_1^\top a) \cdot w_1^\top u : w_1 \in \mathbb{R}^d, \|w_1\|_1 \leq 1\}$. In the following we show that $\mathfrak{R}_T(\mathcal{G}_2) \lesssim \mathfrak{R}_T(\mathcal{G}_3)$. For any sequence $u_1, \cdots, u_T$ and $\mathcal{A}$-valued tree $\boldsymbol{a}$, we have

$$\mathfrak{R}_T(\mathcal{G}_2) = \mathbb{E}_\epsilon \left[ \sup_{\substack{W_2 : \|W_2\|_1 \leq 1 \\ g_1, \cdots, g_w \in \mathcal{G}_3}} \sum_{t=1}^T \epsilon_t \left( \sum_{j=1}^w [W_2]_j g_j(\boldsymbol{a}_t(\epsilon)) \right) \right] \tag{55}$$

$$\leq \mathbb{E}_\epsilon \left[ \sup_{\substack{W_2 : \|W_2\|_1 \leq 1 \\ g_1, \cdots, g_w \in \mathcal{G}_3}} \|W_2\|_1 \sup_{j \in [w]} \left| \sum_{t=1}^T \epsilon_t g_j(\boldsymbol{a}_t(\epsilon)) \right| \right] \tag{56}$$

$$\leq \mathbb{E}_\epsilon \left[ \sup_{g \in \mathcal{G}_3} \left| \sum_{t=1}^T \epsilon_t (g_j(\boldsymbol{a}_t(\epsilon))) \right| \right]. \tag{57}$$

Since we have $0 \in \mathcal{G}_3$ by taking $w_1 = 0$, by symmetricity we have

$$\mathbb{E}_\epsilon \left[ \sup_{g \in \mathcal{G}_3} \left| \sum_{t=1}^T \epsilon_t (g_j(\boldsymbol{a}_t(\epsilon))) \right| \right] \leq 2 \mathbb{E}_\epsilon \left[ \sup_{g \in \mathcal{G}_3} \sum_{t=1}^T \epsilon_t (g_j(\boldsymbol{a}_t(\epsilon))) \right] = 2 \mathfrak{R}_T(\mathcal{G}_3). \tag{58}$$

Now we bound $\mathfrak{R}_T(\mathcal{G}_3)$ by applying the composition lemma of sequential Rademacher complexity (namely Rakhlin et al. [62, Lemma 4]). First of all we define a relaxed function hypothesis $\mathcal{G}_4 = \{\sigma'((w_1')^\top a) \cdot w_1^\top u : w_1, w_1' \in \mathbb{R}^d, \|w_1\|_1 \leq 1, \|w_1'\|_1 \leq 1\}$. Since $\mathcal{G}_3 \subset \mathcal{G}_4$ we have $\mathfrak{R}_T(\mathcal{G}_3) \leq \mathfrak{R}_T(\mathcal{G}_4)$. Note that we have $|\sigma'(w_1^\top a)| \leq 1$ and $w_1^\top u \leq \|u\|_\infty$. Let $\phi(x, y) = xy$, which is $(3c)$-Lipschitz for $|x|, |y| \leq c$. Define $\mathcal{G}_5 = \{\sigma'(w_1^\top a) : w_1 \in \mathbb{R}^d, \|w_1\|_1 \leq 1\}$ and $\mathcal{G}_6 = \{w_1^\top u : w_1 \in \mathbb{R}^d, \|w_1\|_1 \leq 1\}$. Rakhlin et al. [62, Lemma 4] gives $\mathfrak{R}_T(\mathcal{G}_4) \lesssim B \log^{3/2}(T^2)(\mathfrak{R}_T(\mathcal{G}_5) + \mathfrak{R}_T(\mathcal{G}_6))$. Note that $\mathcal{G}_5$ is a generalized linear hypothesis and $\mathcal{G}_6$ is linear, we have $\mathfrak{R}_T(\mathcal{G}_5) \lesssim B \log^{3/2}(T^2)\sqrt{T \log(d)}$ and $\mathfrak{R}_T(\mathcal{G}_6) \lesssim B\sqrt{T \log(d)}$.

In summary, we get $\mathfrak{R}_T(\mathcal{G}_1) = \mathcal{O}\left(\text{poly}(B)\text{polylog}(d, T)\sqrt{T}\right)$. Since the input $u_t \sim \mathcal{N}(0, I_{d \times d})$, we have $B \lesssim \log(dT)$ with probability $1/T$. As a result, the distribution dependent Rademacher complexity of $\tilde{\Delta}_{t,3}^2$ in this case is bounded by $\mathcal{O}\left(\text{polylog}(d, T)\sqrt{T}\right)$.

Similarly, we can bound the sequential Rademacher complexity of the Hessian term $\tilde{\Delta}_{t,4}^2$ by $\mathcal{O}\left(\text{polylog}(d, T)\sqrt{T}\right)$ by applying composition lemma with Lipschitz function $\phi(x, y, z) = xyz$

with bounded $|x|, |y|, |z|$. By Rakhlin et al. [62, Lemma 4], composing with the min operator only introduces $\text{poly}(\log(T))$ terms in the sequential Rademacher complexity. As a result, the sequential Rademacher complexity of the loss function can be bounded by

$$\mathfrak{R}_T = \mathcal{O}\Big(\text{polylog}(d,T)\sqrt{T}\Big).$$

Applying Theorem 3.1, the sample complexity of Alg. 1 is bounded by $\widetilde{\mathcal{O}}\big(\epsilon^{-8}\text{polylog}(d)\big)$.

When the neural network is input concave (see [5]), the regularized reward $\tilde{\eta}(\theta, a)$ is $\Omega(1)$-strongly concave. As a result, for any $a \in \mathfrak{A}_{\epsilon,1}$ we have $\tilde{\eta}(\theta^\star, a) \geq \tilde{\eta}(\theta^\star, a^\star) - \mathcal{O}(\epsilon^2)$. Hence, Alg. 1 finds an $\epsilon$-suboptimal action for regularized loss in $\widetilde{\mathcal{O}}\big(\epsilon^{-4}\text{polylog}(d)\big)$ steps. $\qquad\square$

# D   Missing Proofs in Section 4

First of all, we present our algorithm in Alg. 2.

---

**Algorithm 2 Vi**rtual Ascent with **Onlin**e Model Learner (ViOlin for RL)
---

1:  Let $\mathcal{H}_0 = \emptyset$; choose $a_0 \in \mathcal{A}$ arbitrarily.
2:  **for** $t = 1, 2, \cdots$ **do**
3:      Run $\mathcal{R}$ on $\mathcal{H}_{t-1}$ with loss function $\ell$ (defined in Eq. (6)) and obtain $p_t = \mathcal{A}(\mathcal{H}_{t-1})$.
4:      $\psi_t \leftarrow \text{argmax}_\psi \mathbb{E}_{\theta_t \sim p_t}[\eta(\theta_t, \psi)]$;
5:      Sample one trajectory $\tau_t$ from policy $\pi_{\psi_t}$, and one trajectory $\tau_t'$ from policy $\pi_{\psi_{t-1}}$.
6:      Update $\mathcal{H}_t \leftarrow \mathcal{H}_{t-1} \cup \{(\tau, \tau')\}$

---

The approximate local maximum is defined in the same as in the bandit setting, except that the gradient and Hessian matrix are taken w.r.t to the policy parameter space $\psi$. We also assume realizability ($\theta^\star \in \Theta$) and the Lipschitz assumptions as in Assumption 2.1 (with action $a$ replaced by policy parameter $\psi$).

In the following we present the proof sketch for Theorem 4.4. Compare to the bandit case, we only need to prove an analog of Lemma C.2, which means that we need to upper-bound the error term $\Delta_t$ by the difference of dynamics, as discussed before. Formally speaking, let $\tau_t = (s_1, a_1, \cdots, s_H, a_H)$ be a trajectory sampled from policy $\pi_{\psi_t}$ under the ground-truth dynamics $T_{\theta^\star}$. By telescope lemma (Lemma F.16) we get

$$V_\theta^\psi(s_1) - V_{\theta^\star}^\psi(s_1) = \mathbb{E}_{\tau \sim \rho_{\theta^\star}^\psi}\left[\sum_{h=1}^H \Big(V_\theta^\psi(T_\theta(s_h, a_h)) - V_\theta^\psi(T_{\theta^\star}(s_h, a_h))\Big)\right]. \tag{59}$$

Lipschitz assumption (Assumption 4.1) yields,

$$\left|V_\theta^\psi(T_\theta(s_h, a_h)) - V_\theta^\psi(T_{\theta^\star}(s_h, a_h))\right| \leq L_0 \left\|T_\theta(s_h, a_h) - T_{\theta^\star}(s_h, a_h)\right\|_2. \tag{60}$$

Combining Eq. (59) and Eq. (60) and apply Cauchy-Schwartz inequality gives an upper bound for $[\Delta_t]_1^2$ and $[\Delta_t]_2^2$. As for the gradient term, we will take gradient w.r.t. $\psi$ to both sides of Eq. (59). The gradient inside expectation can be dealt with easily. And the gradient w.r.t. the distribution $\rho_{\theta^\star}^\psi$ can be computed by policy gradient lemma (Lemma F.17). As a result we get

$$\nabla_\psi V_\theta^\psi(s_1) - \nabla_\psi V_{\theta^\star}^\psi(s_1)$$
$$= \mathbb{E}_{\tau \sim \rho_{\theta^\star}^\psi}\left[\left(\sum_{h=1}^H \nabla_\psi \log \pi_\psi(a_h \mid s_h)\right)\left(\sum_{h=1}^H \Big(V_\theta^\psi(T_\theta(s_h, a_h)) - V_\theta^\psi(T_{\theta^\star}(s_h, a_h))\Big)\right)\right]$$
$$+ \mathbb{E}_{\tau \sim \rho_{\theta^\star}^\psi}\left[\sum_{h=1}^H \Big(\nabla_\psi V_\theta^\psi(T_\theta(s_h, a_h)) - \nabla_\psi V_\theta^\psi(T_{\theta^\star}(s_h, a_h))\Big)\right]. \tag{61}$$

The first term can be bounded by vector-form Cauchy-Schwartz and Assumption 4.2, and the second term is bounded by Assumption 4.1. Similarly, this approach can be extended to second order term. As a result, we have the following lemma.

**Lemma D.1.** *Under the setting of Theorem 4.4, we have*

$$c_1 \mathbb{E}_{\tau_{1:t}, \tau'_{1:t}, \theta_{1:t}} \left[ \bar{\Delta}_t^2 \right] \geq \mathbb{E}_{\theta_{1:t}} \left[ \Delta_t^2 \right]. \tag{62}$$

Proof of Lemma D.1 is shown in Appendix D.1. Proof of Theorem 4.4 is exactly the same as that of Theorem 3.1 except for replacing Lemma C.2 with Lemma D.1.

## D.1 Proof of Lemma D.1

*Proof.* The lemma is proven by combining standard telescoping lemma and policy gradient lemma. Specifically, let $\rho_T^\pi$ be the distribution of trajectories generated by policy $\pi$ and dynamics $T$. By telescoping lemma (Lemma F.16) we have,

$$V_{\theta_t}^{\psi_t}(s_1) - V_{\theta^\star}^{\psi_t}(s_1) = \mathbb{E}_{\tau \sim \rho_{\theta^\star}^{\psi_t}} \left[ \sum_{h=1}^H \left( V_{\theta_t}^{\psi_t}(T_{\theta_t}(s_h, a_h)) - V_{\theta_t}^{\psi_t}(T_{\theta^\star}(s_h, a_h)) \right) \right]. \tag{63}$$

By the Lipschitz assumption (Assumption 4.1),

$$\left| V_{\theta_t}^{\psi_t}(T_{\theta_t}(s_h, a_h)) - V_{\theta_t}^{\psi_t}(T_{\theta^\star}(s_h, a_h)) \right| \leq L_0 \left\| T_{\theta_t}(s_h, a_h) - T_{\theta^\star}(s_h, a_h)) \right\|_2. \tag{64}$$

Consequently

$$\Delta_{t,1}^2 = \left( V_{\theta_t}^{\psi_t}(s_0) - V_{\theta^\star}^{\psi_t}(s_0) \right)^2 \leq H L_0^2 \mathbb{E}_{\tau \sim \rho_{\theta^\star}^{\psi_t}} \left[ \sum_{h=1}^H \left\| T_{\theta_t}(s_h, a_h) - T_{\theta^\star}(s_h, a_h)) \right\|_2^2 \right]. \tag{65}$$

Similarly we get,

$$\Delta_{t,2}^2 = \left( V_{\theta_t}^{\psi_{t-1}}(s_0) - V_{\theta^\star}^{\psi_{t-1}}(s_0) \right)^2 \leq H L_0^2 \mathbb{E}_{\tau \sim \rho_{\theta^\star}^{\psi_{t-1}}} \left[ \sum_{h=1}^H \left\| T_{\theta_t}(s_h, a_h) - T_{\theta^\star}(s_h, a_h)) \right\|_2^2 \right]. \tag{66}$$

Now we turn to higher order terms. First of all, by Hölder inequality and Assumption 4.2, we can prove the following:

- $\left\| \mathbb{E}_{\tau \sim \rho_{\theta^\star}^\psi} \left[ \left( \sum_{h=1}^H \nabla_\psi \log \pi_\psi(a_h \mid s_h) \right) \left( \sum_{h=1}^H \nabla_\psi \log \pi_\psi(a_h \mid s_h) \right)^\top \right] \right\|_{\text{sp}} \leq H^2 \chi_g, \forall \psi \in \Psi$;

- $\left\| \mathbb{E}_{\tau \sim \rho_{\theta^\star}^\psi} \left[ \left( \sum_{h=1}^H \nabla_\psi \log \pi_\psi(a_h \mid s_h) \right)^{\otimes 4} \right] \right\|_{\text{sp}} \leq H^4 \chi_f, \forall \psi \in \Psi$;

- $\left\| \mathbb{E}_{\tau \sim \rho_{\theta^\star}^\psi} \left[ \left( \sum_{h=1}^H \nabla_\psi^2 \log \pi_\psi(a_h \mid s_h) \right) \left( \sum_{h=1}^H \nabla_\psi^2 \log \pi_\psi(a \mid s) \right)^\top \right] \right\|_{\text{sp}} \leq H^2 \chi_h, \forall \psi \in \Psi$.

Indeed, consider the first statement. Define $g_h = \nabla_\psi \log \pi_\psi(a_h \mid s_h)$ for shorthand. Then we have

$$\left\| \mathbb{E}_{\tau \sim \rho_{\theta^\star}^\psi} \left[ \left( \sum_{h=1}^H g_h \right) \left( \sum_{h=1}^H g_h \right)^\top \right] \right\|_{\text{sp}} = \sup_{u \in S^{d-1}} u^\top \mathbb{E}_{\tau \sim \rho_{\theta^\star}^\psi} \left[ \left( \sum_{h=1}^H g_h \right) \left( \sum_{h=1}^H g_h \right)^\top \right] u \tag{67}$$

$$= \sup_{u \in S^{d-1}} \mathbb{E}_{\tau \sim \rho_{\theta^\star}^\psi} \left[ \left\langle u, \left( \sum_{h=1}^H g_h \right) \right\rangle^2 \right] \leq \sup_{u \in S^{d-1}} \mathbb{E}_{\tau \sim \rho_{\theta^\star}^\psi} \left[ H \sum_{h=1}^H \langle u, g_h \rangle^2 \right] \tag{68}$$

$$\leq \mathbb{E}_{\tau \sim \rho_{\theta^\star}^\psi} \left[ H \sum_{h=1}^H \sup_{u \in S^{d-1}} \langle u, g_h \rangle^2 \right] = \mathbb{E}_{\tau \sim \rho_{\theta^\star}^\psi} \left[ H \sum_{h=1}^H \left\| g g^\top \right\|_{\text{sp}} \right] \leq H^2 \chi_g. \tag{69}$$

Similarly we can get the second and third statement.

For any fixed $\psi$ and $\theta$ we have

$$V_\theta^\psi(s_1) - V_{\theta^\star}^\psi(s_1) = \mathbb{E}_{\tau \sim \rho_{\theta^\star}^\psi}\left[\sum_{h=1}^H \left(V_\theta^\psi(T_\theta(s_h, a_h)) - V_\theta^\psi(T_{\theta^\star}(s_h, a_h))\right)\right]. \tag{70}$$

Applying policy gradient lemma (namely, Lemma F.17) to RHS of Eq. (70) we get,

$$\nabla_\psi V_\theta^\psi(s_1) - \nabla_\psi V_{\theta^\star}^\psi(s_1)$$
$$= \mathbb{E}_{\tau \sim \rho_{\theta^\star}^\psi}\left[\left(\sum_{h=1}^H \nabla_\psi \log \pi_\psi(a_h \mid s_h)\right)\left(\sum_{h=1}^H \left(V_\theta^\psi(T_\theta(s_h, a_h)) - V_\theta^\psi(T_{\theta^\star}(s_h, a_h))\right)\right)\right]$$
$$+ \mathbb{E}_{\tau \sim \rho_{\theta^\star}^\psi}\left[\sum_{h=1}^H \left(\nabla_\psi V_\theta^\psi(T_\theta(s_h, a_h)) - \nabla_\psi V_\theta^\psi(T_{\theta^\star}(s_h, a_h))\right)\right]. \tag{71}$$

Define the following shorthand:

$$G_\theta^\psi(s, a) = V_\theta^\psi(T_\theta(s, a)) - V_\theta^\psi(T_{\theta^\star}(s, a)), \tag{72}$$

$$f = \sum_{h=1}^H \nabla_\psi \log \pi_\psi(a_h \mid s_h). \tag{73}$$

In the following we also omit the subscription in $\mathbb{E}_{\tau \sim \rho_{\theta^\star}^\psi}$ when the context is clear. It followed by Eq. (71) that

$$\left\|\nabla_\psi V_\theta^\psi(s_1) - \nabla_\psi V_{\theta^\star}^\psi(s_1)\right\|_2^2$$
$$\leq 2\left\|\mathbb{E}\left[f\left(\sum_{h=1}^H G_\theta^\psi(s_h, a_h)\right)\right]\right\|_2^2 + 2\left\|\mathbb{E}\left[\sum_{h=1}^H \nabla_\psi G_\theta^\psi(s_h, a_h)\right]\right\|_2^2$$
$$\leq 2\left\|\mathbb{E}[ff^\top]\right\|_{\mathrm{sp}} \mathbb{E}\left[\left(\sum_{h=1}^H G_\theta^\psi(s_h, a_h)\right)^2\right] + 2\left\|\mathbb{E}\left[\sum_{h=1}^H \nabla_\psi G_\theta^\psi(s_h, a_h)\right]\right\|_2^2 \quad \text{(By Lemma F.7)}$$
$$\leq 2H\left\|\mathbb{E}[ff^\top]\right\|_{\mathrm{sp}} \mathbb{E}\left[\sum_{h=1}^H G_\theta^\psi(s_h, a_h)^2\right] + 2H\mathbb{E}\left[\sum_{h=1}^H \left\|\nabla_\psi G_\theta^\psi(s_h, a_h)\right\|_2^2\right].$$

Now, plugin $\psi = \psi_{t-1}, \theta = \theta_t$ and apply Assumption 4.1 we get

$$\Delta_{t,3}^2 = \left\|\nabla_\psi V_{\theta_t}^{\psi_{t-1}}(s_1) - \nabla_\psi V_{\theta^\star}^{\psi_{t-1}}(s_1)\right\|_2^2$$
$$\leq (2HL_1^2 + 2H^3\chi_g L_0^2)\mathbb{E}_{\tau \sim \rho_{\theta^\star}^{\psi_{t-1}}}\left[\sum_{h=1}^H \|T_{\theta_t}(s_h, a_h) - T_{\theta^\star}(s_h, a_h)\|_2^2\right].$$

For any fixed $\psi, \theta$, define the following shorthand:

$$g = \sum_{h=1}^H \left(V_\theta^\psi(T_\theta(s_h, a_h)) - V_\theta^\psi(T_{\theta^\star}(s_h, a_h))\right). \tag{74}$$

Apply policy gradient lemma again to RHS of Eq. (71) we get

$$\nabla_\psi^2 V_\theta^\psi(s_1) - \nabla_\psi^2 V_{\theta^\star}^\psi(s_1)$$
$$= \mathbb{E}\left[(\nabla_\psi g)f^\top\right] + \mathbb{E}\left[f(\nabla_\psi g)^\top\right] + \mathbb{E}\left[\nabla_\psi^2 g\right] + \mathbb{E}\left[g\left(\sum_{h=1}^H \nabla_\psi^2 \log \pi_\psi(a_h \mid s_h)\right)\right] + \mathbb{E}\left[g(ff^\top)\right].$$

As a result of Lemma F.8 and Lemma F.9 that,

$$\left\|\nabla_\psi^2 V_\theta^\psi(s_1) - \nabla_\psi^2 V_{\theta^\star}^\psi(s_1)\right\|_{\mathrm{sp}}^2$$

$$= 4 \left\| \mathbb{E}\big[(\nabla_\psi g) f^\top\big] + \mathbb{E}\Big[f(\nabla_\psi g)^\top\Big] \right\|_{\mathrm{sp}}^2 + 4 \left\| \mathbb{E}\big[\nabla_\psi^2 g\big]\right\|_{\mathrm{sp}}^2 + 4 \left\| \mathbb{E}\big[g(ff^\top)\big]\right\|_{\mathrm{sp}}^2$$

$$+ 4 \left\| \mathbb{E}\left[g\left(\sum_{h=1}^{H} \nabla_\psi^2 \log \pi_\psi(a_h \mid s_h)\right)\right]\right\|_{\mathrm{sp}}^2$$

$$\leq 8 \sup_{u,v \in S^{d-1}} \mathbb{E}[\langle \nabla_\psi g, u\rangle \langle f, v\rangle]^2 + 4\mathbb{E}\Big[\big\|\nabla_\psi^2 g\big\|_{\mathrm{sp}}^2\Big] + 4\mathbb{E}[g^2]\, \big\|\mathbb{E}[f^{\otimes 4}]\big\|_{\mathrm{sp}}$$

$$+ 4\mathbb{E}[g^2] \left\| \mathbb{E}\left[\left(\sum_{h=1}^{H} \nabla_\psi^2 \log \pi_\psi(a_h \mid s_h)\right)\left(\sum_{h=1}^{H} \nabla_\psi^2 \log \pi_\psi(a_h \mid s_h)\right)^\top\right]\right\|_{\mathrm{sp}}. \quad (75)$$

Note that by Hölder's inequality,

$$\sup_{u,v \in S^{d-1}} \mathbb{E}[\langle \nabla_\psi g, u\rangle \langle f, v\rangle]^2 \leq \sup_{u,v \in S^{d-1}} \mathbb{E}\Big[\langle \nabla_\psi g, u\rangle^2\Big]\mathbb{E}\Big[\langle f, v\rangle^2\Big] \leq \mathbb{E}\Big[\|\nabla_\psi g\|_2^2\Big]\big\|\mathbb{E}[ff^\top]\big\|_{\mathrm{sp}}.$$

By Assumption 4.1 we get,

$$\mathbb{E}[g^2] = \mathbb{E}\left[\left(\sum_{h=1}^{H}\left(V_\theta^\psi(T_\theta(s_h, a_h)) - V_\theta^\psi(T_{\theta^\star}(s_h, a_h))\right)\right)^2\right] \quad (76)$$

$$\leq H\mathbb{E}\left[\sum_{h=1}^{H}\left(V_\theta^\psi(T_\theta(s_h, a_h)) - V_\theta^\psi(T_{\theta^\star}(s_h, a_h))\right)^2\right] \quad (77)$$

$$\leq HL_0^2\mathbb{E}\left[\sum_{h=1}^{H}\|T_\theta(s_h, a_h) - T_{\theta^\star}(s_h, a_h)\|_2^2\right]. \quad (78)$$

Similarly, we have

$$\mathbb{E}[\|\nabla_\psi g\|_2^2] \leq HL_1^2\mathbb{E}\left[\sum_{h=1}^{H}\|T_\theta(s_h, a_h) - T_{\theta^\star}(s_h, a_h)\|_2^2\right], \quad (79)$$

$$\mathbb{E}[\|\nabla_\psi^2 g\|_{\mathrm{sp}}^2] \leq HL_2^2\mathbb{E}\left[\sum_{h=1}^{H}\|T_\theta(s_h, a_h) - T_{\theta^\star}(s_h, a_h)\|_2^2\right]. \quad (80)$$

Combining with Eq. (75) we get,

$$\Delta_{t,4}^2 = \left\|\nabla_\psi^2 V_{\theta_t}^{\psi_t}(s_1) - \nabla_\psi V_{\theta^\star}^{\psi_t}(s_1)\right\|_{\mathrm{sp}}^2$$

$$\leq \big(8H^3 L_1^2 \chi_g + 4HL_2^2 + 4L_0^2(H^3\chi_h + H^5\chi_f)\big)\mathbb{E}_{\tau \sim \rho_{\theta^\star}^{\psi_{t-1}}}\left[\sum_{h=1}^{H}\|T_{\theta_t}(s_h, a_h) - T_{\theta^\star}(s_h, a_h)\|_2^2\right].$$

By noting that $\Delta_t^2 = \sum_{i=1}^{4}\Delta_{i,t}^2$, we get the desired upper bound. $\qquad \square$

## E   Analysis of Example 4.3

Recall that our RL instance is given as follows:

$$T(s, a) = \mathrm{N}_\theta(s + a), \quad (81)$$

$$\pi_\psi(s) = \mathcal{N}(\psi s, \sigma^2 I). \quad (82)$$

And the assumptions are listed below.

- Lipschitzness of reward function: $|r(s_1, a_1) - r(s_2, a_2)| \leq L_r(\|s_1 - s_2\|_2 + \|a_1 - a_2\|_2)$.
- Bounded Parameter: we assume $\|\psi\|_{\mathrm{op}} \leq \mathcal{O}(1)$.

In the sequel we verify the assumptions of Theorem 4.4.

### E.1 Verifying Assumption 4.2.

**Verifying item 1.** Recall that $\psi \in \mathbb{R}^{d \times d}$. By algebraic manipulation, for all $s, a$ we get,

$$\nabla_\psi \log \pi_\psi(a \mid s) = \frac{1}{\sigma^2} \mathrm{vec}((a - \psi s) \otimes s) \tag{83}$$

where $\mathrm{vec}(x)$ denotes the vectorization of tensor $x$. Define random variable $u = a - \psi s$. By the definition of policy $\pi_\psi(s)$ we have $u \sim \mathcal{N}(0, \sigma^2 I)$. As a result,

$$\|\mathbb{E}_{a \sim \pi_\psi(\cdot \mid s)}[(\nabla_\psi \log \pi_\psi(a \mid s))(\nabla_\psi \log \pi_\psi(a \mid s))^\top]\|_{\mathrm{sp}} \tag{84}$$

$$= \frac{1}{\sigma^4} \sup_{v \in S^{d \times d - 1}} \mathbb{E}_{u \sim \mathcal{N}(0, \sigma^2 I)} \left[ \langle v, \mathrm{vec}(u \otimes s) \rangle^2 \right]. \tag{85}$$

Note that $\langle v, \mathrm{vec}(u \otimes s) \rangle = \sum_{1 \leq i, j \leq d} [u]_i [s]_j [v]_{i,j}$. Because $u$ is isotropic, $[u]_i$ are independent random variables where $[u]_i \sim \mathcal{N}(0, \sigma^2)$. Therefore $\langle v, \mathrm{vec}(u \otimes s) \rangle \sim \mathcal{N}\left(0, \sigma^2 \sum_{i=1}^d \left( \sum_{j=1}^d s_j v_{i,j} \right)^2 \right)$. Combining with Eq. (85) we get,

$$\mathbb{E}_{u \sim \mathcal{N}(0, \sigma^2 I)} \left[ \langle v, \mathrm{vec}(u \otimes s) \rangle^2 \right] = \sigma^2 \sum_{i=1}^d \left( \sum_{j=1}^d s_j v_{i,j} \right)^2 \tag{86}$$

$$\leq \sigma^2 \sum_{i=1}^d \left( \sum_{j=1}^d s_j^2 \right) \left( \sum_{j=1}^d v_{i,j}^2 \right) \leq \|s\|_2^2 \|v\|_2^2. \tag{87}$$

Consequently we have

$$\|\mathbb{E}_{a \sim \pi_\psi(\cdot \mid s)}[(\nabla_\psi \log \pi_\psi(a \mid s))(\nabla_\psi \log \pi_\psi(a \mid s))^\top]\|_{\mathrm{sp}} \leq \frac{1}{\sigma^2} \triangleq \chi_g. \tag{88}$$

**Verifying item 2.** Similarly, using the equation where $\mathbb{E}_{x \sim \mathcal{N}(0, \sigma^2)}[x^4] = 3\sigma^4$ we have

$$\|\mathbb{E}_{a \sim \pi_\psi(\cdot \mid s)}[(\nabla_\psi \log \pi_\psi(a \mid s))^{\otimes 4}]\|_{\mathrm{sp}} = \frac{1}{\sigma^8} \sup_{v \in S^{d \times d - 1}} \mathbb{E}_{u \sim \mathcal{N}(0, \sigma^2 I)} \left[ \langle v, \mathrm{vec}(u \otimes s) \rangle^4 \right]. \tag{89}$$

$$\leq \frac{3}{\sigma^8} \left( \sigma^2 \sum_{i=1}^d \left( \sum_{j=1}^d s_j v_{i,j} \right)^2 \right)^2 \leq \frac{3}{\sigma^4} \|s\|_2^4 \|v\|_2^4 \leq \frac{3}{\sigma^4} \triangleq \chi_f. \tag{90}$$

**Verifying item 3.** Since $\nabla_\psi^2 \log \pi_\psi(a \mid s)$ is PSD, we have

$$\|\mathbb{E}_{a \sim \pi_\psi(\cdot \mid s)}[(\nabla_\psi^2 \log \pi_\psi(a \mid s))(\nabla_\psi^2 \log \pi_\psi(a \mid s))^\top]\|_{\mathrm{sp}} \tag{91}$$

$$= \sup_v \mathbb{E}\left[ v^\top (\nabla_\psi^2 \log \pi_\psi(a \mid s))(\nabla_\psi^2 \log \pi_\psi(a \mid s))^\top v \right] \tag{92}$$

$$= \sup_v \mathbb{E}\left[ \|(\nabla_\psi^2 \log \pi_\psi(a \mid s))^\top v\|_2^2 \right] = \sup_v \mathbb{E}\left[ \left( v^\top (\nabla_\psi^2 \log \pi_\psi(a \mid s)) v \right)^4 \right]. \tag{93}$$

By algebraic manipulation, for all $s, a \in \mathbb{R}^d$ and $v \in \mathbb{R}^{d \times d}$ we have

$$v^\top (\nabla_\psi^2 \log \pi_\psi(a \mid s)) v = - \sum_{i=1}^d \left( \sum_{j=1}^d v_{i,j} s_j \right)^2. \tag{94}$$

Consequently,

$$\left( v^\top (\nabla_\psi^2 \log \pi_\psi(a \mid s)) v \right)^4 \leq \|s\|_2^4 \|v\|_2^4 \leq 1 \triangleq \chi_h. \tag{95}$$

### E.2 Verifying Assumption 2.1.

**Verifying item 1.** We verify Assumption 2.1 by applying policy gradient lemma. Recall that

$$\eta(\theta, \psi) = \mathbb{E}_{\tau \sim \rho_\theta^\psi} \left[ \sum_{h=1}^{H} r(s_h, a_h) \right]. \tag{96}$$

By policy gradient lemma (Lemma F.17) we have

$$\nabla_\psi \eta(\theta, \psi) = \mathbb{E}_{\tau \sim \rho_\theta^\psi} \left[ \left( \sum_{h=1}^{H} \nabla_\psi \log \pi_\psi(a_h \mid s_h) \right) \left( \sum_{h=1}^{H} r(s_h, a_h) \right) \right]. \tag{97}$$

By Eq. (83), condition on $s_h$ we get

$$\nabla_\psi \log \pi_\psi(a_h \mid s_h) = \frac{1}{\sigma^2} \text{vec}(u \otimes s_h) \tag{98}$$

where $u = a_h - \psi s_h \sim \mathcal{N}(0, \sigma^2 I)$. Define the shorthand $g = \sum_{h=1}^{H} r(s_h, a_h)$. Note that by Hölder inequality,

$$\|\mathbb{E}[\nabla_\psi \log \pi_\psi(a_h \mid s_h) g]\|_2^2 = \sup_{v \in \mathbb{R}^{d \times d}, \|v\|_2 = 1} \mathbb{E}[\langle \nabla_\psi \log \pi_\psi(a_h \mid s_h), v \rangle g]^2 \tag{99}$$

$$\leq \sup_{v \in \mathbb{R}^{d \times d}, \|v\|_2 = 1} \mathbb{E}\left[ \langle \nabla_\psi \log \pi_\psi(a_h \mid s_h), v \rangle^2 \right] \mathbb{E}[g^2]. \tag{100}$$

Since $v \in \mathbb{R}^{d \times d}$, if we view $v$ as a $d \times d$ matrix then $\langle \nabla_\psi \log \pi_\psi(a_h \mid s_h), v \rangle = \frac{1}{\sigma^2} \langle v s_h, u \rangle$. Because $u$ is an isotropic Gaussian random vector, $\langle v s_h, u \rangle \sim \mathcal{N}(0, \sigma^2 \|v s_h\|_2^2)$. Consequently,

$$\mathbb{E}\left[ \langle \nabla_\psi \log \pi_\psi(a_h \mid s_h), v \rangle^2 \right] = \frac{1}{\sigma^2} \|v s_h\|_2^2 \leq \frac{1}{\sigma^2} \|v\|_F^2 \|s_h\|_2^2 \leq \frac{1}{\sigma^2}. \tag{101}$$

It follows that $\|\mathbb{E}[\nabla_\psi \log \pi_\psi(a_h \mid s_h) g]\|_2^2 \leq \frac{H^2}{\sigma^2}$. By triangular inequality and Eq. (97) we get

$$\|\nabla_\psi \eta(\theta, \psi)\|_2 \leq H^2/\sigma. \tag{102}$$

**Verifying item 2.** Define the shorthand $f = \sum_{h=1}^{H} \nabla_\psi \log \pi_\psi(a_h \mid s_h)$. Use policy gradient lemma on Eq. (97) again we get, for any $v, w \in \mathbb{R}^{d \times d}$,

$$v^\top \nabla_\psi^2 \eta(\theta, \psi) w = \mathbb{E}_{\tau \sim \rho_\theta^\psi} \left[ \langle f, v \rangle \langle f, w \rangle g + \left( \sum_{h=1}^{H} v^\top \nabla_\psi^2 \log \pi_\psi(a_h \mid s_h) w \right) g \right]. \tag{103}$$

For the first term inside the expectation, we bound it by using Hölder inequality twice. Specifically, for any $h, h' \in [H]$ we have

$$\mathbb{E}[\langle \nabla_\psi \log \pi_\psi(a_h \mid s_h), v \rangle \langle \nabla_\psi \log \pi_\psi(a_{h'} \mid s_{h'}), w \rangle g] \tag{104}$$

$$\leq \mathbb{E}\left[ \langle \nabla_\psi \log \pi_\psi(a_h \mid s_h), v \rangle^4 \right]^{1/4} \mathbb{E}\left[ \langle \nabla_\psi \log \pi_\psi(a_{h'} \mid s_{h'}), w \rangle^4 \right]^{1/4} \mathbb{E}[g^2]^{1/2}. \tag{105}$$

Similarly, $\langle \nabla_\psi \log \pi_\psi(a_h \mid s_h), v \rangle \sim \frac{1}{\sigma^2} \mathcal{N}(0, \sigma^2 \|v s_h\|_2^2)$ and $\langle \nabla_\psi \log \pi_\psi(a_{h'} \mid s_{h'}), w \rangle \sim \frac{1}{\sigma^2} \mathcal{N}(0, \sigma^2 \|w s_{h'}\|_2^2)$. As a result,

$$\mathbb{E}[\langle \nabla_\psi \log \pi_\psi(a_h \mid s_h), v \rangle \langle \nabla_\psi \log \pi_\psi(a_{h'} \mid s_{h'}), w \rangle g] \tag{106}$$

$$\leq 3 \frac{1}{\sigma^2} \|v\|_2 \|s_h\|_2 \|w\|_2 \|s_{h'}\|_2 H \leq \frac{3H}{\sigma^2}. \tag{107}$$

Therefore the first term of Eq. (103) can be bounded by $\frac{3H^3}{\sigma^2}$. Now we bound the second term of Eq. (103). By algebraic manipulation we have

$$v^\top \nabla_\psi^2 \log \pi_\psi(a_h \mid s_h) w = -\frac{1}{\sigma^2} \langle w s_h, v s_h \rangle. \tag{108}$$

Consequently,

$$\mathbb{E}\left[ \left( \sum_{h=1}^{H} v^\top \nabla_\psi^2 \log \pi_\psi(a_h \mid s_h) w \right) g \right] \leq \frac{H^2}{\sigma^2} \|w\|_2 \|v\|_2 \leq \frac{H^2}{\sigma^2}. \tag{109}$$

In summary, we have $\left\| \nabla_\psi^2 \eta(\theta, \psi) \right\|_{\text{op}} \leq \frac{4H^3}{\sigma^2}$.

**Verifying item 3.** Now we turn to the last item in Assumption 2.1. First of all, following Eq. (108), we have $\nabla_\psi^3 \log \pi_\psi(a_h \mid s_h) = 0$. As a result, applying policy gradient lemma to Eq. (103) again we get

$$\left\langle \nabla_\psi^3 \eta(\theta, \psi), v \otimes w \otimes x \right\rangle = \mathbb{E}_{\tau \sim \rho_\theta^\psi} \left[ \langle f, v \rangle \langle f, w \rangle \langle f, w \rangle g \right] \tag{110}$$

$$+ \mathbb{E}_{\tau \sim \rho_\theta^\psi} \left[ \langle f, x \rangle \left( \sum_{h=1}^H v^\top \nabla_\psi^2 \log \pi_\psi(a_h \mid s_h) w \right) g \right] \tag{111}$$

$$+ \mathbb{E}_{\tau \sim \rho_\theta^\psi} \left[ \langle f, v \rangle \left( \sum_{h=1}^H x^\top \nabla_\psi^2 \log \pi_\psi(a_h \mid s_h) w \right) g \right] \tag{112}$$

$$+ \mathbb{E}_{\tau \sim \rho_\theta^\psi} \left[ \langle f, w \rangle \left( \sum_{h=1}^H v^\top \nabla_\psi^2 \log \pi_\psi(a_h \mid s_h) x \right) g \right]. \tag{113}$$

Following the same argument, by Hölder inequality, for any $h_1, h_2, h_3 \in [H]$ we have

$$\mathbb{E}[\langle \nabla_\psi \log \pi_\psi(a_h \mid s_h), v \rangle \langle \nabla_\psi \log \pi_\psi(a_{h'} \mid s_{h'}), w \rangle \langle \nabla_\psi \log \pi_\psi(a_{h'} \mid s_{h'}), x \rangle g]$$

$$\leq \mathbb{E} \left[ \langle \nabla_\psi \log \pi_\psi(a_h \mid s_h), v \rangle^6 \right]^{1/6} \mathbb{E} \left[ \langle \nabla_\psi \log \pi_\psi(a_{h'} \mid s_{h'}), w \rangle^6 \right]^{1/6} \mathbb{E} \left[ \langle \nabla_\psi \log \pi_\psi(a_{h'} \mid s_{h'}), x \rangle^6 \right]^{1/6} H$$

$$\leq \frac{\sqrt{15} H}{\sigma^3}.$$

On the other hand,

$$\mathbb{E} \left[ \langle f, x \rangle \left( \sum_{h=1}^H v^\top \nabla_\psi^2 \log \pi_\psi(a_h \mid s_h) w \right) g \right]$$

$$\leq \mathbb{E} \left[ \langle f, x \rangle^2 \right]^{1/2} \mathbb{E} \left[ \left( \left( \sum_{h=1}^H v^\top \nabla_\psi^2 \log \pi_\psi(a_h \mid s_h) w \right) g \right)^2 \right]^{1/2}$$

$$\leq \frac{H^3}{\sigma^3}.$$

By symmetricity, Eq. (110) can be upper bounded by

$$\left\langle \nabla_\psi^3 \eta(\theta, \psi), v \otimes w \otimes x \right\rangle \leq \frac{7 H^4}{\sigma^3}. \tag{114}$$

### E.3 Verifying Assumption 4.1.

**Verifying item 1.** We verify Assumption 4.1 by coupling argument. First of all, consider the Lipschitzness of value function. By Bellman equation we have

$$V_\theta^\psi(s) = \mathbb{E}_{a \sim \pi_\psi(s)} \left[ r(s, a) + V_\theta^\psi(T(s, a)) \right] \tag{115}$$

$$= \mathbb{E}_{u \sim \mathcal{N}(0, \sigma^2 I)} \left[ r(s, \psi s + u) + V_\theta^\psi(N_\theta(s + \psi s + u)) \right]. \tag{116}$$

Define $B = 1 + \|\psi\|_{\text{op}}$ for shorthand. For two states $s_1, s_2 \in \mathcal{S}$, by the Lipschitz assumption on reward function we have

$$|r(s_1, \psi s_1 + u) - r(s_2, \psi s_2 + u)| \leq L_r B \|s_1 - s_2\|_2. \tag{117}$$

Then consider the second term in Eq. (116). Since we have $|V_\theta^\pi| \leq H$ and

$$\text{TV}\left( \mathcal{N}(s_1 + \psi s_1, \sigma^2 I), \mathcal{N}(s_2 + \psi s_2, \sigma^2 I) \right) \leq \frac{1}{2\sigma} \|s_1 + \psi s_1 - s_2 - \psi s_2\|_2 \leq \frac{B \|s_1 - s_2\|_2}{2\sigma},$$

it follows that

$$\left| \mathbb{E}_{u \sim \mathcal{N}(0, \sigma^2 I)} \left[ V_\theta^\psi(N_\theta(s_1 + \psi s_1 + u)) \right] - \mathbb{E}_{u \sim \mathcal{N}(0, \sigma^2 I)} \left[ V_\theta^\psi(N_\theta(s_2 + \psi s_2 + u)) \right] \right| \leq \frac{HB}{2\sigma} \|s_1 - s_2\|_2.$$

As a result, item 1 of Assumption 4.1 holds as follows

$$\left| V_\theta^\psi(s_1) - V_\theta^\psi(s_2) \right| \leq \left( \frac{HB}{2\sigma} + L_r B \right) \|s_1 - s_2\|_2. \tag{118}$$

**Verifying item 2.** Now we turn to verifying the Lipschitzness of gradient term. Recall that by policy gradient lemma we have for every $v \in \mathbb{R}^{d \times d}$,

$$\left\langle \nabla_\psi V_\theta^\psi(s), v \right\rangle \tag{119}$$

$$= \mathbb{E}_{a \sim \pi_\psi(s)} \left[ \left\langle \nabla_\psi V_\theta^\psi(\mathrm{N}_\theta(s+a)), v \right\rangle \right] \tag{120}$$

$$+ \mathbb{E}_{a \sim \pi_\psi(s)} \left[ \langle \nabla_\psi \log \pi_\psi(a \mid s), v \rangle \left( r(s,a) + V_\theta^\psi(\mathrm{N}_\theta(s+a)) \right) \right] \tag{121}$$

$$= \mathbb{E}_{u \sim \mathcal{N}(0,\sigma^2 I)} \left[ \left\langle \nabla_\psi V_\theta^\psi(\mathrm{N}_\theta(s+\psi s+u)), v \right\rangle \right] \tag{122}$$

$$+ \mathbb{E}_{u \sim \mathcal{N}(0,\sigma^2 I)} \left[ \langle \nabla_\psi \log \pi_\psi(\psi s + u \mid s), v \rangle \left( r(s, \psi s + u) + V_\theta^\psi(\mathrm{N}_\theta(s+\psi s+u)) \right) \right]. \tag{123}$$

Because for any two vectors $g_1, g_2 \in \mathbb{R}^{d \times d}$ $\|g_1 - g_2\|_2 = \sup_{v \in S^{d \times d-1}} \langle g_1 - g_2, v \rangle$, Lipschitzness of Eq. (119) for every $v \in \mathbb{R}^{d \times d}, \|v\|_2 = 1$ implies Lipschitzness of $\nabla_\psi V_\theta^\psi(s)$.

By the boundness of $\left\| \nabla_\psi V_\theta^\psi(s) \right\|_2$ (specifically, item 1 of Assumption 2.1), we have

$$\left| \mathbb{E}_{u \sim \mathcal{N}(0,\sigma^2 I)} \left[ \left\langle \nabla_\psi V_\theta^\psi(\mathrm{N}_\theta(s_1+\psi s_1+u)), v \right\rangle \right] - \mathbb{E}_{u \sim \mathcal{N}(0,\sigma^2 I)} \left[ \left\langle \nabla_\psi V_\theta^\psi(\mathrm{N}_\theta(s_2+\psi s_2+u)), v \right\rangle \right] \right|$$

$$\leq \frac{H^2}{\sigma} \mathrm{TV}\left( \mathcal{N}(s_1 + \psi s_1, \sigma^2 I), \mathcal{N}(s_2 + \psi s_2, \sigma^2 I) \right) \leq \frac{H^2}{\sigma} \frac{B}{2\sigma} \|s_1 - s_2\|_2.$$

For the reward term in Eq. (123), recalling $v \in \mathbb{R}^{d \times d}$ we have

$$\mathbb{E}_{a \sim \pi_\psi(s)}[\langle \nabla_\psi \log \pi_\psi(\psi s + u \mid s), v \rangle r(s, \psi s + u)] = \mathbb{E}_{u \sim \mathcal{N}(0,\sigma^2 I)}[\langle vs, u \rangle r(s, \psi s + u)].$$

Note that

$$\mathbb{E}_{u \sim \mathcal{N}(0,\sigma^2 I)}[\langle vs_1, u \rangle r(s_1, \psi s_1 + u)] - \mathbb{E}_{u \sim \mathcal{N}(0,\sigma^2 I)}[\langle vs_2, u \rangle r(s_2, \psi s_2 + u)] \tag{124}$$

$$= \mathbb{E}_{u \sim \mathcal{N}(0,\sigma^2 I)}[\langle vs_1, u \rangle (r(s_1, \psi s_1 + u) - r(s_2, \psi s_2 + u))] \tag{125}$$

$$+ \mathbb{E}_{u \sim \mathcal{N}(0,\sigma^2 I)}[(\langle vs_1, u \rangle - \langle vs_2, u \rangle)r(s_2, \psi s_2 + u)]. \tag{126}$$

Note that $u$ is isotropic. Applying Lemma F.10 we have

$$\mathbb{E}_{u \sim \mathcal{N}(0,\sigma^2 I)}[(\langle vs_1, u \rangle - \langle vs_2, u \rangle)r(s_2, \psi s_2 + u)] \leq \sigma \|vs_1 - vs_2\|_2 \leq \sigma \|s_1 - s_2\|_2. \tag{127}$$

We can also bound the term in Eq. (125) by

$$\mathbb{E}_{u \sim \mathcal{N}(0,\sigma^2 I)}[\langle vs_1, u \rangle (r(s_1, \psi s_1 + u) - r(s_2, \psi s_2 + u))] \tag{128}$$

$$\leq \mathbb{E}_{u \sim \mathcal{N}(0,\sigma^2 I)} \left[ \langle vs_1, u \rangle^2 \right]^{1/2} \mathbb{E}_{u \sim \mathcal{N}(0,\sigma^2 I)} \left[ (r(s_1, \psi s_1 + u) - r(s_2, \psi s_2 + u))^2 \right]^{1/2} \tag{129}$$

$$\leq \sigma \mathbb{E}_{u \sim \mathcal{N}(0,\sigma^2 I)} \left[ L_r^2 B^2 \|s_1 - s_2\|_2^2 \right]^{1/2} \leq \sigma L_r B \|s_1 - s_2\|_2. \tag{130}$$

Now we deal with the last term in Eq. (123). Let $f(s, u) = V_\theta^\psi(\mathrm{N}_\theta(s + \psi s + u))$ for shorthand. Similarly we have

$$\mathbb{E}_{u \sim \mathcal{N}(0,\sigma^2 I)} \left[ \langle \nabla_\psi \log \pi_\psi(\psi s + u \mid s), v \rangle V_\theta^\psi(\mathrm{N}_\theta(s + \psi s + u)) \right] = \mathbb{E}_{u \sim \mathcal{N}(0,\sigma^2 I)}[\langle vs, u \rangle f(s, u)]. \tag{131}$$

By the same telescope sum we get,

$$\mathbb{E}_{u \sim \mathcal{N}(0,\sigma^2 I)}[\langle vs_1, u \rangle f(s_1, u)] - \mathbb{E}_{u \sim \mathcal{N}(0,\sigma^2 I)}[\langle vs_2, u \rangle f(s_2, u)] \tag{132}$$

$$= \mathbb{E}_{u \sim \mathcal{N}(0,\sigma^2 I)}[\langle vs_1, u \rangle (f(s_2, u) - f(s_2, u))] \tag{133}$$

$$+ \mathbb{E}_{u \sim \mathcal{N}(0,\sigma^2 I)}[(\langle vs_1, u \rangle - \langle vs_2, u \rangle)f(s_2, u)]. \tag{134}$$

Applying Lemma F.10 we have

$$\mathbb{E}_{u \sim \mathcal{N}(0,\sigma^2 I)}[(\langle vs_1, u \rangle - \langle vs_2, u \rangle)f(s_2, u)] \leq \sigma H \|vs_1 - vs_2\|_2 \leq \sigma H \|s_1 - s_2\|_2. \tag{135}$$

Applying Lemma F.12 we have

$$\mathbb{E}_{u \sim \mathcal{N}(0,\sigma^2 I)}[\langle vs_1, u \rangle (f(s_2, u) - f(s_2, u))] \leq 6BH \|s_1 - s_2\|_2 \left( 1 + \frac{1}{\sigma} \right). \tag{136}$$

In summary, we have

$$\left\| \nabla_\psi V_\theta^\psi(s_1) - \nabla_\psi V_\theta^\psi(s_2) \right\|_2 \leq \mathrm{poly}(H, B, \sigma, 1/\sigma, L_r) \|s_1 - s_2\|_2. \tag{137}$$

**Verifying item 3.** Lastly, we verify the Lipschitzness of Hessian term. Applying policy gradient lemma to Eq. (119) again we have

$$w^\top \nabla_\psi^2 V_\theta^\psi(s) v \tag{138}$$

$$= \mathbb{E}_{a \sim \pi_\psi(s)} \left[ w^\top \nabla_\psi^2 V_\theta^\psi(\mathrm{N}_\theta(s+a)) v \right] \tag{139}$$

$$+ \mathbb{E}_{a \sim \pi_\psi(s)} \left[ \left\langle \nabla_\psi V_\theta^\psi(\mathrm{N}_\theta(s+a)), v \right\rangle \left\langle \nabla_\psi \log \pi_\psi(a \mid s), w \right\rangle \right] \tag{140}$$

$$+ \mathbb{E}_{a \sim \pi_\psi(s)} \left[ \left\langle \nabla_\psi \log \pi_\psi(a \mid s), v \right\rangle \left\langle \nabla_\psi V_\theta^\psi(\mathrm{N}_\theta(s+a)), w \right\rangle \right] \tag{141}$$

$$+ \mathbb{E}_{a \sim \pi_\psi(s)} \left[ \left\langle \nabla_\psi \log \pi_\psi(a \mid s), v \right\rangle \left\langle \nabla_\psi \log \pi_\psi(a \mid s), w \right\rangle \left( r(s,a) + V_\theta^\psi(\mathrm{N}_\theta(s+a)) \right) \right]. \tag{142}$$

Recall that $a \sim \psi s + \mathcal{N}(0, \sigma^2 I)$. In the sequel, we bound the Lipschitzness of above four terms separately.

By the upper bound of $\left\| \nabla_\psi^2 V_\theta^\psi(\mathrm{N}_\theta(s+a)) \right\|_{\mathrm{op}}$ (specifically, item 2 of Assumption 2.1) we have

$$\left| \mathbb{E}_{u \sim \mathcal{N}(0, \sigma^2 I)} \left[ w^\top \nabla_\psi^2 V_\theta^\psi(\mathrm{N}_\theta(s_1 + \psi s_1 + u)) v \right] - \mathbb{E}_{u \sim \mathcal{N}(0, \sigma^2 I)} \left[ w^\top \nabla_\psi^2 V_\theta^\psi(\mathrm{N}_\theta(s_1 + \psi s_1 + u)) v \right] \right|$$

$$\leq \frac{4H^3}{\sigma^2} \mathrm{TV}\left( \mathcal{N}(s_1 + \psi s_1, \sigma^2 I), \mathcal{N}(s_2 + \psi s_2, \sigma^2 I) \right) \leq \frac{3H^3}{\sigma^2} \frac{B}{2\sigma} \|s_1 - s_2\|_2 .$$

For the terms in Eq. (140), let $f(s, u) = \left\langle \nabla_\psi V_\theta^\psi(\mathrm{N}_\theta(s + \psi s + a)), v \right\rangle$. Repeat the same argument when verifying item 2 again, we have

$$\mathbb{E}_{u \sim \mathcal{N}(0, \sigma^2 I)}[\langle w s_1, u \rangle (f(s_2, u) - f(s_2, u))] \leq 6B \frac{H^2}{\sigma} \|s_1 - s_2\|_2 \left( 1 + \frac{1}{\sigma} \right). \tag{143}$$

Similarly, term in Eq. (141) also has the same Lipschitz constant.

Finally, we bound the term in Eq. (142). For the reward term in Eq. (142), recalling $v, w \in \mathbb{R}^{d \times d}$ we have

$$\mathbb{E}_{a \sim \pi_\psi(s)}[\langle \nabla_\psi \log \pi_\psi(a \mid s), v \rangle \langle \nabla_\psi \log \pi_\psi(a \mid s), w \rangle r(s, a)] = \mathbb{E}_{u \sim \mathcal{N}(0, \sigma^2 I)}[\langle vs, u \rangle \langle ws, u \rangle r(s, \psi s + u)].$$

Note that

$$\mathbb{E}_{u \sim \mathcal{N}(0, \sigma^2 I)}[\langle v s_1, u \rangle \langle w s_1, u \rangle r(s_1, \psi s_1 + u)] - \mathbb{E}_{u \sim \mathcal{N}(0, \sigma^2 I)}[\langle v s_2, u \rangle \langle w s_2, u \rangle r(s_2, \psi s_2 + u)] \tag{144}$$

$$= \mathbb{E}_{u \sim \mathcal{N}(0, \sigma^2 I)}[\langle v s_1, u \rangle \langle w s_1, u \rangle (r(s_1, \psi s_1 + u) - r(s_2, \psi s_2 + u))] \tag{145}$$

$$+ \mathbb{E}_{u \sim \mathcal{N}(0, \sigma^2 I)}[(\langle v s_1, u \rangle \langle w s_1, u \rangle - \langle v s_2, u \rangle \langle w s_2, u \rangle) r(s_2, \psi s_2 + u)]. \tag{146}$$

Note that $u$ is isotropic. Applying Lemma F.15 we have

$$\mathbb{E}_{u \sim \mathcal{N}(0, \sigma^2 I)}[(\langle v s_1, u \rangle \langle w s_1, u \rangle - \langle v s_2, u \rangle \langle w s_2, u \rangle) r(s_2, \psi s_2 + u)]$$
$$\leq \sqrt{3} \sigma^2 (\|v s_1 - v s_2\|_2 + \|w s_1 - w s_2\|_2) \leq 2\sqrt{3} \sigma^2 \|s_1 - s_2\|_2 ,$$

We can also bound the term in Eq. (145) by

$$\mathbb{E}_{u \sim \mathcal{N}(0, \sigma^2 I)}[\langle v s_1, u \rangle \langle w s_1, u \rangle (r(s_1, \psi s_1 + u) - r(s_2, \psi s_2 + u))]$$

$$\leq \mathbb{E}_{u \sim \mathcal{N}(0, \sigma^2 I)} \left[ \langle v s_1, u \rangle^4 \right]^{1/4} \mathbb{E}_{u \sim \mathcal{N}(0, \sigma^2 I)} \left[ \langle w s_1, u \rangle^4 \right]^{1/4} \mathbb{E}_{u \sim \mathcal{N}(0, \sigma^2 I)} \left[ (r(s_1, \psi s_1 + u) - r(s_2, \psi s_2 + u))^2 \right]^{1/2}$$

$$\leq \sqrt{3} \sigma^2 \mathbb{E}_{u \sim \mathcal{N}(0, \sigma^2 I)} \left[ L_r^2 B^2 \|s_1 - s_2\|_2^2 \right]^{1/2} \leq \sqrt{3} \sigma^2 L_r B \|s_1 - s_2\|_2 .$$

Now we deal with the last term in Eq. (142). Let $f(s, u) = V_\theta^\psi(\mathrm{N}_\theta(s + \psi s + u))$ for shorthand. Similarly we have

$$\mathbb{E}_{a \sim \pi_\psi(s)} \left[ \langle \nabla_\psi \log \pi_\psi(a \mid s), v \rangle \langle \nabla_\psi \log \pi_\psi(a \mid s), w \rangle V_\theta^\psi(\mathrm{N}_\theta(s+a)) \right] \tag{147}$$

$$= \mathbb{E}_{u \sim \mathcal{N}(0,\sigma^2 I)}[\langle vs, u \rangle \langle ws, u \rangle f(s, u)]. \tag{148}$$

By the same telescope sum we get,

$$\mathbb{E}_{u \sim \mathcal{N}(0,\sigma^2 I)}[\langle vs_1, u \rangle \langle ws_1, u \rangle f(s_1, u)] - \mathbb{E}_{u \sim \mathcal{N}(0,\sigma^2 I)}[\langle vs_2, u \rangle \langle ws_2, u \rangle f(s_2, u)] \tag{149}$$

$$= \mathbb{E}_{u \sim \mathcal{N}(0,\sigma^2 I)}[\langle vs_1, u \rangle \langle ws_1, u \rangle (f(s_1, u) - f(s_2, u))] \tag{150}$$

$$+ \mathbb{E}_{u \sim \mathcal{N}(0,\sigma^2 I)}[(\langle vs_1, u \rangle \langle ws_1, u \rangle - \langle vs_2, u \rangle \langle ws_2, u \rangle) f(s_2, u)]. \tag{151}$$

Applying Lemma F.15 we have

$$\mathbb{E}_{u \sim \mathcal{N}(0,\sigma^2 I)}[(\langle vs_1, u \rangle \langle ws_1, u \rangle - \langle vs_2, u \rangle \langle ws_2, u \rangle) f(s_2, u)] \tag{152}$$

$$\leq \sqrt{3}\sigma^2 H(\|vs_1 - vs_2\|_2 + \|ws_1 - ws_2\|_2) \leq 2\sqrt{3}\sigma^2 H \|s_1 - s_2\|_2. \tag{153}$$

Applying Lemma F.13 we have

$$\mathbb{E}_{u \sim \mathcal{N}(0,\sigma^2 I)}[\langle vs_1, u \rangle \langle ws_1, u \rangle (f(s_2, u) - f(s_2, u))] \leq \mathrm{poly}(H, \sigma, 1/\sigma) B \|s_1 - s_2\|_2. \tag{154}$$

In summary, we have

$$\left\| \nabla^2_\psi V^\psi_\theta(s_1) - \nabla^2_\psi V^\psi_\theta(s_2) \right\|_{\mathrm{op}} \leq \mathrm{poly}(H, B, \sigma, 1/\sigma, L_r) \|s_1 - s_2\|_2. \tag{155}$$

# F   Helper Lemmas

In this section, we list helper lemmas that are used in previous sections.

## F.1   Helper Lemmas on Probability Analysis

The following lemma provides a concentration inequality on the norm of linear transformation of a Gaussian vector, which is used to prove Lemma F.3.

**Lemma F.1** (Theorem 1 of Hsu et al. [36]). *For $v \sim \mathcal{N}(0, I)$ be a $n$ dimensional Gaussian vector, and $A \in \mathbb{R}^{n \times n}$. Let $\Sigma = A^\top A$, then*

$$\forall t > 0, \Pr\left[\|Av\|_2^2 \geq \mathrm{Tr}(\Sigma) + 2\sqrt{\mathrm{Tr}(\Sigma^2)t} + 2\|\Sigma\|_{\mathrm{op}} t\right] \leq \exp(-t). \tag{156}$$

**Corollary F.2.** *Under the same settings of Lemma F.1,*

$$\forall t > 1, \Pr\left[\|Av\|_2^2 \geq \|A\|_{\mathrm{F}}^2 + 4\|A\|_{\mathrm{F}}^2 t\right] \leq \exp(-t). \tag{157}$$

*Proof.* Let $\lambda_i$ be the $i$-th eigenvalue of $\Sigma$. By the definition of $\Sigma$ we have $\lambda_i \geq 0$. Then we have

$$\mathrm{Tr}(\Sigma) = \sum_{i=1}^n \lambda_i = \|A\|_{\mathrm{F}}^2,$$

$$\mathrm{Tr}(\Sigma^2) = \sum_{i=1}^n \lambda_i^2 \leq \left(\sum_{i=1}^n \lambda_i\right)^2 = \|A\|_{\mathrm{F}}^4,$$

$$\|\Sigma\|_{\mathrm{op}} = A x_{i \in [n]} \lambda_i \leq \sum_{i=1}^n \lambda_i = \|A\|_{\mathrm{F}}^2.$$

Plug in Eq. (156), we get the desired equation. $\qquad\square$

Next lemma proves a concentration inequality on which Lemma C.2 relies.

**Lemma F.3.** *Given a symmetric matrix $H$, let $u, v \sim \mathcal{N}(0, I)$ be two independent random vectors, we have*

$$\forall t \geq 1, \Pr\left[(u^\top H v)^2 \geq t \|H\|_{\mathrm{F}}^2\right] \leq 3\exp(-\sqrt{t}/4). \tag{158}$$

*Proof.* Condition on $v$, $u^\top H v$ is a Gaussian random variable with mean zero and variance $\|Hv\|_2^2$. Therefore we have,

$$\forall v, \Pr\left[\left(u^\top H v\right)^2 \geq \sqrt{t} \|Hv\|_2^2\right] \leq \exp(-\sqrt{t}/2). \tag{159}$$

By Corollary F.2 and basic algebra we get,

$$\Pr\left[\|Hv\|_2^2 \geq \sqrt{t} \|H\|_F^2\right] \leq 2\exp(-\sqrt{t}/4). \tag{160}$$

Consequently,

$$\begin{aligned}
&\mathbb{E}\left[\mathbb{I}\left[(u^\top H v)^2 \geq t \|H\|_F^2\right]\right] \\
\leq\ &\mathbb{E}\left[\mathbb{I}\left[(u^\top H v)^2 \geq \sqrt{t}\|Hv\|_2^2 \text{ or } \|Hv\|_2^2 \geq \sqrt{t}\|H\|_F^2\right]\right] \\
\leq\ &\mathbb{E}\left[\mathbb{I}\left[(u^\top H v)^2 \geq \sqrt{t}\|Hv\|_2^2\right] \mid v\right] + \mathbb{E}\left[\mathbb{I}\left[\|Hv\|_2^2 \geq \sqrt{t}\|H\|_F^2\right]\right] \\
\leq\ &3\exp(-\sqrt{t}/4). \qquad\qquad\text{(Combining Eq. (159) and Eq. (160))}
\end{aligned}$$

$\square$

The next two lemmas are dedicated to prove anti-concentration inequalities that is used in Lemma C.2.

**Lemma F.4** (Lemma 1 of Laurent, Massart [47]). *Let* $(y_1, \cdots, y_n)$ *be i.i.d.* $\mathcal{N}(0,1)$ *Gaussian variables. Let* $a = (a_1, \cdots, a_n)$ *be non-negative coefficient. Let*

$$\|a\|_2^2 = \sum_{i=1}^n a_i^2.$$

*Then for any positive t,*

$$\Pr\left(\sum_{i=1}^n a_i y_i^2 \leq \sum_{i=1}^n a_i - 2\|a\|_2 \sqrt{t}\right) \leq \exp(-t). \tag{161}$$

**Lemma F.5.** *Given a symmetric matrix* $H \in \mathbb{R}^{n\times n}$, *let* $u, v \sim \mathcal{N}(0, I)$ *be two independent random vectors. Then*

$$\Pr\left[(u^\top H v)^2 \geq \frac{1}{8}\|H\|_F^2\right] \geq \frac{1}{64}. \tag{162}$$

*Proof.* Since $u, v$ are independent, by the isotropy of Guassian vectors we can assume that $H = \operatorname{diag}(\lambda_1, \cdots, \lambda_n)$. Note that condition on $v$, $u^\top H v$ is a Gaussian random variable with mean zero and variance $\|Hv\|_2^2$. As a result,

$$\forall v, \Pr\left[\left(u^\top H v\right)^2 \geq \frac{1}{4}\|Hv\|_2^2 \mid v\right] \geq \frac{1}{2}. \tag{163}$$

On the other hand, $\|Hv\|_2^2 = \sum_{i=1}^n \lambda_i^2 v_i^2$. Invoking Lemma F.4 we have

$$\begin{aligned}
&\Pr\left[\|Hv\|_2^2 \geq \frac{1}{2}\|H\|_F^2\right] \\
\geq\ &\Pr\left[\|Hv\|_2^2 \geq \|H\|_F^2 - \frac{1}{2}\sqrt{\sum_{i=1}^n \lambda_i^4}\right] \\
=\ &\Pr\left[\sum_{i=1}^n \lambda_i^2 v_i^2 \geq \sum_{i=1}^n \lambda_i^2 - \frac{1}{2}\sqrt{\sum_{i=1}^n \lambda_i^4}\right] \qquad\text{(By definition)} \\
\geq\ &1 - \exp(-1/16) \geq \frac{1}{32}. \tag{164}
\end{aligned}$$

Combining Eq. (163) and Eq. (164) we get,

$$\Pr\left[(u^\top H v)^2 \ge \frac{1}{8}\left\|H\right\|_F^2\right] \ge \Pr\left[(u^\top H v)^2 \ge \frac{1}{4}\left\|Hv\right\|_2^2, \left\|Hv\right\|_F^2 \ge \frac{1}{2}\left\|H\right\|_F^2\right] \ge \frac{1}{64}.$$

$\square$

The following lemma justifies the cap in the loss function.

**Lemma F.6.** *Given a symmetric matrix $H$, let $u, v \sim \mathcal{N}(0, I)$ be two independent random vectors. Let $\kappa_2, c_1 \in \mathbb{R}_+$ be two numbers satisfying $\kappa_2 \ge 640\sqrt{2}c_1$, then*

$$\min\left(c_1^2, \left\|H\right\|_F^2\right) \le 2\mathbb{E}\left[\min\left(\kappa_2^2, \left(u^\top H v\right)^2\right)\right]. \tag{165}$$

*Proof.* Let $x = \left(u^\top H v\right)^2$ for simplicity. Consider the following two cases:

**Case 1:** $\left\|H\right\|_F \le \kappa_2/40$. In this case we exploit the tail bound of random variable $x$. Specifically,

$$
\begin{aligned}
&\mathbb{E}\left[\left(u^\top H v\right)^2\right] - \mathbb{E}\left[\min\left(\kappa_2^2, \left(u^\top H v\right)^2\right)\right] \\
&= \int_{\kappa_2^2}^\infty \Pr\left[x \ge t\right]dt \\
&\le 3\int_{\kappa_2^2}^\infty \exp\left(-\frac{1}{4}\sqrt{\frac{t}{\left\|H\right\|_F^2}}\right)dt && \text{(By Lemma F.3)} \\
&= 24\exp\left(-\frac{\kappa_2}{4\left\|H\right\|_F}\right)\left\|H\right\|_F\left(\kappa_2 + 4\left\|H\right\|_F\right) \\
&\le 48\exp\left(-\frac{\kappa_2}{4\left\|H\right\|_F}\right)\left\|H\right\|_F\kappa_2 && (4\left\|H\right\|_F \le \kappa_2 \text{ in this case}) \\
&\le 48\cdot\frac{4\left\|H\right\|_F}{384\kappa_2}\left\|H\right\|_F\kappa_2 && (\exp(-x) \le \tfrac{1}{384x} \text{ when } x \ge 10) \\
&\le \frac{\left\|H\right\|_F^2}{2}.
\end{aligned}
$$

As a result,

$$\mathbb{E}\left[\min\left(\kappa_2^2, \left(u^\top H v\right)^2\right)\right] \ge \mathbb{E}\left[\left(u^\top H v\right)^2\right] - \frac{\left\|H\right\|_F^2}{2} = \frac{\left\|H\right\|_F^2}{2}. \tag{166}$$

**Case 2:** $\left\|H\right\|_F > \kappa_2/40$. In this case, we exploit the anti-concentration result of random variable $x$. Note that by the choice of $\kappa_2$, we have

$$\left\|H\right\|_F > \kappa_2/40 \implies \frac{1}{8}\left\|H\right\|_F^2 \ge 64c_1^2.$$

As a result,

$$
\begin{aligned}
&\mathbb{E}\left[\min\left(\kappa_2^2, \left(u^\top H v\right)^2\right)\right] \\
&\ge 64c_1^2\Pr\left[\min\left(\kappa_2^2, \left(u^\top H v\right)^2\right) \ge 64c_1^2\right] \\
&\ge 64c_1^2\Pr\left[\left(u^\top H v\right)^2 \ge 64c_1^2\right] && \text{(By definition of } \kappa_2) \\
&\ge 64c_1^2\Pr\left[\left(u^\top H v\right)^2 \ge \frac{1}{8}\left\|H\right\|_F^2\right] \\
&\ge c_1^2. && \text{(By Lemma F.5)}
\end{aligned}
$$

Therefore, in both cases we get

$$\mathbb{E}\left[\min\left(\kappa_2^2, \left(u^\top H v\right)^2\right)\right] \ge \frac{1}{2}\min\left(c_1^2, \left\|H\right\|_F^2\right), \tag{167}$$

which proofs Eq. (165). $\square$

Following lemmas are analogs to Cauchy-Schwartz inequality (in vector/matrix forms), which are used to prove Lemma D.1 for reinforcement learning case.

**Lemma F.7.** *For a random vector $x \in \mathbb{R}^d$ and random variable $r$, we have*

$$\|\mathbb{E}[rx]\|_2^2 \leq \|\mathbb{E}[xx^\top]\|_{\mathrm{op}} \mathbb{E}[r^2]. \tag{168}$$

*Proof.* Note that for any vector $g \in \mathbb{R}^d$, $\|g\|_2^2 = \sup_{u \in S^{d-1}} \langle u, g \rangle^2$. As a result,

$$
\begin{aligned}
\|\mathbb{E}[rx]\|_2^2 &= \sup_{u \in S^{d-1}} \langle u, \mathbb{E}[rx] \rangle^2 = \sup_{u \in S^{d-1}} \mathbb{E}[r \langle u, x \rangle]^2 \\
&\leq \sup_{u \in S^{d-1}} \mathbb{E}\left[ \langle u, x \rangle^2 \right] \mathbb{E}[r^2] \qquad\qquad \text{(Hölder Ineqaulity)} \\
&= \|\mathbb{E}[xx^\top]\|_{\mathrm{op}} \mathbb{E}[r^2].
\end{aligned}
$$

$\square$

**Lemma F.8.** *For a symmetric random matrix $H \in \mathbb{R}^{d \times d}$ and random variable $r$, we have*

$$\|\mathbb{E}[rH]\|_{\mathrm{sp}}^2 \leq \|\mathbb{E}[HH^\top]\|_{\mathrm{sp}} \mathbb{E}[r^2]. \tag{169}$$

*Proof.* Note that for any matrix $\boldsymbol{G} \in \mathbb{R}^d$, $\|H\|_{\mathrm{sp}}^2 = \sup_{u,v \in S^{d-1}} (u^\top \boldsymbol{G} v)^2$. As a result,

$$
\begin{aligned}
\|\mathbb{E}[rH]\|_2^2 &= \sup_{u,v \in S^{d-1}} (u^\top \mathbb{E}[rH] v)^2 = \sup_{u,v \in S^{d-1}} \mathbb{E}[r(u^\top H v)]^2 \\
&\leq \sup_{u,v \in S^{d-1}} \mathbb{E}\left[ (u^\top H v)^2 \right] \mathbb{E}[r^2] \qquad\qquad \text{(Hölder Ineqaulity)} \\
&= \sup_{u,v \in S^{d-1}} \mathbb{E}[u^\top H v v^\top H^\top u] \mathbb{E}[r^2] \\
&\leq \sup_{u \in S^{d-1}} \mathbb{E}[u^\top H H^\top u] \mathbb{E}[r^2] \\
&= \|\mathbb{E}[HH^\top]\|_{\mathrm{sp}} \mathbb{E}[r^2].
\end{aligned}
$$

$\square$

**Lemma F.9.** *For a random matrix $x \in \mathbb{R}^d$ and a positive random variable $r$, we have*

$$\left\|\mathbb{E}[rxx^\top]\right\|_{\mathrm{sp}}^2 \leq \|\mathbb{E}[x^{\otimes 4}]\|_{\mathrm{sp}} \mathbb{E}[r^2]. \tag{170}$$

*Proof.* Since $r$ is non-negative, we have $\mathbb{E}[rxx^\top] \succeq 0$. As a result,

$$\left\|\mathbb{E}[rxx^\top]\right\|_{\mathrm{sp}} = \sup_{u \in S^{d-1}} u^\top \mathbb{E}[rxx^\top] u.$$

It follows that

$$
\begin{aligned}
\left\|\mathbb{E}[rxx^\top]\right\|_{\mathrm{sp}}^2 &= \sup_{u \in S^{d-1}} (u^\top \mathbb{E}[rxx^\top] u)^2 = \sup_{u \in S^{d-1}} \mathbb{E}\left[ r \langle u, x \rangle^2 \right]^2 \\
&\leq \sup_{u \in S^{d-1}} \mathbb{E}\left[ \langle u, x \rangle^4 \right] \mathbb{E}[r^2] \qquad\qquad \text{(Hölder Inequality)} \\
&= \sup_{u \in S^{d-1}} \langle u^{\otimes 4}, \mathbb{E}[x^{\otimes 4}] \rangle \mathbb{E}[r^2] \\
&= \|\mathbb{E}[x^{\otimes 4}]\|_{\mathrm{sp}} \mathbb{E}[r^2].
\end{aligned}
$$

$\square$

Following lemmas exploit the isotropism of Gaussian vectors, and are used to verify the Lipschitzness assumption of Example 4.3. In fact, we heavily rely on the fact that, for a fixed vector $g \in \mathbb{R}^d$, $\langle g, u \rangle \sim \mathcal{N}(0, \|g\|_2^2)$ when $u \sim \mathcal{N}(0, I)$,

**Lemma F.10.** *For two vectors $p, q \in \mathbb{R}^d$ and a bounded function $f : \mathbb{R}^d \to [-B, B]$, we have*

$$\mathbb{E}_{u \sim \mathcal{N}(0, \sigma^2 I)}[(\langle p, u \rangle - \langle q, u \rangle) f(u)] \le \sigma B \|p - q\|_2. \tag{171}$$

*Proof.* By Hölder inequality we have

$$\mathbb{E}_{u \sim \mathcal{N}(0, \sigma^2 I)}[(\langle p, u \rangle - \langle q, u \rangle) f(u)] \tag{172}$$

$$\le \mathbb{E}_{u \sim \mathcal{N}(0, \sigma^2 I)}\left[(\langle p, u \rangle - \langle q, u \rangle)^2\right]^{1/2} \mathbb{E}_{u \sim \mathcal{N}(0, \sigma^2 I)}\left(f(u)^2\right)^{1/2}. \tag{173}$$

Note that $u$ is isotropic. As a result $\langle p - q, u \rangle \sim \mathcal{N}(0, \sigma^2 \|p - q\|_2^2)$. It follows that

$$\mathbb{E}_{u \sim \mathcal{N}(0, \sigma^2 I)}\left[(\langle p, u \rangle - \langle q, u \rangle)^2\right]^{1/2} \mathbb{E}_{u \sim \mathcal{N}(0, \sigma^2 I)}\left(f(u)^2\right)^{1/2} \tag{174}$$

$$\le \sigma \|p - q\|_2 B. \tag{175}$$

$\square$

**Lemma F.11.** *For two vectors $x, y \in \mathbb{R}^d$, if $\|x\|_2 = 1$ we have*

$$\|x - y\|_2^2 \ge (1 - \langle x, y \rangle)^2. \tag{176}$$

*Proof.* By basic algebra we get

$$\|x - y\|_2^2 = \|x - \langle x, y \rangle x + \langle x, y \rangle x - y\|_2^2 \tag{177}$$

$$= \|x - \langle x, y \rangle x\|_2^2 + \|\langle x, y \rangle x - y\|_2^2 - 2(1 - \langle x, y \rangle) \langle x, \langle x, y \rangle x - y \rangle \tag{178}$$

$$= \|x - \langle x, y \rangle x\|_2^2 + \|\langle x, y \rangle x - y\|_2^2 \tag{179}$$

$$= (1 - \langle x, y \rangle)^2 + \|\langle x, y \rangle x - y\|_2^2 \ge (1 - \langle x, y \rangle)^2. \tag{180}$$

$\square$

**Lemma F.12.** *For vectors $p, x_1, x_2 \in \mathbb{R}^d$ and a bounded function $f : \mathbb{R}^d \to [0, B]$, we have*

$$\mathbb{E}_{u \sim \mathcal{N}(0, \sigma^2 I)}[\langle p, u \rangle (f(x_1 + u) - f(x_2 + u))] \le B \|p\|_2 \|x_1 - x_2\|_2 \left(6 + \frac{3(\|x_1\|_2 + \|x_2\|_2)}{\sigma}\right). \tag{181}$$

*Proof.* The lemma is proved by coupling argument. With out loss of generality, we assume that $x_1 = Ce_1$ and $x_2 = -Ce_1$ where $e_1$ is the first basis vector. That is, $\|x_1 - x_2\|_2 = 2C$. For a vector $x \in \mathbb{R}^d$, let $F(x)$ be the density of distribution $\mathcal{N}(0, \sigma^2 I)$ at $x$. Then we have,

$$\mathbb{E}_{u \sim \mathcal{N}(0, \sigma^2 I)}[\langle p, u \rangle f(x_1 + u)] = \int_{y \in \mathbb{R}^d} F(y - x_1) \langle p, y - x_1 \rangle f(y) \mathrm{d}y. \tag{182}$$

As a result,

$$\mathbb{E}_{u \sim \mathcal{N}(0, \sigma^2 I)}[\langle p, u \rangle (f(x_1 + u) - f(x_2 + u))] \tag{183}$$

$$= \int_{y \in \mathbb{R}^d} F(y - x_1) \langle p, y - x_1 \rangle f(y) \mathrm{d}y - F(y - x_2) \langle p, y - x_2 \rangle f(y) \mathrm{d}y. \tag{184}$$

Define $G(y) = \min(F(y - x_1), F(y - x_2))$. It follows that,

$$\int_{y \in \mathbb{R}^d} F(y - x_1) \langle p, y - x_1 \rangle f(y) \mathrm{d}y - F(y - x_2) \langle p, y - x_2 \rangle f(y) \mathrm{d}y \tag{185}$$

$$\le \int_{y \in \mathbb{R}^d} G(y)|\langle p, y - x_1 \rangle - \langle p, y - x_2 \rangle| f(y) \mathrm{d}y \tag{186}$$

$$+ \int_{y \in \mathbb{R}^d} (F(y - x_1) - G(y))|\langle p, y - x_1 \rangle| f(y) \mathrm{d}y \tag{187}$$

$$+ \int_{y \in \mathbb{R}^d} (F(y - x_2) - G(y))|\langle p, y - x_2 \rangle| f(y) \mathrm{d}y. \tag{188}$$

The term in Eq. (186) can be bounded by

$$\int_{y\in\mathbb{R}^d} G(y)|\langle p, y-x_1\rangle - \langle p, y-x_2\rangle|f(y)\mathrm{d}y \tag{189}$$

$$\leq \int_{y\in\mathbb{R}^d} G(y)\mathrm{d}y \sup_{y\in\mathbb{R}^d} |\langle p, x_2-x_1\rangle f(y)| \leq \|x_2-x_1\|_2 \|p\|_2 B. \tag{190}$$

Note that the terms in Eq. (187) and Eq. (188) are symmetric. Therefore in the following we only prove an upper bound for Eq. (187). In the following, we use the notation $[y]_{-1}$ to denote the $(d-1)$-dimensional vector generated by removing the first coordinate of $y$. Let $P(x)$ be the density of distribution $\mathcal{N}(0,\sigma^2)$ at point $x \in \mathbb{R}$. By the symmetricity of Gaussian distribution, $F(y) = P([y]_1)F([y]_{-1})$.

By definition, $F(y-x_1) - G(y) = 0$ for $y$ such that $[y]_1 \leq 0$. As a result,

$$\int_{y:[y]_1>0} (F(y-x_1) - G(y))|\langle p, y-x_1\rangle|f(y)\mathrm{d}y \tag{191}$$

$$= \int_{y:[y]_1>0} (F(y-x_1) - F(y-x_2))|\langle p, y-x_1\rangle|f(y)\mathrm{d}y \tag{192}$$

$$\leq \int_{[y]_1>0} \mathrm{d}[y]_1(P([y]_1 - [x_1]_1) - P([y]_1 - [x_2]_1))\mathbb{E}_{[y]_{-1}}[(|\langle [p]_{-1}, [y-x_1]_{-1}\rangle + [p]_1[y-x_1]_1|)f(y)]. \tag{193}$$

Note that conditioned on $[y]_1$, $[y-x_1]_{-1} \sim \mathcal{N}(0,\sigma^2 I)$. Consequently,

$$\mathbb{E}_{[y]_{-1}}[|\langle [p]_{-1}, [y-x_1]_{-1}\rangle|f(y)] \tag{194}$$

$$\leq \mathbb{E}_{[y]_{-1}}\left[\langle [p]_{-1}, [y-x_1]_{-1}\rangle^2\right]^{1/2} \mathbb{E}_{[y]_{-1}}\left[f(y)^2\right]^{1/2} \tag{195}$$

$$\leq B\sigma \|[p]_{-1}\|_2 \leq B\sigma \|p\|_2. \tag{196}$$

On the other hand, we have

$$\int_{[y]_1>0} \mathrm{d}[y]_1(P([y]_1 - [x_1]_1) - P([y]_1 - [x_2]_1)) \leq \mathrm{TV}\big(\mathcal{N}(-C,\sigma^2), \mathcal{N}(C,\sigma^2)\big) \leq \frac{1}{\sigma}\|x_1-x_2\|_2. \tag{197}$$

It follows that,

$$\int_{y:[y]_1>0} (F(y-x_1) - G(y))|\langle p, y-x_1\rangle|f(y)\mathrm{d}y \tag{198}$$

$$\leq B\|p\|_2 \|x_1-x_2\|_2 + B\int_{[y]_1>0} \mathrm{d}[y]_1(P([y]_1 - [x_1]_1) - P([y]_1 - [x_2]_1))|[p]_1[y-x_1]_1|. \tag{199}$$

Note that the second term in Eq. (199) involves only one dimensional Gaussian distribution. Invoking Lemma F.14, the second term can be bounded by $\|p\|_2 \left(\frac{3\|x_1-x_2\|_2}{\sigma}\|x_1\|_2 + 4\|x_1-x_2\|_2\right)$. Therefore, we have

$$\int_{y:[y]_1>0} (F(y-x_1) - G(y))|\langle p, y-x_1\rangle|f(y)\mathrm{d}y \leq 5B\|p\|_2 \|x_1-x_2\|_2 + \|p\|_2 \frac{3B\|x_1-x_2\|_2}{\sigma}. \tag{200}$$

$$\square$$

**Lemma F.13.** *For vectors $p, q, x_1, x_2 \in \mathbb{R}^d$ with $\|x_1 \leq 1\| \leq 1 \|x_2\|_2 \leq 1$ and a bounded function $f : \mathbb{R}^d \to [0, B]$, we have*

$$\mathbb{E}_{u\sim\mathcal{N}(0,\sigma^2 I)}[\langle p, u\rangle \langle q, u\rangle (f(x_1+u) - f(x_2+u))] \leq \mathrm{poly}(B, \sigma, 1/\sigma, \|p\|_2, \|q\|_2) \|x_1-x_2\|_2. \tag{201}$$

*Proof.* Proof of this lemma is similar to that of Lemma F.13. With out loss of generality, we assume that $x_1 = Ce_1$ and $x_2 = -Ce_1$ where $e_1$ is the first basis vector. That is, $\|x_1 - x_2\|_2 = 2C$. For a vector $x \in \mathbb{R}^d$, let $F(x)$ be the density of distribution $\mathcal{N}(0, \sigma^2 I)$ at $x$. Then we have,

$$\mathbb{E}_{u \sim \mathcal{N}(0, \sigma^2 I)}[\langle p, u \rangle \langle q, u \rangle f(x_1 + u)] = \int_{y \in \mathbb{R}^d} F(y - x_1) \langle p, y - x_1 \rangle \langle q, y - x_1 \rangle f(y) \mathrm{d}y. \quad (202)$$

As a result,

$$\mathbb{E}_{u \sim \mathcal{N}(0, \sigma^2 I)}[\langle p, u \rangle \langle q, u \rangle (f(x_1 + u) - f(x_2 + u))] \quad (203)$$

$$= \int_{y \in \mathbb{R}^d} F(y - x_1) \langle p, y - x_1 \rangle \langle q, y - x_1 \rangle f(y) \mathrm{d}y - F(y - x_2) \langle p, y - x_2 \rangle \langle q, y - x_2 \rangle f(y) \mathrm{d}y. \quad (204)$$

Define $G(y) = \min(F(y - x_1), F(y - x_2))$. It follows that,

$$\int_{y \in \mathbb{R}^d} F(y - x_1) \langle p, y - x_1 \rangle \langle q, y - x_1 \rangle f(y) \mathrm{d}y - F(y - x_2) \langle p, y - x_2 \rangle \langle q, y - x_2 \rangle f(y) \mathrm{d}y \quad (205)$$

$$\leq \int_{y \in \mathbb{R}^d} G(y) |\langle p, y - x_1 \rangle \langle q, y - x_1 \rangle - \langle p, y - x_2 \rangle \langle q, y - x_2 \rangle| f(y) \mathrm{d}y \quad (206)$$

$$+ \int_{y \in \mathbb{R}^d} (F(y - x_1) - G(y)) |\langle p, y - x_1 \rangle \langle q, y - x_1 \rangle| f(y) \mathrm{d}y \quad (207)$$

$$+ \int_{y \in \mathbb{R}^d} (F(y - x_2) - G(y)) |\langle p, y - x_2 \rangle \langle q, y - x_2 \rangle| f(y) \mathrm{d}y. \quad (208)$$

By basic algebra we have

$$\int_{y \in \mathbb{R}^d} G(y) |\langle p, y - x_1 \rangle \langle q, y - x_1 \rangle - \langle p, y - x_2 \rangle \langle q, y - x_2 \rangle| f(y) \mathrm{d}y \quad (209)$$

$$\leq \int_{y \in \mathbb{R}^d} G(y) |\langle p, x_2 - x_1 \rangle \langle q, y - x_1 \rangle| f(y) \mathrm{d}y \quad (210)$$

$$+ \int_{y \in \mathbb{R}^d} G(y) |\langle p, y - x_2 \rangle \langle q, x_2 - x_1 \rangle| f(y) \mathrm{d}y. \quad (211)$$

Continue with the first term we get

$$\int_{y \in \mathbb{R}^d} G(y) |\langle p, x_2 - x_1 \rangle \langle q, y - x_1 \rangle| f(y) \mathrm{d}y \quad (212)$$

$$\leq \|x_2 - x_1\|_2 \|p\|_2 \int_{y \in \mathbb{R}^d} G(y) |\langle q, y - x_1 \rangle| f(y) \mathrm{d}y \quad (213)$$

$$\leq \|x_2 - x_1\|_2 \|p\|_2 \int_{y \in \mathbb{R}^d} F(y - x_1) |\langle q, y - x_1 \rangle| f(y) \mathrm{d}y \quad (214)$$

$$= \|x_2 - x_1\|_2 \|p\|_2 \mathbb{E}_{u \sim \mathcal{N}(0, \sigma^2 I)}[|\langle q, u \rangle| f(u + x_1)] \quad (215)$$

$$\leq \|x_2 - x_1\|_2 \|p\|_2 \mathbb{E}_{u \sim \mathcal{N}(0, \sigma^2 I)}\left[\langle q, u \rangle^2\right]^{1/2} \mathbb{E}_{u \sim \mathcal{N}(0, \sigma^2 I)}\left[f(u + x_1)^2\right]^{1/2} \quad (216)$$

$$\leq \sigma B \|x_2 - x_1\|_2 \|p\|_2 \|q\|_2. \quad (217)$$

For the same reason, the second term in Eq. (211) is also bounded by $\sigma B \|x_2 - x_1\|_2 \|p\|_2 \|q\|_2$. As a result,

$$\int_{y \in \mathbb{R}^d} G(y) |\langle p, y - x_1 \rangle \langle q, y - x_1 \rangle - \langle p, y - x_2 \rangle \langle q, y - x_2 \rangle| f(y) \mathrm{d}y \leq 2\sigma B \|x_2 - x_1\|_2 \|p\|_2 \|q\|_2. \quad (218)$$

Now we turn to the term in Eq. (207). Note that the terms in Eq. (207) and Eq. (208) are symmetric. Therefore in the following we only prove an upper bound for Eq. (207). In the following, we use the

notation $[y]_{-1}$ to denote the $(d-1)$-dimensional vector generated by removing the first coordinate of $y$. Let $P(x)$ be the density of distribution $\mathcal{N}(0, \sigma^2)$ at point $x \in \mathbb{R}$. By the symmetricity of Gaussian distribution, $F(y) = P([y]_1)F([y]_{-1})$.

By definition, $F(y - x_1) - G(y) = 0$ for $y$ such that $[y]_1 \leq 0$. Define the shorthand $I = |[p]_1[y - x_1]_1|$, $J = |\langle [p]_{-1}, [y - x_1]_{-1} \rangle|$, $C = |[q]_1[y - x_1]_1|$, $D = |\langle [q]_{-1}, [y - x_1]_{-1} \rangle|$. When condition on $[y]_1$, $A, C$ are constants. As a result,

$$\int_{y:[y]_1>0} (F(y - x_1) - G(y))|\langle p, y - x_1 \rangle \langle q, y - x_1 \rangle|f(y)\mathrm{d}y \tag{219}$$

$$= \int_{y:[y]_1>0} (F(y - x_1) - F(y - x_2))|\langle p, y - x_1 \rangle \langle q, y - x_1 \rangle|f(y)\mathrm{d}y \tag{220}$$

$$\leq \int_{[y]_1>0} \mathrm{d}[y]_1(P([y]_1 - [x_1]_1) - P([y_1] - [x_2]_1))\mathbb{E}_{[y]_{-1}}[|(I + J)(C + D)|f(y)] \tag{221}$$

$$\leq \int_{[y]_1>0} \mathrm{d}[y]_1(P([y]_1 - [x_1]_1) - P([y_1] - [x_2]_1))IC\mathbb{E}_{[y]_{-1}}[f(y)] \tag{222}$$

$$+ \int_{[y]_1>0} \mathrm{d}[y]_1(P([y]_1 - [x_1]_1) - P([y_1] - [x_2]_1))I\mathbb{E}_{[y]_{-1}}[Df(y)] \tag{223}$$

$$+ \int_{[y]_1>0} \mathrm{d}[y]_1(P([y]_1 - [x_1]_1) - P([y_1] - [x_2]_1))C\mathbb{E}_{[y]_{-1}}[Jf(y)] \tag{224}$$

$$+ \int_{[y]_1>0} \mathrm{d}[y]_1(P([y]_1 - [x_1]_1) - P([y_1] - [x_2]_1))\mathbb{E}_{[y]_{-1}}[JDf(y)]. \tag{225}$$

Note that conditioned on $[y]_1$, $[y - x_1]_{-1} \sim \mathcal{N}(0, \sigma^2 I)$. Consequently,

$$\mathbb{E}_{[y]_{-1}}[Df(y)] \leq \mathbb{E}_{[y]_{-1}}[D^2]^{1/2}\mathbb{E}_{[y]_{-1}}[f(y)^2]^{1/2} \leq B\sigma \|q\|_2 \tag{226}$$

$$\mathbb{E}_{[y]_{-1}}[Jf(y)] \leq \mathbb{E}_{[y]_{-1}}[J^2]^{1/2}\mathbb{E}_{[y]_{-1}}[f(y)^2]^{1/2} \leq B\sigma \|p\|_2 \tag{227}$$

$$\mathbb{E}_{[y]_{-1}}[JDf(y)] \leq \mathbb{E}_{[y]_{-1}}[J^4]^{1/4}\mathbb{E}_{[y]_{-1}}[D^4]^{1/4}\mathbb{E}_{[y]_{-1}}[f(y)^2]^{1/2} \leq \sqrt{3}B\sigma^2 \|p\|_2 \|q\|_2. \tag{228}$$

Invoking Lemma F.14 we get

$$\int_{[y]_1>0} \mathrm{d}[y]_1(P([y]_1 - [x_1]_1) - P([y_1] - [x_2]_1)) \leq \frac{1}{\sigma}\|x_1 - x_2\|_2, \tag{229}$$

$$\int_{[y]_1>0} \mathrm{d}[y]_1(P([y]_1 - [x_1]_1) - P([y_1] - [x_2]_1))I \leq \mathrm{poly}(B, \sigma, 1/\sigma)\|p\|_2 \|x_1 - x_2\|_2, \tag{230}$$

$$\int_{[y]_1>0} \mathrm{d}[y]_1(P([y]_1 - [x_1]_1) - P([y_1] - [x_2]_1))C \leq \mathrm{poly}(B, \sigma, 1/\sigma)\|q\|_2 \|x_1 - x_2\|_2, \tag{231}$$

$$\int_{[y]_1>0} \mathrm{d}[y]_1(P([y]_1 - [x_1]_1) - P([y_1] - [x_2]_1))IC \leq \mathrm{poly}(B, \sigma, 1/\sigma)\|p\|_2 \|q\|_2 \|x_1 - x_2\|_2. \tag{232}$$

As a result, we get

$$\int_{y:[y]_1>0} (F(y - x_1) - G(y))|\langle p, y - x_1 \rangle \langle q, y - x_1 \rangle|f(y)\mathrm{d}y \leq \mathrm{poly}(B, \sigma, 1/\sigma, \|p\|_2, \|q\|_2)\|x_1 - x_2\|_2.$$

$\square$

**Lemma F.14.** *Let $P(x)$ be the density function of $\mathcal{N}(0, \sigma^2)$. Given a scalar $x \geq 0$. we have*

$$\int_{y>0} (P(y - x) - P(y + x))\mathrm{d}y \leq \frac{x}{\sigma}, \tag{233}$$

$$\int_{y>0} (P(y - x) - P(y + x))|y - x|\mathrm{d}y \leq \frac{3x^2}{\sigma} + 4x, \tag{234}$$

$$\int_{y>0} (P(y - x) - P(y + x))|y - x|^2\mathrm{d}y \leq \frac{x^2}{\sigma} + 4\sigma x. \tag{235}$$

*Proof.* Note that $\mathrm{TV}\big(\mathcal{N}(-x,\sigma^2),\mathcal{N}(x,\sigma^2)\big) \leq \frac{x}{\sigma}$. Consequently,

$$\int_{y>0}(P(y-x)-P(y+x))\mathrm{d}y \leq \frac{x}{\sigma}. \tag{236}$$

Using the same TV-distance bound, we get

$$\int_{y>0}(P(y-x)-P(y+x))|y-x|\mathrm{d}y \tag{237}$$

$$\leq \int_{y>0}(P(y-x)-P(y+x))((y-x)+2x)\mathrm{d}y \tag{238}$$

$$\leq \int_{y>2x}(P(y-x)-P(y+x))(y-x)\mathrm{d}y + \frac{3x^2}{\sigma}. \tag{239}$$

Recall $P(x) = \frac{1}{\sqrt{2\pi}}\exp\left(-\frac{x^2}{2\sigma^2}\right)$. By algebraic manipulation we have

$$\int_{y>2x}\left(\exp\left(-\frac{(y-x)^2}{2\sigma^2}\right)-\exp\left(-\frac{(y+x)^2}{2\sigma^2}\right)\right)(y-x)\mathrm{d}y \tag{240}$$

$$\leq \int_{y>2x}\exp\left(-\frac{(y-x)^2}{2\sigma^2}\right)\left(1-\exp\left(-\frac{4xy}{2\sigma^2}\right)\right)(y-x)\mathrm{d}y \tag{241}$$

$$\leq \int_{y>2x}\exp\left(-\frac{(y-x)^2}{2\sigma^2}\right)\frac{4xy}{2\sigma^2}(y-x)\mathrm{d}y \tag{242}$$

$$\leq \int_{y>2x}\exp\left(-\frac{(y-x)^2}{2\sigma^2}\right)\frac{4x}{\sigma^2}(y-x)^2\mathrm{d}y \tag{243}$$

$$\leq 4x. \tag{244}$$

Now we turn to the third inequality. Because $|y-x| \leq x$ for $y \in [0,2x]$, using the TV-distance bound we get

$$\int_{y>0}(P(y-x)-P(y+x))|y-x|^2\mathrm{d}y \leq \int_{y>2x}(P(y-x)-P(y+x))(y-x)^2\mathrm{d}y + \frac{x^2}{\sigma}. \tag{245}$$

By algebraic manipulation we have

$$\int_{y>2x}\left(\exp\left(-\frac{(y-x)^2}{2\sigma^2}\right)-\exp\left(-\frac{(y+x)^2}{2\sigma^2}\right)\right)(y-x)^2\mathrm{d}y \tag{246}$$

$$\leq \int_{y>2x}\exp\left(-\frac{(y-x)^2}{2\sigma^2}\right)\left(1-\exp\left(-\frac{4xy}{2\sigma^2}\right)\right)(y-x)^2\mathrm{d}y \tag{247}$$

$$\leq \int_{y>2x}\exp\left(-\frac{(y-x)^2}{2\sigma^2}\right)\frac{4xy}{2\sigma^2}(y-x)^2\mathrm{d}y \tag{248}$$

$$\leq \int_{y>2x}\exp\left(-\frac{(y-x)^2}{2\sigma^2}\right)\frac{4x}{\sigma^2}(y-x)^3\mathrm{d}y \tag{249}$$

$$\leq 4\sigma x. \tag{250}$$

$\square$

**Lemma F.15.** *For four vectors $p,q,v,w \in \mathbb{R}^d$ with unit norm and a bounded function $f : \mathbb{R}^d \to [-B,B]$, we have*

$$\mathbb{E}_{u\sim\mathcal{N}(0,\sigma^2 I)}[(\langle p,u\rangle\langle v,u\rangle - \langle q,u\rangle\langle w,u\rangle)f(u)] \leq \sqrt{3}\sigma^2 B(\|p-q\|_2 + \|v-w\|_2). \tag{251}$$

*Proof.* First of all, by telescope sum we get

$$\mathbb{E}_{u\sim\mathcal{N}(0,\sigma^2 I)}[(\langle p,u\rangle\langle v,u\rangle - \langle q,u\rangle\langle w,u\rangle)f(u)] \tag{252}$$

$$= \mathbb{E}_{u \sim \mathcal{N}(0,\sigma^2 I)}[(\langle p, u \rangle \langle v, u \rangle - \langle p, u \rangle \langle w, u \rangle)f(u)] \tag{253}$$

$$+ \mathbb{E}_{u \sim \mathcal{N}(0,\sigma^2 I)}[(\langle p, u \rangle \langle w, u \rangle - \langle q, u \rangle \langle w, u \rangle)f(u)]. \tag{254}$$

By Hölder inequality we have

$$\mathbb{E}_{u \sim \mathcal{N}(0,\sigma^2 I)}[(\langle p, u \rangle \langle v, u \rangle - \langle p, u \rangle \langle w, u \rangle)f(u)] \tag{255}$$

$$\leq \mathbb{E}_{u \sim \mathcal{N}(0,\sigma^2 I)}\left[\langle p, u \rangle^4\right]^{1/4} \mathbb{E}_{u \sim \mathcal{N}(0,\sigma^2 I)}\left[\langle v - w, u \rangle^4\right]^{1/4} \mathbb{E}_{u \sim \mathcal{N}(0,\sigma^2 I)}\left[f(u)^2\right]^{1/2} \tag{256}$$

$$\leq \sqrt{3}\sigma^2 \|p\|_2 \|v - w\|_2 B. \tag{257}$$

Similarly we have

$$\mathbb{E}_{u \sim \mathcal{N}(0,\sigma^2 I)}[(\langle p, u \rangle \langle w, u \rangle - \langle q, u \rangle \langle w, u \rangle)f(u)] \leq \sqrt{3}\sigma^2 \|p - q\|_2 \|w\|_2 B. \tag{258}$$

$\square$

## F.2  Helper Lemmas on Reinforcement Learning

**Lemma F.16** (Telescoping or Simulation Lemma, see Luo et al. [55], Agarwal et al. [2]). *For any policy $\pi$ and deterministic dynamical model $T, \hat{T}$, we have*

$$V_{\hat{T}}^\pi(s_1) - V_T^\pi(s_1) = \mathbb{E}_{\tau \sim \rho_T^\pi}\left[\sum_{h=1}^{H}\left(V_{\hat{T}}^\pi(\hat{T}(s_h, a_h)) - V_{\hat{T}}^\pi(T(s_h, a_h))\right)\right]. \tag{259}$$

**Lemma F.17** (Policy Gradient Lemma, see Sutton, Barto [69]). *For any policy $\pi_\psi$, deterministic dynamical model $T$ and reward function $r(s_h, a_h)$, we have*

$$\nabla_\psi V_T^{\pi_\psi} = \mathbb{E}_{\tau \sim \rho_T^{\pi_\psi}}\left[\left(\sum_{h=1}^{H} \nabla_\psi \log \pi_\psi(a_h \mid s_h)\right)\left(\sum_{h=1}^{H} r(s_h, a_h)\right)\right] \tag{260}$$

*Proof.* Note that

$$V_T^{\pi_\psi} = \int_\tau \Pr[\tau] \sum_{h=1}^{H} r(s_h, a_h)\, d\tau.$$

Take gradient w.r.t. $\psi$ in both sides, we have

$$\nabla_\psi V_T^{\pi_\psi} = \nabla_\psi \int_\tau \Pr[\tau] \sum_{h=1}^{H} r(s_h, a_h)\, d\tau$$

$$= \int_\tau (\nabla_\psi \Pr[\tau]) \sum_{h=1}^{H} r(s_h, a_h)\, d\tau$$

$$= \int_\tau \Pr[\tau](\nabla_\psi \log \Pr[\tau]) \sum_{h=1}^{H} r(s_h, a_h)\, d\tau$$

$$= \int_\tau \Pr[\tau]\left(\nabla_\psi \log \prod_{h=1}^{H} \pi_\psi(a_h \mid s_h)\right) \sum_{h=1}^{H} r(s_h, a_h)\, d\tau$$

$$= \int_\tau \Pr[\tau]\left(\sum_{h=1}^{H} \nabla_\psi \log \pi_\psi(a_h \mid s_h)\right) \sum_{h=1}^{H} r(s_h, a_h)\, d\tau$$

$$= \mathbb{E}_{\tau \sim \rho_T^{\pi_\psi}}\left[\left(\sum_{h=1}^{H} \nabla_\psi \log \pi_\psi(a_h \mid s_h)\right) \sum_{h=1}^{H} r(s_h, a_h)\right]$$

$\square$