# OpenReview forum: "Provable Model-based Nonlinear Bandit and Reinforcement Learning: Shelve Optimism, Embrace Virtual Curvature"
_NeurIPS.cc/2021/Conference — NeurIPS 2021 Poster_

### Official Review · Reviewer_SzXN · 2021-07-11

**Rating:** 6
**Confidence:** 2

**Summary:**

This work investigates model-based bandit and RL under non-linear function approximations. In particular, it focuses on convergence to approximate local maxima under deterministic reward (and deterministic dynamics for RL). Several technical results are derived showing that the presented model-based algorithm can have a favorable sample complexity that only depends on the sequential Rademacher complexity of the model class. The paper also contains some hardness results to better contextualize the contribution of this work.

**Limitations And Societal Impact:**

Yes

**Main Review:**

The paper is technically dense but the authors manage to provide enough intuition and explanation for the results. There are generally enough context for one to understand the corresponding results. I didn't go into the technical details, but the contributions seem solid, and the results and technical novelty could be interesting for the theoretical community.

One question I have is that to what extend, the results depend on the deterministic assumptions (deterministic dynamics and reward). The technical novelty as claimed by the authors is to explore with model-based curvature estimates. To achieve this, it seems from page 6 that the algorithm leverages the deterministic property to estimate the projections. The authors also mention some motivation for considering deterministic rewards (lines 136 -138). Overall, I would appreciate a concrete remark on both the deterministic dynamics and reward, explaining the potential difficulties/extensions of the current framework.

========after response======
I thank the authors for the response. I'm happy to keep my positive evaluation and hope that the authors could also include few sentences in the final version to clarity the assumptions.

**Time Spent Reviewing:**

2.5h

---

> ### Author Response · Authors · 2021-08-10
> **Response to Reviewer SzXN**
>
> We thank the reviewer for the review. We appreciate it that the reviewer finds “the contributions seem solid, and the results and technical novelty could be interesting for the theoretical community.” In the following, we address the reviewer’s comments in detail.
>
> > To what extend, the results depend on the deterministic assumptions (deterministic dynamics and reward)
>
> For bandits, we only use the deterministic assumptions to calculate the projection of gradient and Hessian by finite difference. In other words, our framework is compatible with any assumptions that lead to an efficient estimation of the gradient and Hessian, such as multiple-point feedback assumption in zero-order optimization literature [1].
>
> For model-based reinforcement learning, we do require deterministic dynamics to define the loss function in Eq. (6).
>
> [1] Liu Sijia, Chen Pin-Yu, Kailkhura Bhavya, Zhang Gaoyuan, Hero III Alfred O, Varshney Pramod K. A Primer on Zeroth-Order Optimization in Signal Processing and Machine Learning: Principals, Recent Advances, and Applications.

---

### Official Review · Reviewer_XRrF · 2021-07-16

**Rating:** 7
**Confidence:** 4

**Summary:**

This paper considers learning in deterministic bandit and reinforcement learning problems with function approximation. They quantify the complexity of the function class using the sequential Rademacher complexity of a related loss function (although in some settings, it is not super clear how this relates to the complexity of the reward/value function). In the bandit setting, they provide a general regret bound and instantiate this for various different assumptions on the model, including two layer neural networks, which is a nice result. To extend their results to reinforcement learning some assumptions are made so that the value function can be written as a non-linear function of the policy parameters and optimized accordingly. They then provide a slightly simpler loss function and show that the sample complexity can be bounded in terms of the sequential Rademacher complexity of this. Finally, the authors provide some interesting lower bounds showing that finding a global maximum is exponentially hard, and motivating them to study the sample complexity to find an approximate local maximum.

**Limitations And Societal Impact:**

Some limitations discussed but not in huge detail. There is no discussion of societal impact but since this is a theoretical paper, that is probably OK.

**Main Review:**

The discussion of the pros/cons of the UCB algorithm in lines 198-208 is nice as is the non-technical motivation of the algorithm in lines 219-228. However, I have very little intuition about why the loss function and definition of y_t used are the most appropriate/natural. It is nice to see a general result bounding the sample complexity in terms of the sequential Rademacher complexity in Theorem 1. In the examples following Theorem 3.1, it always seems that zeta_h^4 epsilon^{-8}> zeta^2_{3rd} epsilon^{-6}. Are there any realistic cases where the converse is true? Does their result ever lead to epsilon^{-6} rather than epsilon^{-8} complexity? The epsilon^{-8} dependence seems sub-optimal, I imagine it should be epsilon^{-2}  but do the authors have any intuition/lower bound to indicate what the optimal dependence on epsilon is in this setting? In the discussion about the application of their results for linear bandit in Section 3.1, it would be helpful to also include the dependence on epsilon of some of the other algorithms. It is difficult to compare fairly when the authors only highlight the terms that their algorithm improves on. In some cases it may be that d<1/epsilon so a worse dependence on d would be preferred to a worse dependence on epsilon in such a case. I am also curious about the results for sparse linear bandits when Theta is infinite? In such a case I imagine we may replace |Theta| with the covering number, which would then lead to linear dependence on d, do other approaches have worse dependence on d in that case too?

It is nice that there is an extension to reinforcement learning. The assumptions made in the extension to model-based reinforcement learning seem to reformulate the problem as a contextual bandit problem, thus making the extension simpler. However they seem quite strong, even though the authors give a simple example where they are satisfied. The loss function for the reinforcement learning setting seems slightly more natural than that in the bandit case. I would appreciate more discussion of how this is calculated within the algorithm though (e.g. in round t, do we need to play according to both policies pi_t and pi_{t-1}?) The loss function depends mainly on the transition, and so the sample complexity depends on the difficulty of learning some function of the transition function. However, the introduction to section 4 is written as if we can make an assumption on the value function only, but this may not necessarily lead to having a transition function st the Rademacher complexity of the loss is small. Instead it seems that the idea is that we assume the dynamics are members of some function class and then show that this leads to a value function of the desired form as in example 4.3. It would be helpful to clarify this.

The lower bounds in Section 5 are nice. In particular, the result in Theorem 5.1 showing that the sample complexity can be exponential in d when comparing to global maximum for one-layer neural networks is nice, and well motivates comparing to local maxima as is done in this paper. Similarly the result in theorem 5.2 is also nice (although it is proved concurrently in another paper). In theorem 5.3, is the sample complexity exponential when compared to the local maximum or global maximum? And is the only reason for the exponential sample complexity the fact that the action set is chosen to be exponentially large?

Lots of content seems to be deferred to the appendix making the main paper hard to follow at times.

Minor comments:
- It would be good to have a more thorough discussion of [54]. Saying the conditions can ‘likely’ not be satisfied is not rigorous enough.
- the discussion in lines 136-138 is incomplete and needs expanding upon. I thought the remark was to motivate the need to consider deterministic rewards, but Appendix C.1 seems to be about the difficulty of finding a global maximum. It is not clear to me how this relates to considering local maxima in a noisy setting.
- I don’t follow the discussion of why the number of random projections in line 234 does not depend on the dimension, can more details be given?
- I’m not sure what the ‘local regret’ means in line 21 is nor where it is defined.
- the definition of the loss function in (5) is quite involved. I therefore have no intuition for how difficult this will be to learn as a function of the original parameters of the problem and why this is an appropriate loss definition. Also what is kappa here?
- A couple of times in Section 3.1, the authors ‘claim’ that the sample complexity is bounded by XXX. Is this just a claim or is it proven?
- I don’t follow why it is beneficial to consider stochastic policies in Section 4, could more details be provided on this?


=========
After response:
Thank you to the authors for answering most of my questions. I hope that for the final version of the paper the authors clarify some of the discussions and add further details about the dependence on epsilon and the construction of the exponential lower bound.


**Time Spent Reviewing:**

9

---

> ### Author Response · Authors · 2021-08-10
> **Response to Reviewer XRrF**
>
> We thank the reviewer for the detailed review. We appreciate it that the reviewer finds our paper contains “a nice result”, and “the discussion of the pros/cons of the UCB algorithm ... is nice”. In the following, we address the reviewer’s comments in detail.
>
> > I have very little intuition about why the loss function and definition of y_t used are the most appropriate/natural.
>
> As discussed in the beginning of Section 3, we need to guarantee that (a) the virtual reward converges to the true reward, and (b) the algorithm achieves a high virtual reward. The first two terms of the loss function in Eq.(5) are to guarantee convergence (a). The gradient and Hessian terms of the loss function are used to guarantee (b): when the gradient and Hessian of the virtual reward are close to the true ones, the virtual reward will improve iteratively if the real reward is not yet near a local maximum.
>
> A more formal discussion of the loss function is given in Appendix D.1 (Proof Sketch of Theorem 3.1), and in particular, Eq. (15)-(18). In short, the loss function upper bounds the difference of real reward $\eta(\theta^\star,a_t)$ and virtual reward $\eta(\theta_t,a_t).$ So if the loss function converges to zero, the virtual reward must be close to the real reward, and optimizing virtual reward is equivalent to optimizing real reward.
>
> > The introduction to section 4 is written as if we can make an assumption on the value function only, but this may not necessarily lead to having a transition function s.t. the Rademacher complexity of the loss is small. It would be helpful to clarify this.
>
> The assumptions 4.1 and 4.2 on the value function are in addition to the assumption that the transition function class has a low Rademacher complexity. We demonstrate the feasibility of them in Example 4.3 mostly because Assumptions 4.1 and 4.2 are not obviously true. In example 4.3, the neural $NN_\theta$ is not specified, and if $NN_\theta$ belongs to a low complexity family of neural networks, then the loss function will also have low Rademacher complexity.
>
>
> > Why it is beneficial to consider stochastic policies in Section 4?
>
> Verifying Assumption 4.1 and 4.2 on stochastic policies is easier: we verify Assumption 4.1 by coupling the distribution of the next state (see Appendix F.3). On the other hand, if both the policy and the transition are deterministic, the next state is deterministic and the Lipschitzness of value function may blow up exponentially.
>
> > Does their result ever lead to $\epsilon^{-6}$ rather than $\epsilon^{-8}$ complexity? The $\epsilon^{-8}$ dependence seems sub-optimal;
>
> The leading term is  $\epsilon^{-8}$ when $\epsilon$ is reasonably small. But when $\epsilon>\zeta_h^2/\zeta_{3rd}$ the $\epsilon^{-6}$ term is larger. We mainly focus on high dimensional cases where $1/\epsilon \gg d$ and we didn’t optimize the polynomial dependency because it’s already highly non-trivial to prove a dimension-independent sample complexity.
>
> > What is the optimal dependence on $\epsilon$ is in this setting?
>
> The optimal dependence on $\epsilon$ is still an interesting open question. Due to the $\Omega(d/\epsilon^2)$ lower bounds in zero-order optimization literature (Proposition 1 of [4]), the optimal dependence on $\epsilon$ might be $1/\epsilon^2$.
>
> > It would be helpful to include the dependence on epsilon of some of the other algorithms [for linear bandits]
>
> The sample complexity of zero order optimization is $d/\epsilon^2.$ The sample complexity of the SquareCB algorithm (Algorithm 2 of [3]) is $d^2/\epsilon^2$. We remark that the SquareCB algorithm only works for rewards with linear structure, where our algorithm is designed for general rewards.
>
> > In [sparse linear bandit] I imagine we may replace |Theta| with the covering number, which would then lead to linear dependence on d. Do other approaches have worse dependence on d in that case too?
>
> We agree that the $\log |\Theta|$ term can be replaced by log covering number. But the log covering number is $O(s\log d)$ instead of $O(d)$, where $s$ is the sparsity level. This is because the total number of possible supports of a $s$ sparse vector is upper bounded by ${d \choose s}\le d^s.$
>
> As discussed in Section 3 (Line 263-270), algorithms based on the Eluder dimension (such as LinUCB) and gradient (such as zero order optimization) have a at least linear dependence on $d$ instead of $s$.
>
> > In theorem 5.3, is the sample complexity exponential when compared to the local maximum or global maximum?
>
> The sample complexity is exponential when compared to local maximum. In fact the UCB algorithm will keep exploring for an exponential number of steps. So, UCB doesn’t even converge to local max, indicating more sophisticated algorithms like ours are necessary.
>
> >  Is the only reason for the exponential sample complexity the fact that the action set is chosen to be exponentially large?
>
> The continuous action set is not the only reason for the exponential sample complexity. Our lower bound is a result of BOTH nonlinear reward and continuous action set. A continuous/exponential action set alone is not enough: linear bandits, even with continuous action, can be solved with polynomial samples.
>
> > how [the loss function] is calculated within the algorithm [in reinforcement learning setting]?
>
> The algorithm samples two trajectories $\tau_t$ and $\tau_{t}’$ according to policies $\pi_{\psi_t}$ and $\pi_{\psi_{t-1}}$ respectively. Then, the loss function is computed by Eq. (6).
>
> > It would be good to have a more thorough discussion of [54]. Saying the conditions can ‘likely’ not be satisfied is not rigorous enough.
>
> Please see the discussion by the authors of [54] below Theorem 3.1 (Page 5) in the newest update of [54]: https://arxiv.org/abs/1807.03858v5.
>
> > the discussion in lines 136-138 is incomplete … the remark was to motivate the need to consider deterministic rewards, but appendix C.1 seems to be about the difficulty of finding a global maximum.
>
> Line 136-138 is referring to Theorem C.1 in Appendix C.4. In Theorem C.1, we prove that under standard Gaussian noise, no algorithm can find a global or local maxima in less than d (dimension of action set) steps. So the dimension dependence is unavoidable when considering mild stochastic rewards.
>
> > why the number of random projections in line 234 does not depend on the dimension?
>
> This is similar to the intuition of Johnson–Lindenstrauss lemma. For simplicity, here we focus on finite hypotheses. In our loss function we only care about $\|\nabla \eta(\theta^\star,a)-\nabla \eta(\theta,a)\|^2$ (which is explained in Line 804-811, Appendix D.1). Therefore we only need $\log |\Theta|$ projections to accurately approximate $\|\nabla \eta(\theta^\star,a)-\nabla \eta(\theta,a)\|^2$ for any $\theta\in \Theta$. So the number of random projections is independent of the dimension.
>
> > I’m not sure what the ‘local regret’ means in line 251 is nor where it is defined.
>
> We define the local regret in Appendix D.5. Roughly speaking, the local regret is the regret compared with a local maximum.
>
> > the definition of the loss function in (5) is quite involved. I have no intuition for how difficult [the loss function] will be to learn.
>
> Indeed the augmented loss function is involved, and bounding the sequential Rademacher complexity (SRC) for the augmented loss class can be non-trivial. But we generally believe that the augmented loss has similar complexities as the original function class, and we provide several pieces of evidence in Appendix B (also listed below):
> - In the neural net example, the $(1,\infty)$-norm bound for the sequential Rademacher complexities of the gradient, hessian,and the augmented loss classes are the same as the best known bound of the neural net family.
> - If a neural net has $p$ parameters, its directional gradient and hessian can both be expressed by neural nets with $O(p)$ parameters.
> - For finite model class, the sequential Rademacher complexity for the augmented loss function classes are also bounded by log size of the hypothesis.
>
> > what is $\kappa$?
>
> $\kappa$ is truncate threshold of the loss function, defined in Line 1 of Algorithm 1 and explained in Line 830-810, Appendix D.1.
>
> > A couple of times in Section 3.1, the authors ‘claim’ that the sample complexity is bounded by XXX. Is this just a claim or is it proven?
>
> All the sample complexity results in Section 3.1 are proved in Appendix D.7.
>
> [1] Carpentier Alexandra, Munos Rémi. Bandit theory meets compressed sensing for high dimensional stochastic linear bandit.
>
> [2] Rakhlin, Alexander, Karthik Sridharan, and Ambuj Tewari. Online learning via sequential complexities.
>
> [3] Foster, Dylan, and Alexander Rakhlin. Beyond ucb: Optimal and efficient contextual bandits with regression oracles.
>
> [4] Duchi, John C., et al. Finite Sample Convergence Rates of Zero-Order Stochastic Optimization Methods.

---

### Official Review · Reviewer_uBiC · 2021-07-19

**Rating:** 7
**Confidence:** 4

**Summary:**

This paper studies model-based nonlinear bandit and reinforcement learning and proposes the ViOlin algorithm (Virtual Ascent with Online Model Learner). The main contributions are as follows.

1) For bandit problems, with deterministic reward functions satisfying Assumption 2.1 (Lipschitz over continuous actions, smooth over actions, and the third order derivative over action is bounded), and with sublinear sequential Rademacher complexity, they show that the ViOlin algorithm achieves an $\epsilon$ approximate local maximum using $\tilde{O}(\epsilon^{-8})$ iterations, which does not depend on the dimension of action space.

Some instantiations are shown, including linear bandit with finite model class, with sparse or structured model vectors, and two-layer neural nets, which is the first result of non-linear bandit with neural network parameterization.

The main idea is to fit the estimation of gradient and Hessian, and show that the current greedy action can be improved until the parameter is close enough to an approximate local maximum.

2) For RL, with deterministic transition, and Assumptions 4.1 and 4.2 (regularity of value function and policy), they show that the similar ViOlin (Algorithm 2) achieves an $\epsilon$ approximate local maximum using also sublinear iterations.

The difference with bandit design is that the loss function is using one-step model prediction errors rather than gradient and Hessian.

3) The authors show some lower bound results. In particular, (i) using ReLU activations in one-layer neural network bandits, finding global optimal action takes exponentially large iterations. (ii) the failure of Eluder dimension: one-layer neural networks has its Eluder dimensions exponentially large. (iii) the failure of UCB: with linear reward function, and two-layer neural network as the hypothesis function class, UCB would take exponential sample complexity to find the optimal action.

**Limitations And Societal Impact:**

The paper is mainly theoretical and the algorithm is not implemented. There is no specific potential negative societal impact of this work.

**Main Review:**

Overall, this is a good paper, making good contributions to RL theory community. The presentation of the main paper is clear and easy to follow. The intuition is very well explained. There are following questions I would ask the authors to clarify.

1) In lower bound results, it is mentioned (at least for one layer NNs) ReLU is needed. Does the "Two-layer neural nets" instantiation also apply to NNs with ReLU activations? It seems from "link function with bounded derivatives up to the third order", it does not apply to ReLU.

Then is it possible that for the link functions with "bounded derivatives up to the third order", finding a global optimal solution is also possibly efficient?

2) Is Hessian used for eliminating saddle points? What kind of results can be obtained if just using gradient?

3) The deterministic reward and deterministic transition assumption makes the algorithm kind of restrictive. Any idea how to remove those assumptions?

Minors:

4) What is the meaning of $\mathcal{R}$ in Algorithm 1 (line 3)?

5) What is the definition of $\mathfrak{A}$ in line 887? Does this mean the approximate local minima set?

======update after rebuttal======

Thank you for the feedback. I would keep my current rating and vote for accepting the paper.


**Time Spent Reviewing:**

7

---

> ### Author Response · Authors · 2021-08-10
> **Response to Reviewer uBiC**
>
> We thank the reviewer for the detailed review and summary. The reviewer notes that our paper “makes good contributions to RL theory community” and “the intuition is very well explained”. In the following, we address the reviewer’s comments in detail.
>
> > Does the "Two-layer neural nets" instantiation also apply to NNs with ReLU activations? Is it possible that for the link functions with "bounded derivatives up to the third order", finding a global optimal solution is also possibly efficient?
>
> The ReLU doesn’t have a bounded derivative up to third order, and therefore doesn’t satisfy the assumptions of our algorithm. But if we replace ReLU(x) by ReLU(x)$^3$, the lower bound for finding global maxima still holds (by virtually the same proof). That is, even if the link function has a bounded derivative up to third order, finding a global optimal solution is statistically hard.
>
> On the other hand, if we use ReLU(x)$^3$ as activation, our algorithm converges to a local max in polynomial steps (in fact, because the reward has a very large flat region, a random action is local max with high probability).
>
> > Is Hessian used for eliminating saddle points? What kind of results can be obtained if just using gradient?
>
> Yes, Hessian is used for escaping saddle points. In particular, we prove that the algorithm improves the reward when the current action is not an approximate local max (Case 2 of Lemma D.1, lines 849-856). Using just the gradient term, we can prove that our algorithm converges to an approximate critical point (that is, an action with gradient smaller than $\epsilon$).
>
> > The deterministic reward and deterministic transition assumption makes the algorithm kind of restrictive. Any idea how to remove those assumptions?
>
> The stochastic case is generally hard because the signal-to-noise ratio is very small even with standard Gaussian noise (see Appendix C.4). So to deal with stochastic rewards, we need to make restrictions on the type of noise. We leave this direction as an important future work.
>
> > What is the meaning of $\mathcal{R}$ in Algorithm 1 (line 3)?
>
> $\mathcal{R}$ is the online learner defined in Line 165. The online learner takes in a sequence of action-loss pairs and outputs a distribution of actions.
>
> > What is the definition of $\mathfrak{A}$ in line 887? Does this mean the approximate local minima set?
>
> Yes, $\mathfrak{A}_(\epsilon_g, \epsilon_h)$ is the set of all $(\epsilon_g, \epsilon_h)$-approximate local maximum, defined in Line 891.
>
> [1] Liu Sijia, Chen Pin-Yu, Kailkhura Bhavya, Zhang Gaoyuan, Hero III Alfred O, Varshney Pramod K. A Primer on Zeroth-Order Optimization in Signal Processing and Machine Learning: Principals, Recent Advances, and Applications.

---

### Official Review · Reviewer_n7MF · 2021-07-20

**Rating:** 7
**Confidence:** 4

**Summary:**

The paper proposes new nonlinear bandit and reinforcement learning algorithms whose sample complexity toward local maxima are bounded by the sequential Rademacher complexity of particular loss functions (and do not depend on the action dimension). An interesting insight obtained in this paper is that when dealing with nonlinear models and dynamics (in settings where the action dimension may be very large), maximizing the virtual return suggested by a strong online learner (with carefully augmented loss) may lead to good convergence guarantees to local maxima.

**Limitations And Societal Impact:**

The authors have discussed the potential negative societal impact in their Appendix A.

**Main Review:**

The paper is well-written. The theoretical results are sound and the proofs are correct. The problem studied in this paper is well-motivated and very interesting. The main algorithmic idea of augmenting the loss function of the online learner is quite clever. I recommend  acceptance.

On the downside, I would like to say that the algorithms and results in this paper are more like "proofs of concept", and some places of the paper are still clunky. I discuss several limitations and provide some suggestions below.

1. The paper seems to mainly focus on the "pure exploration" or PAC setting of bandits and RL. While there is some discussion on the "local regret" in Appendix D.5, it is just a slightly different analysis of the same algorithm (without additional efforts tailored for the regret setting). Do the authors have any ideas on whether one can get substantially better regret bounds by incorporating additional considerations for the regret setting? By the way, an interesting thing that I find in this paper is that the current algorithm already tries to avoid "over exploration", even in a "pure exploration" setting --- is it because of the fact that the paper only asks for convergence to local maxima? Do the authors have any ideas on whether the notion of "local regret" would encourage certain behavior of learning algorithms?

2. The assumptions on deterministic reward and deterministic dynamics are restrictive. While the authors provide some impossibility results on the noisy reward setting, a framework that can only deal with deterministic reward and dynamics are far from being practical. I would encourage the authors to improve their framework and try to understand this issue better in future work.

3. The theoretical bounds proved in this paper have a much worse dependence on $\epsilon$ compared with existing dimension-dependent bounds. This may make the statistical guarantees undesirable compared with dimension-dependent or specialized algorithms in many cases.

4. Since the algorithms need to *augment the loss functions to be non-standard ones* (see equation (5)), the tasks of bounding the sequential Rademacher complexity and designing computationally efficient online learners both become more challenging.

I notice that the authors actually have some very related discussions in Appendix B. It would be good if some key points of the discussions can also be mentioned in the main text.

Minor comments:
1. In lines 145-146, the notation $[T]$ appears without any definition (I understand what $[T]$ means but it is good to write everything clearly).
2. I find the following two references also relevant.
[1] Xu, Y., & Zeevi, A. (2020). Upper counterfactual confidence bounds: a new optimism principle for contextual bandits. arXiv preprint arXiv:2007.07876.
[2] Foster, D. J., Gentile, C., Mohri, M., & Zimmert, J. (2020). Adapting to misspecification in contextual bandits. NeurIPS.
Both [1] and [2] consider model-based nonlinear (contextual) bandits with infinite actions and obtain regret bounds that depend on the action dimension.

======update after rebuttal======

Thank you for the feedback. I keep my positive rating and vote for accepting the paper.

**Time Spent Reviewing:**

6

---

> ### Author Response · Authors · 2021-08-10
> **Response to Reviewer n7MF**
>
> We thank the reviewer for detailed review and constructive suggestions. The reviewer notes that our paper is “well-motivated and very interesting” and “augmenting the loss function of the online learner is quite clever”. We will incorporate minor issues and update references upon revision. In the following, we address the reviewer’s comments in detail.
>
> > The paper seems to mainly focus on the ‘pure exploration’ or PAC setting. While there is some discussion on the ‘local regret’ in Appendix D.5, it is … the same algorithm. Do the authors have any ideas on whether the notion of "local regret" would encourage certain behavior of learning algorithms?
>
> We appreciate it that the reviewer finds “the current algorithm already tries to avoid over exploration even in a pure exploration setting”. In fact, our algorithm was first designed and analyzed for the regret minimization setting, and we chose to mainly present the “PAC setting”  only for simpler exposition.
>
> We believe that the notion of “local regret” encourages the algorithm to improve its output gradually instead of explore-then-commit -- just like the “regret” notation in standard bandit settings. Our algorithm indeed tries to optimize its output $a_t$ in each step (Line 4 of Alg. 1) to actively exploit. In contrast, pure exploration algorithms typically focus on collecting information. Formally, Lemma D.1 proves that the action is improved in each step unless the action is already a local max.
>
> >  I would encourage the authors to … understand [why they require deterministic rewards]  in future work.
>
> We thank the reviewer for the constructive suggestion. As a starting point, in Appendix C.4 we prove that no algorithm can achieve dimension-free sample complexity with standard Gaussian noise. On the other hand, the deterministic assumption is only used to compute the projection of gradient and Hessian by finite difference. We discuss in Appendix B that our method could be extended to stochastic rewards with multiple-point feedback, which is a standard setting in zero-order optimization literature [1].
>
> > “The theoretical bounds proved in this paper have a much worse dependence on $\epsilon$.”
>
> Indeed, the polynomial dependency on $\epsilon$ is far from optimal. We didn’t optimize the dependency because it’s already challenging to achieve a polynomial dependency. We believe this is partly an artifact of our analysis because of several loose reductions. Improving the dependence on $\epsilon$ is left as an important future work.
>
> > “The algorithms need to augment the loss ... the tasks of bounding the sequential Rademacher complexity and designing computationally efficient online learners both become more challenging.”
>
> We agree that bounding the sequential Rademacher complexity (SRC) for the loss class can be non-trivial. But we generally believe that the class of gradients, hessians, and the augmented losses have similar complexities as the original hypothesis class, and we provide several pieces of evidence in Appendix B (also listed below):
> - In the neural net example, the $(1,\infty)$-norm bound for the sequential Rademacher complexities of the gradient, hessian,and the augmented loss classes are the same as the best known bound of the neural net family.
> - If a neural net has $p$ parameters, its directional gradient and hessian can both be expressed by neural nets with $O(p)$ parameters.
> - For finite model class, the sequential Rademacher complexity for the augmented loss function classes are also bounded by log size of the hypothesis.
>
> There are also attempts to design computationally efficient online learners to achieve SRC bounds [2], but for general function classes this question is still open and of independent interest.
>
> [1] Liu Sijia, Chen Pin-Yu, Kailkhura Bhavya, Zhang Gaoyuan, Hero III Alfred O, Varshney Pramod K. A Primer on Zeroth-Order Optimization in Signal Processing and Machine Learning: Principals, Recent Advances, and Applications.
>
> [2] A. Rakhlin, O. Shamir, and K. Sridharan. Relax and randomize: From value to algorithms.

---

### Decision · Program_Chairs · 2021-09-27

**Decision:**

Accept (Poster)

**Comment:**

This paper takes a new approach to the theory of exploration in reinforcement learning with function approximation, with the main insight being to aim for local optimality rather than global optimality. The reviewers agree that the paper is well-motivated, technically interesting,
 and fairly creative result that will likely inspire further research along this direction in RL. While there are many directions in which the technical results themselves can likely be improved (e.g., moving beyond deterministic dynamics), it seems appropriate to leave this for future work.

The authors are encouraged to incorporate the clarifications suggested by the reviewers.